# RoboMME: Benchmarking and Understanding Memory for Robotic Generalist Policies

**Yinpei Dai** [1]   **Hongze Fu** [1]   **Jayjun Lee** [1]   **Yuejiang Liu** [2]   **Haoran Zhang** [1]   **Jianing Yang** [3]   **Chelsea Finn** [2]
**Nima Fazeli** [1 †]   **Joyce Chai** [1 †]

## Abstract

Memory is critical for long-horizon and history-dependent robotic manipulation. Such tasks often involve counting repeated actions or manipulating objects that become temporarily occluded. Recent vision-language-action (VLA) models have begun to incorporate memory mechanisms; however, their evaluations remain confined to narrow, non-standardized settings. This limits systematic understanding, comparison, and progress measurement. To address these challenges, we introduce **RoboMME**: a large-scale standardized benchmark for evaluating and advancing VLA models in long-horizon, history-dependent scenarios. Our benchmark comprises 16 manipulation tasks constructed under a carefully designed taxonomy that evaluates *temporal*, *spatial*, *object*, and *procedural* memory. We further develop a suite of 14 memory-augmented VLA variants built on the $\pi_{0.5}$ backbone to systematically explore different memory representations across multiple integration strategies. We show that the effectiveness of memory representations is highly task-dependent, with each design offering distinct advantages and limitations across different tasks. Videos and code can be found at https://robomme.github.io/

## 1. Introduction

Open-world robotic manipulation often requires reasoning over history and recalling information from past interactions.

For instance, a household robot may be asked to return books to their original positions on a shelf, wipe a table for a specified number of cleaning passes, or fold laundry after observing a human demonstration. In such scenarios, relying solely on immediate perception for action prediction is insufficient. Effective execution depends on the ability to retain and reuse relevant information across time, which we broadly refer to as *memory*.

Prior work incorporates memory into robotic manipulation policies through three main representations: (1) *symbolic memory*, which summarizes history using non-differentiable abstractions, such as point-tracking trajectories (Chen et al., 2026) and language-based subgoals (Sridhar et al., 2026; Torne et al., 2026); (2) *perceptual memory*, which represents history as a group of visual features, including multi-frame visual tokens (Jang et al., 2025) and memory banks (Fang et al., 2025; Shi et al., 2026); and (3) *recurrent memory*, which compresses contextual features into fixed-size latent states through recurrent models (Zhou et al., 2025; Liu et al., 2024). While demonstrating the importance of memory, these methods rely on different policy backbones and inconsistent evaluation protocols, making it unclear which memory designs generalize across tasks. Progress is further limited by the absence of benchmarks that capture diverse and challenging memory requirements. Memory-Bench (Fang et al., 2025) is the first benchmark to explicitly evaluate spatial memory, but it contains only three near-solved tasks. MIKASA-Robo (Cherepanov et al., 2025) introduces several history-dependent tasks, yet they remain short-horizon and lack sufficient high-quality demonstrations for effective vision-language-action (VLA) imitation learning. Consequently, existing benchmarks neither capture realistic memory demands nor provide a standardized testbed for systematically evaluating memory-augmented manipulation policies.

To address these limitations, we present **RoboMME**: a unified large-scale **Robo**tic simulation benchmark designed for **M**emory-augmented **M**anipulation **E**valuation. Drawing inspiration from cognitive theories of human memory (Atkinson & Shiffrin, 1968), RoboMME categorizes memory into four cognitive dimensions: (1) *temporal memory* for event accumulation and ordering; (2) *spatial memory* for

---

[1]University of Michigan, Ann Arbor, MI, USA [2]Stanford University, Stanford, CA, USA [3]Figure AI, Sunnyvale, CA, USA. [†]Equal Advising. Correspondence to: Yinpei Dai <daiyp@umich.edu>, Hongze Fu <hongzefu@umich.edu>, Nima Fazeli <nfz@umich.edu>, Joyce Chai <chaijy@umich.edu>.

*Proceedings of the 43rd International Conference on Machine Learning*, Seoul, South Korea. PMLR 306, 2026. Copyright 2026 by the author(s).

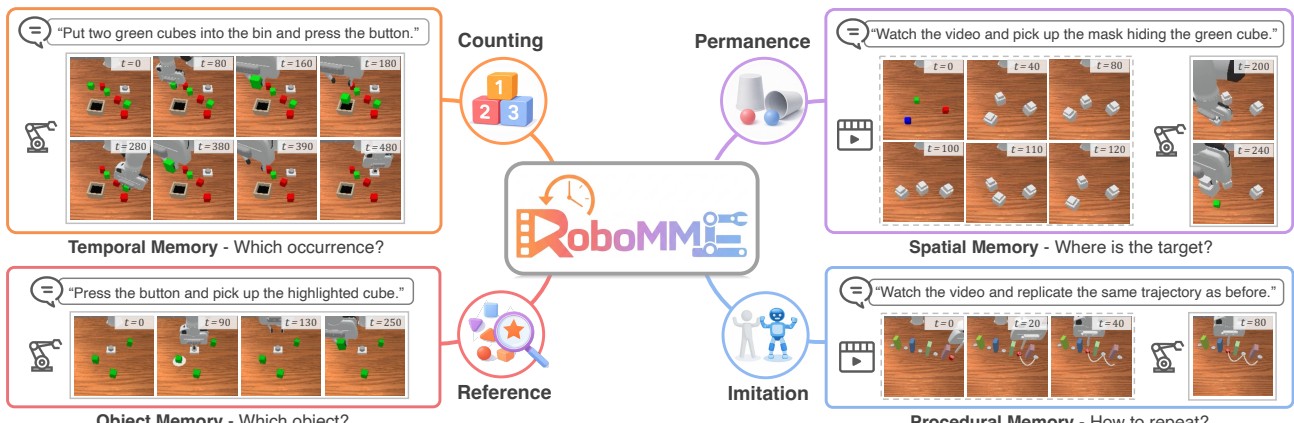

*Figure 1.* **RoboMME** is a large-scale robotic benchmark for evaluating memory-augmented manipulation, comprising four task suites that emphasize distinct memory demands. (1) The `Counting` suite targets *temporal memory*, requiring robots to accumulate and reason over past events, e.g., counting placed green cubes and stopping correctly (top-left). (2) The `Permanence` suite focuses on *spatial memory*, requiring tracking of object locations under occlusion and environmental changes, e.g., resolving the correct mask after green and blue cubes swap positions (top-right). (3) The `Reference` suite evaluates *object memory*, requiring identification under varied referential cues, e.g., recalling a briefly highlighted cube during a button press (bottom-left). (4) The `Imitation` suite assesses *procedural memory*, measuring the ability to reproduce previously demonstrated behaviors, e.g., replicating a demonstrated trajectory (bottom-right). Overall, RoboMME includes 16 tasks with 770k high-quality training timesteps for systematic evaluation of memory-augmented policies.

tracking object locations under occlusion and scene change; (3) *object memory* for resolving referential identity across time; and (4) *procedural memory* for reproducing previously demonstrated motion patterns. These memory types define four corresponding task suites, `Counting`, `Permanence`, `Reference`, and `Imitation`, respectively, each composed of four carefully designed tasks. In total, RoboMME contains 16 diverse long-horizon tasks with 1,600 demonstrations, yielding 770k high-quality timesteps for comprehensive evaluation of memory-augmented policies.

Building on RoboMME, we develop a family of 14 memory-augmented VLA models based on the $\pi_{0.5}$ backbone (Black et al., 2025) to systematically study how different memory representations influence manipulation performance. Symbolic memory is implemented as language subgoals concatenated with task instructions, explicitly encoding history in natural language without modifying the backbone. Differentiable neural representations, including perceptual and recurrent memory, are integrated through three mechanisms: (1) *memory-as-context*, which appends memory embeddings to the inputs for joint processing; (2) *memory-as-modulator*, which conditions the action expert via adaptive LayerNorm (Peebles & Xie, 2023) to modulate intermediate activations; and (3) *memory-as-expert*, which adds a dedicated memory expert that interacts with the action expert through blockwise causal attention (Black et al., 2024).

Interestingly, our experiments show that no single memory representation or integration strategy consistently performs well across all tasks, indicating that conclusions from prior work may not generalize. Instead, effectiveness is highly

task-dependent: symbolic memory excels at counting and short-horizon reasoning, while perceptual memory is critical for time-sensitive and motion-centric behaviors. Among all variants, perceptual memory combined with the memory-as-modulator design achieves the best balance between performance and computational efficiency. Together, these results establish the first comprehensive framework for understanding memory in manipulation, and represent a step toward reliable, long-horizon, history-dependent robotic generalist policies.

## 2. Related Work

**Memory-based Robotic Manipulation Benchmarks.** Most prior robotic manipulation benchmarks (Liu et al., 2023; Li et al., 2024b; Mees et al., 2022; Zhu et al., 2020) implicitly involve temporal memory but rarely require it: policies conditioned only on the current observation can already achieve high success rates (Black et al., 2025; Wang et al., 2026b; Dai et al., 2025b). This suggests that success often relies on local perception rather than true history-based reasoning. For example, object states are continually updated during online execution, allowing the agent to infer the next subtask without relying on past observations. MemoryBench (Fang et al., 2025) is the first attempt to introduce explicit memory requirements, focusing on spatial location recall, but its tasks remain simple and nearly solved. MIKASA-Robo (Cherepanov et al., 2025), originally designed for reinforcement learning, includes some memory-related tasks, yet these tasks remain short in horizon, are limited in long-term dependencies, and lack sufficient high-

quality demonstrations for effective imitation learning. Recent memory-based policies (Shi et al., 2026; Sridhar et al., 2026) are typically evaluated on narrow, self-designed tasks, hindering systematic comparison and understanding. In contrast, RoboMME provides a unified and challenging testbed with *explicit* history dependence, systematically spanning diverse memory types and offering sufficient demonstrations to support large-scale imitation-based training and evaluation.

**Memory-based Robotic Manipulation Models.** Existing approaches can be broadly categorized according to their memory representations. (1) *Symbolic memory* relies on non-differentiable abstractions derived from interaction history or external modules. For example, HistRISE (Chen et al., 2026) tracks object-centric 3D points as symbolic states, while UniVLA (Bu et al., 2025) incorporates temporal context by adding past actions to input prompts. More recent methods, such as MemER (Sridhar et al., 2026), Gemini-Robotics-1.5 (Team et al., 2025), and MEM (Torne et al., 2026), use large vision-language models (VLMs) to generate language-based subgoals or memory summaries, thereby offloading long-term memory to external reasoning pipelines. However, this modular design incurs costly inference and prevents end-to-end optimization. (2) *Perceptual memory* represents history as differentiable visual or multimodal features. ContextVLA (Jang et al., 2025), for example, appends past visual tokens directly to the transformer input as raw contextual tokens. In contrast, memory-bank approaches encode visual features together with auxiliary signals, such as task instructions (Shi et al., 2026; Koo et al., 2026) or action heatmaps (Fang et al., 2025), and cache these multimodal embeddings for subsequent retrieval. (3) *Recurrent memory* compresses history into fixed-size latent states through iterative updates. Early work, such as BC-RNN (Mandlekar et al., 2021), models temporal dependencies using recurrent neural networks. More recent approaches, including MITL (Zhou et al., 2025) and RoboMamba (Liu et al., 2024), adopt Mamba-style state-space models (Gu & Dao, 2024) to better capture long-horizon dependencies. Despite showing the importance of memory, these approaches vary widely in architecture and evaluation, making systematic comparison difficult. To address this gap, we construct a family of memory-augmented VLA models on the same backbone and evaluate diverse memory representations and integration strategies under a controlled, standardized setting.

## 3. RoboMME Benchmark

The aim of RoboMME is to rigorously evaluate history-dependent behavior in robotic manipulation. All tasks are intentionally constructed to be non-Markovian, requiring models to reason over past observations that are no longer visible at the current step. Therefore, memory is essential

for these tasks because identical observations can arise from different histories yet require different actions.

### 3.1. Cognitively-Motivated Task Design

To systematically evaluate history-dependent, long-horizon manipulation, we ground our task design in established cognitive models of human memory. Classical theories (Atkinson & Shiffrin, 1968) divide long-term memory into *procedural* and *declarative* types, with declarative memory further divided into *episodic* and *semantic* forms. In particular, episodic memory supports recall of events and experiences, including temporal order, spatial context, and object identity (Burgess et al., 2002; Hasselmo et al., 2017; Babb & Johnson, 2011), whereas procedural memory encodes motor skills acquired through practice (Squire, 2004). Both are essential and relevant for memory-augmented manipulation, motivating the four memory types defined below.

- **Temporal memory**: Tracks event counts, sequence ordering, and transition conditions across steps, e.g., determining when to proceed to the next subtask.

- **Spatial memory**: Maintains object locations and spatial relationships, especially when current visual information becomes unreliable due to occlusion or scene changes.

- **Object memory**: Preserves referential consistency over time, enabling reliable object identification from visual, language, or action cues.

- **Procedural memory**: Enables the policy to reproduce motion patterns or manipulation behaviors observed in prior demonstrations.

These four memory types correspond to the cognitive dimensions of when (temporal), where (spatial), what (object), and how (procedural). RoboMME is organized around these dimensions into four task suites, `Counting`, `Permanence`, `Reference`, and `Imitation`, each emphasizing a primary memory type for controlled evaluation. Figure 1 illustrates an overview of RoboMME. Specifically, the `Counting` suite targets *temporal memory* by requiring agents to repeat actions a specified number of times, including pick-and-place, linear swinging, and time-critical tasks. The `Permanence` suite focuses on *spatial memory*, evaluating object location tracking from pre-recorded videos or during concurrent manipulation. The `Reference` suite evaluates *object memory* through persistent identity resolution under visual, action-based, and language-based referential cues. The `Imitation` suite targets *procedural memory* by requiring agents to reproduce demonstrated motion patterns, such as pushing, inserting, and sequential linear or circular motions. Together, these suites provide a complementary evaluation of memory-augmented manipulation across diverse memory demands. Table 1 summarizes the tasks, with detailed descriptions provided in Appendix E.

*Table 1.* **Task Summary**. Overview of all tasks, their corresponding memory types, average total timesteps, and key challenges.

| Task Name | Memory Type | Avg. #Steps | Task Challenge | Brief Description |
|---|---|---|---|---|
| **Task Suite:** Counting | | | | |
| PickXTimes | T | 538 | count | Pick and place a cube of a given color for a specified number of repetitions. |
| BinFill | T | 604 | count | Place a specified number of cubes of a given color into the bin as the cubes appear over time. |
| SwingXTimes | T | 435 | count, swing-motion | Swing a cube back and forth between two targets for a specified number of cycles. |
| StopCube | T | 317 | count, time-critical | Press a button *exactly* when a moving cube reaches the target at a specified occurrence. |
| **Task Suite:** Permanence | | | | |
| VideoUnmask | S | 217 | occlusion | Given a video in which all cubes are masked, uncover the cube of a specified color. |
| ButtonUnmask | S | 267 | occlusion | Press the button, during which all cubes are masked, then uncover the cube of a specified color. |
| VideoUnmaskSwap | S | 348 | occlusion, tracking | Given a video in which all cubes are masked and containers dynamically swap positions, uncover the cube of a specified color. |
| ButtonUnmaskSwap | S | 400 | occlusion, tracking | Press the button, during which all cubes are masked and containers dynamically swap positions, then uncover the cube of a specified color. |
| **Task Suite:** Reference | | | | |
| PickHighlight | O | 346 | visual-referential | Pick up all cubes that were visually highlighted in a short time during interaction. |
| VideoRepick | O + T | 687 | action-referential, tracking, count | Given a video showing a cube being manipulated and relocated, pick up the same cube for a specified number of repetitions. |
| VideoPlaceButton | O + T | 974 | language-referential, long-video, tracking | Given a video with interleaved cube placement and button pressing, place the cube on the target specified by a language-described temporal reference (e.g., *the target after pressing the button*). |
| VideoPlaceOrder | O + T | 1134 | language-referential, long-video, tracking | Given a video showing cube placement across multiple targets, place the cube on the target specified by a language-described ordinal reference (e.g., *the second target*). |
| **Task Suite:** Imitation | | | | |
| MoveCube | P | 394 | contact-mode, tool-use | Given a video showing cube transport, replicate the same demonstrated manipulation strategy (e.g., *pick-and-place, pushing with the gripper, or hooking with a stick*). |
| InsertPeg | P + O | 479 | precise-motion | Given a video showing peg insertion, grasp the same peg at the same end and insert it into a box following the same demonstrated direction. |
| PatternLock | P | 208 | linear-motion | Given a video showing a linear moving pattern, reproduce the same trajectory on the targets. |
| RouteStick | P | 370 | circular-motion | Given a video showing a circular routing pattern, reproduce the same trajectory around sticks. |

## 3.2. Benchmark Construction

We build on the ManiSkill simulator (Mu et al., 2021) using a tabletop environment with a 7-DOF Franka Panda arm. Training episodes are generated by replaying predefined keyframe waypoints and are recorded as dense trajectories.

**Observations and Actions.** At each timestep, the robot receives multi-view RGB observations from front- and wrist-mounted cameras (both $256 \times 256$), along with proprioceptive states including joint positions, end-effector (EEF) pose, and gripper state. The action space is defined in either absolute joint space or absolute EEF space: joint-space actions are 8D (7 joints + gripper), EEF-space actions are 7D (3D position, Euler orientation, and gripper). Tasks in the Imitation suite and those prefixed with "Video" use video-based observations (a sequence of historical frames with paired proprioception) at the initial step, whereas all other tasks use image-based observations (a single current frame with paired proprioception). During execution, all tasks provide image-based observations at every timestep.

**Data Curation.** To enhance behavioral diversity, particularly failure recovery, which is important for imitation learning (Dai et al., 2025a), we inject controlled perturbations during data generation by adding 5% noise to keyframe waypoints and then recovering to the normal trajectory using the original waypoints. Tasks are further stratified into easy,

medium, and hard levels based on scene clutter, horizon length, and environmental dynamics (details are provided in Appendix E). To ensure data quality, we discard episodes in which the built-in planner fails, retaining only successful rollouts for training and environment seeds for evaluation. The final dataset spans 16 tasks with 100 episodes each, yielding 1,600 demonstrations and 770k timesteps. Individual episodes range from a few hundred to over one thousand steps, reflecting long-horizon, history-dependent behaviors.

**Benchmark Comparison.** Table 2 provides a detailed comparison with prior benchmarks. Most existing benchmarks (Han et al., 2025; Zhang et al., 2025; Mees et al., 2022) are designed to emphasize long-horizon planning and task complexity. While these tasks may implicitly require temporal reasoning, they do not explicitly enforce other types of memory, as decisions can typically be made from immediate observations alone. In contrast, RoboMME introduces greater environmental complexity and video-conditioned tasks, and is the first to systematically cover four types of memory: temporal[1], spatial, object, and procedural. This design enables a more comprehensive evaluation of memory-augmented robotic policies.

---

[1]The taxonomy in MIKASA (Cherepanov et al., 2025) defines concepts such as sequential memory and memory capacity, which can be viewed as specific instances of our broader notion of temporal memory.

*Table 2.* **Benchmark Comparison.** The upper section lists general-purpose benchmarks and the lower section lists memory-based benchmarks. Memory types include temporal (T), spatial (S), object (O), and procedural (P). Environment complexity captures non-Markovian structure, partial observability, and dynamic scene change. Condition denotes initial inputs (Vid. for video observations; Lang. for language instructions). Annotation indicates the availability of language subgoals or keyframe timesteps for auxiliary supervision.

| Dataset | Environment Complexity | | | Condition | | Annotation | | Memory Type | Total #Tasks | Total #Demos | Avg. #Steps |
|---|---|---|---|---|---|---|---|---|---|---|---|
| | Non-Markov | Partial Obs. | Dynamic | Vid. | Lang. | Subgoal | Keyframe | | | | |
| RLBench18 (James et al., 2020; Shridhar et al., 2023) | ✗ | ✗ | ✗ | ✗ | ✓ | ✗ | ✓ | T | 18 | 1,800 | 137 |
| CALVIN (Mees et al., 2022) | ✗ | ✗ | ✗ | ✗ | ✓ | ✓ | ✓ | T | 34 | 1,000 | 584 |
| LIBERO (Liu et al., 2023) | ✗ | ✗ | ✗ | ✗ | ✓ | ✗ | ✗ | T | 130 | 6,500 | 162 |
| VLABench (Zhang et al., 2025) | ✗ | ✗ | ✗ | ✗ | ✓ | ✗ | ✗ | T | 100 | 1,600 | 115 |
| RoboCerebra (Han et al., 2025) | ✗ | ✗ | ✓ | ✗ | ✓ | ✓ | ✗ | T | 100 | 1,000 | 2,972 |
| MemoryBench (Fang et al., 2025) | ✓ | ✗ | ✗ | ✗ | ✓ | ✗ | ✓ | T+S | 3 | 300 | 312 |
| MIKASA-robo (VLA) (Cherepanov et al., 2025) | ✓ | ✓ | ✓ | ✗ | ✗ | ✗ | ✗ | T+S+O | 12 | 1,250 | 72 |
| **RoboMME (Ours)** | ✓ | ✓ | ✓ | ✓ | ✓ | ✓ | ✓ | **T+S+O+P** | **16** | **1,600** | **481** |

# 4. Memory-Augmented Manipulation Policies

Building on RoboMME, we construct a family of memory-augmented vision-language-action (VLA) models based on the $\pi_{0.5}$ backbone (Black et al., 2025), collectively termed the **MME-VLA suite**. We systematically compare different memory representations and integration mechanisms under controlled settings, as illustrated in Figure 2. More detailed model formulations are provided in Appendix A.

## 4.1. Memory Representations

We study three representations: *symbolic*, *perceptual*, and *recurrent*. Symbolic memory uses discrete language tokens, while the latter two provide differentiable neural features. These representations differ in both the form of stored history and how they can be used by the downstream policy.

**Symbolic Memory** represents task history as interpretable language subgoals, since accurate subgoals can serve as compact and effective summaries of past observations and actions. Some methods, such as MEM (Torne et al., 2026), further incorporate long reasoning traces during memory generation. However, such traces are often implementation-dependent and difficult to standardize. We therefore adopt a simpler subgoal-only memory formulation. At each step, we use an auxiliary vision-language model (VLM) conditioned on the current image and previous subgoals to generate the next subgoal. We consider both *simple* instructions and *grounded* variants that additionally include front-view image coordinates, e.g., *pick up the green cube* vs. *pick up the green cube at [63, 152]*. This grounded formulation is particularly useful for spatial reasoning. Notably, symbolic memory methods typically require additional subgoal annotations for training the VLM.

**Perceptual Memory** represents task history as a sequence of visual tokens selected from past images and extracted by the $\pi_{0.5}$ vision encoder. Compared with symbolic memory, perceptual memory preserves richer low-level visual information, including object appearance, spatial layout, and motion cues, which can be difficult to fully capture

with language subgoals. Since retaining all historical visual tokens would substantially increase the context length, we adopt two selection strategies: (1) token dropping (Yao et al., 2025), which removes temporally redundant image patches based on RGB differences; and (2) uniform frame sampling (Chen et al., 2024), which evenly downsamples the history and concatenates tokens from the sampled frames. These strategies capture complementary trade-offs: token dropping emphasizes locally informative visual changes, while frame sampling preserves a more global temporal view of the episode. In practice, we find that visual tokens alone are sufficient to encode the relevant history, without explicitly incorporating proprioceptive states.

**Recurrent Memory** compresses the historical visual token sequence into fixed-size latent states through recurrent updates. Unlike perceptual memory, which stores selected visual tokens directly, recurrent memory maintains a bounded memory representation whose size does not grow with episode length. We adopt two models: (1) Test-Time Training (TTT) (Sun et al., 2025; Zhang et al., 2026), which represents memory through online-updated fast weights optimized with a self-supervised loss and then applies these weights to generate output features; and (2) Recurrent Memory Transformers (RMT) (Bulatov et al., 2022; 2023), which process the input sequence into segments and recurrently update a set of learnable memory slots for each segment using a transformer. These recurrent mechanisms provide a compact, differentiable way to summarize long-horizon history for downstream action prediction.

## 4.2. Memory Integration Mechanisms

Symbolic memory can be naturally integrated into $\pi_{0.5}$ by concatenating subgoals with task instructions, since both are represented as language tokens. In contrast, perceptual and recurrent memory produce neural *memory tokens*, which require architectural modifications to connect historical information with action prediction. In the following, we study three integration mechanisms.

**Memory-as-Context.** Memory tokens are concatenated

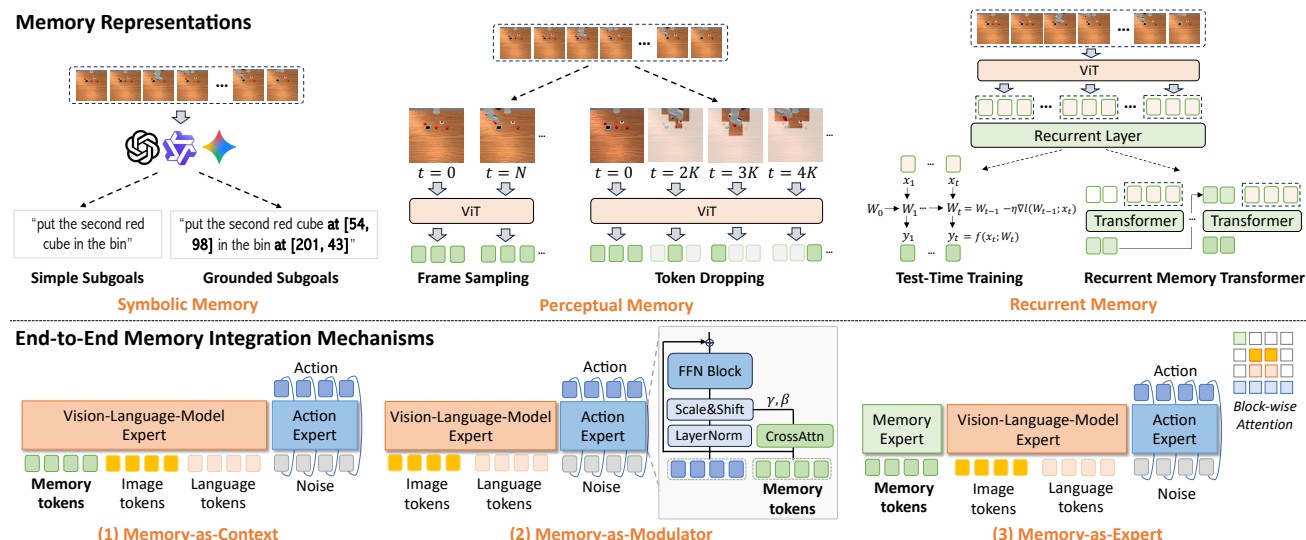

*Figure 2.* **Framework of MME-VLA Suite.** The top part illustrates three memory representations, each with two instantiations: (1) *Symbolic Memory* summarizes past interactions as high-level abstractions via language-based subgoals, optionally grounded to image pixels; (2) *Perceptual Memory* encodes history as raw visual tokens, using either token dropping to remove redundancy or uniform frame sampling to preserve essential context; (3) *Recurrent Memory* compresses history into fixed-size latent states through test-time training or recurrent memory transformers. The bottom part shows three integration strategies consistent with $\pi_{0.5}$: (1) *Memory-as-Context* concatenates memory tokens with observation tokens (e.g., images, task instructions, and proprioception); (2) *Memory-as-Modulator* applies adaptive LayerNorm to modulate the action expert, where action features cross-attend to memory tokens via multi-head attention; (3) *Memory-as-Expert* adds a separate memory expert that interacts with other experts through block-wise causal attention.

with the original input tokens and jointly processed by the VLM expert. This is the simplest integration strategy and requires minimal architectural changes, as memory is treated as additional context alongside the current observation and task instruction. Since the VLM expert processes these tokens together, memory can directly influence the visual-language features passed to the action expert. However, this design also mixes memory with the current observation stream, which may increase context length and introduce interference with the original VLM representations.

**Memory-as-Modulator.** We adopt adaptive LayerNorm (AdaLN) (Peebles & Xie, 2023) to condition the action expert on external memory. Before entering the feed-forward block in each layer, the action features cross-attend to memory tokens via multi-head attention to extract memory-aware representations. These representations are then projected into scale and shift parameters that modulate the normalized action features through AdaLN. This design injects memory directly into the action pathway while keeping the main VLM stream unchanged. As a result, memory acts as a conditioning signal for action prediction rather than as additional input context.

**Memory-as-Expert.** We introduce a lightweight memory expert that processes memory tokens through a dedicated pathway. The three experts interact via a specified block-wise causal attention scheme: the action expert attends to

both the VLM and memory experts, while the VLM and memory experts do not attend to each other. This asymmetric communication allows action prediction to use both current observation features and historical memory features, while preventing memory tokens from perturbing the VLM representations. This separation limits cross-expert interference, preserves the original VLM behavior, and allocates specialized capacity for memory processing.

## 5. Experiments

In this section, we systematically evaluate all MME-VLA variants under controlled settings, first outlining the experimental setup and then analyzing the main results.

### 5.1. Experiment Setup

We evaluate a total of 14 VLA policies in the MME-VLA suite, along with 4 prior methods, for a systematic comparison. All models are trained in a *multi-task* learning setting. We use an action prediction chunk size of 20 steps during training and execute the first 16 steps during evaluation. All models are trained for 80k steps with a batch size of 64, except for recurrent-memory variants, which use a batch size of 16 due to higher memory requirements. More implementation details are provided in Appendix B.

**MME-VLA Suite.** For symbolic memory, we fine-tune two

$\pi_{0.5}$ variants, SIMPLESG and GROUNDSG, which condition on simple or grounded language subgoals. Subgoals are generated by either Gemini-2.5-Pro (`Gemini`), a Qwen3-VL-4B model (`QwenVL`), or the simulator ground truth (`Oracle`). `QwenVL` is fine-tuned on subgoal annotations in our dataset to predict the next subgoal given the current image and the accumulated subgoal history, whereas `Gemini` relies solely on prompt engineering without task-specific fine-tuning. For perceptual memory, we use token dropping (TOKENDROP) or frame sampling (FRAMESAMP), each combined with three integration mechanisms: memory-as-context (`Context`), memory-as-modulator (`Modul`), and memory-as-expert (`Expert`), yielding six additional $\pi_{0.5}$ variants. For recurrent memory, we adopt TTT or RMT with the same integration strategies, also producing six additional policies. Unless otherwise specified, we denote each variant as "METHOD+`Integration`/VLM", e.g., FRAMESAMP+`Modul` or SIMPLESG+`QwenVL`.

**Evaluation Protocols.** For fair comparison across MME-VLA models, we fix the memory budget to 512 tokens, matching the number of input visual tokens from the current observation used by $\pi_{0.5}$. For example, both the learnable memory slots in RMT and the selected visual tokens in TOKENDROP are capped at this limit. We evaluate each model on all 16 tasks using 50 episodes per task, for a total of 800 episodes, with predefined environment seeds distinct from training. Each episode has a maximum horizon of 1,300 steps. Results are averaged over the last three checkpoints and three random seeds (nine runs in total).

**Prior Methods.** We compare against four methods: (1) $\pi_{0.5}$, which serves as a memory-free baseline and conditions only on the current observation and task instruction; (2) $\pi_{0.5}$ w/ past actions, which concatenates past actions with language tokens to form an explicit symbolic memory, inspired by UniVLA (Bu et al., 2025); (3) SAM2Act+ (Fang et al., 2025), which uses SAM2 as the visual backbone and maintains a memory bank for perceptual tracking, then predicts discrete keyframe actions that are executed by an external motion planner; and (4) MemER (Sridhar et al., 2026), which uses a VLM to infer symbolic subgoals from accumulated keyframe images and executes these subgoals with a VLA policy. MemER differs from our symbolic memory variants in how history is provided to the subgoal predictor. Our symbolic variants condition on only the current image and previously generated subgoals at each step, forming an accumulated language-oriented memory. In contrast, MemER selects keyframes from the most recent $N$ images during execution and maintains a buffer of previously selected keyframes as part of the VLM input. The VLM then predicts symbolic subgoals from this accumulated visual-oriented memory. Thus, MemER can be viewed as a hybrid memory method that combines perceptual memory, through stored keyframe images, with symbolic memory, through generated language-based subgoals. When reproducing MemER, we follow its original prompting strategy and fine-tune an additional Qwen3-VL-4B model using grounded subgoal and keyframe annotations. The predicted subgoals are then executed by our GROUNDSG policy. All prior methods are evaluated over three runs. We leave the inclusion of additional methods to future work.

### 5.2. Main Results and Analysis

The main results are summarized in Table 3; and complete results are provided in Appendix C. We analyze these results by addressing several key research questions below.

**Q1: Which memory representations and integration mechanisms yield the strongest performance?** Across all MME-VLA variants, **perceptual memory methods perform best**. In particular, FRAMESAMP+`Modul` achieves the highest overall success rate (44.51%), and most perceptual memory variants outperform the best symbolic and recurrent counterparts. Within perceptual memory, TOKENDROP is less effective than FRAMESAMP, likely because aggressive token pruning removes global spatial context. This loss can degrade performance on tasks such as *StopCube*, which require awareness of object distance and benefit from more holistic views. Across integration mechanisms, **memory-as-modulator performs best** for both TOKENDROP and FRAMESAMP. We attribute this to its lightweight, feature-wise conditioning, which largely preserves the original $\pi_{0.5}$ architecture, whereas other modifications may disrupt pretrained representations of $\pi_{0.5}$. Further analysis of the relative contribution of perceptual memory tokens is provided in Appendix C.5.

We also observe that recurrent methods perform the worst, likely because fine-tuning $\pi_{0.5}$ with shallow recurrent layers leads to unstable training. This suggests that effective recurrence requires deeper architectural integration and stronger recurrence-oriented pretraining, as explored in recent computer vision works (Zhang et al., 2026; Wang et al., 2026a). In contrast, our symbolic variants achieve up to 32.70% success, with GROUNDSG consistently outperforming SIMPLESG on most tasks using `QwenVL`, highlighting the importance of accurate grounding for manipulation. On simpler tasks such as *PickXTimes*, where grounding is less critical, SIMPLESG can outperform GROUNDSG, as grounding errors from `QwenVL` degrade performance. Without fine-tuning, prompt-only `Gemini` suffers from domain shift and often predicts incorrect labels.

SAM2Act+ performs poorly on average but achieves the best result on *ButtonUnmaskSwap*, as it uses a motion planner to execute discrete actions reliably and is therefore less prone to low-level manipulation failures. In contrast, VLA policies sometimes fail to press buttons correctly. MemER, the strongest comparison method, achieves 42.38% success

*Table 3.* **Main Results.** Our MME-VLA suite integrates $\pi_{0.5}$ with three memory representations: (1) symbolic memory, implemented as simple or grounded subgoals (SIMPLESG, GROUNDSG); (2) perceptual memory, using token dropping (TOKENDROP) or frame sampling (FRAMESAMP); and (3) recurrent memory, using test-time training (TTT) or a recurrent memory transformer (RMT). Symbolic subgoals can be generated by Gemini-2.5-Pro (`Gemini`), a fine-tuned Qwen3-VL-4B (`QwenVL`), or simulator ground truth (`Oracle`). Perceptual and recurrent memory are integrated end-to-end via three mechanisms: memory-as-context (`Context`), memory-as-modulator (`Modul`), and memory-as-expert (`Expert`). **Red** marks the best per section; ▢ marks the overall best for non-oracle models.

| Method | | Counting | | | | Permanence | | | | Reference | | | | Imitation | | | | AVG |
|---|---|---|---|---|---|---|---|---|---|---|---|---|---|---|---|---|---|---|
| | | Bin Fill | Pick Xtimes | Swing Xtimes | Stop Cube | Video Umsk | Button Umsk | Video UmskS | Button UmskS | Pick HighL | Video Repick | Video PlcBtn | Video PlcOrd | Move Cube | Insert Peg | Pattern Lock | Route Stick | |
| HUMAN PERFORMANCE | | 96.00 | 100.0 | 80.00 | 78.00 | 90.00 | 92.00 | 92.00 | 90.00 | 92.00 | 92.00 | 98.00 | 90.00 | 90.00 | 98.00 | 84.00 | 86.00 | 90.50 |
| ***MME-VLA w/ Symbolic Memory*** | | | | | | | | | | | | | | | | | | |
| SIMPLESG | `Oracle` | **85.78** | 99.78 | **100.0** | 44.67 | 33.11 | 22.00 | 15.56 | 15.56 | 44.00 | 27.78 | 31.33 | 26.00 | 87.33 | 10.00 | 95.33 | 55.11 | 49.58 |
| GROUNDSG | `Oracle` | **85.78** | **100.0** | **100.0** | **49.67** | **98.78** | **95.00** | **99.22** | **80.22** | **83.33** | **97.33** | **100.0** | **100.0** | **87.78** | **15.56** | **97.00** | **55.56** | **84.08** |
| SIMPLESG | `Gemini` | 46.00 | 63.00 | **45.00** | 2.00 | 29.00 | 9.00 | 14.00 | 2.00 | **20.00** | 15.00 | 26.00 | 29.00 | 61.00 | 4.00 | 7.00 | 0.00 | 23.25 |
| | `QwenVL` | 77.56 | 95.33 | 5.11 | 0.44 | 34.22 | 19.33 | 15.33 | 9.56 | 17.11 | **25.33** | 33.33 | 25.11 | **82.00** | **3.78** | **12.67** | **7.78** | **29.00** |
| GROUNDSG | `Gemini` | 26.00 | 18.00 | 4.00 | **3.00** | 36.00 | 14.00 | 13.00 | 0.00 | 9.00 | 17.00 | 12.00 | 7.00 | 17.00 | 0.00 | 7.00 | 2.00 | 11.56 |
| | `QwenVL` | 52.00 | 92.67 | 7.33 | 0.00 | **88.67** | **24.00** | **30.67** | **14.00** | 15.11 | **25.33** | **54.00** | **31.78** | 71.56 | 3.33 | 6.67 | 6.00 | **32.70** |
| ***MME-VLA w/ Perceptual Memory*** | | | | | | | | | | | | | | | | | | |
| TOKENDROP | `Context` | 48.67 | 85.11 | **94.67** | 3.11 | **33.78** | **31.56** | 26.22 | 16.00 | 20.67 | 17.78 | 31.11 | 25.33 | 81.33 | 4.00 | 12.67 | 20.00 | 34.50 |
| | `Modul` | 34.44 | 83.56 | 86.00 | 5.33 | 28.22 | 29.33 | **28.44** | **21.33** | 21.33 | 22.00 | 59.56 | **36.00** | 62.00 | 7.11 | 32.44 | 51.56 | 38.04 |
| | `Expert` | 54.22 | **87.56** | 91.78 | 4.22 | 26.67 | 30.44 | 18.44 | 18.89 | 19.33 | 20.89 | 36.44 | 24.67 | **87.56** | 2.22 | 16.22 | 18.22 | 34.86 |
| FRAMESAMP | `Context` | 41.22 | 72.00 | 73.67 | 13.67 | 26.89 | 30.22 | 20.89 | 15.22 | 17.67 | 15.22 | 30.00 | 20.89 | 77.22 | 1.22 | 15.22 | 19.67 | 30.68 |
| | `Modul` | 39.56 | 87.33 | 92.00 | **42.00** | 32.67 | 25.11 | 24.44 | 18.22 | **22.89** | **30.44** | **60.00** | 32.00 | 77.78 | **7.56** | **53.56** | **66.67** | **44.51** |
| | `Expert` | **57.33** | 86.22 | **94.67** | 28.89 | 31.78 | 25.78 | 22.89 | 20.22 | 19.11 | 23.11 | 30.00 | 24.22 | 83.11 | 2.00 | 13.56 | 17.11 | 36.25 |
| ***MME-VLA w/ Recurrent Memory*** | | | | | | | | | | | | | | | | | | |
| TTT | `Context` | 35.56 | 62.89 | **42.44** | 3.33 | 29.78 | **22.89** | 18.44 | 14.44 | **20.44** | **13.11** | **34.22** | 20.22 | 32.44 | 1.11 | 1.56 | 3.56 | 22.28 |
| | `Modul` | 34.22 | **65.11** | 36.67 | 2.11 | 27.22 | 22.11 | **25.22** | 14.11 | 14.56 | 12.11 | 32.67 | 22.33 | 31.22 | 1.11 | 3.56 | 7.00 | 21.96 |
| | `Expert` | 34.89 | 63.78 | 41.33 | 4.00 | **31.78** | 22.44 | 19.56 | **18.00** | 12.22 | 9.56 | 34.00 | 22.89 | **33.56** | 0.89 | 3.11 | 5.56 | **22.35** |
| RMT | `Context` | 32.44 | 56.89 | 33.56 | **5.78** | **33.33** | 10.89 | 17.33 | 2.00 | 14.00 | 3.78 | 32.00 | 29.11 | 25.78 | 2.00 | **5.56** | **8.89** | 19.46 |
| | `Modul` | 33.33 | 60.78 | 37.78 | 4.67 | 31.11 | 11.78 | 17.78 | 2.44 | 17.11 | 4.22 | 32.00 | **31.11** | 24.67 | **2.21** | 3.78 | 8.00 | 20.17 |
| | `Expert` | **35.78** | 60.22 | 36.00 | 5.56 | 28.00 | **17.11** | 15.78 | 2.00 | 11.78 | 0.22 | 24.22 | 22.67 | 20.00 | 1.89 | 4.22 | 4.89 | 18.15 |
| ***Other Methods*** | | | | | | | | | | | | | | | | | | |
| $\pi_{0.5}$ | | 30.00 | 42.89 | 35.56 | **6.67** | 20.44 | 22.22 | 18.67 | 6.67 | 11.33 | 0.44 | **31.11** | 25.78 | 26.00 | 1.56 | 2.89 | 4.67 | 17.93 |
| $\pi_{0.5}$ w/ past actions | | 26.67 | 58.33 | 26.67 | 4.67 | 30.67 | 23.67 | 20.67 | 16.00 | 12.33 | 8.67 | 24.00 | 18.67 | 34.00 | 1.00 | 4.00 | 5.67 | 19.73 |
| SAM2Act+ (Fang et al., 2025) | | 40.00 | 76.00 | 25.33 | 0.00 | 27.33 | 32.00 | 18.00 | **26.67** | 17.33 | 5.33 | 24.67 | 20.00 | 29.33 | 0.00 | 0.00 | 0.00 | 21.37 |
| MemER (Sridhar et al., 2026) | | 56.67 | 79.33 | 59.33 | 0.00 | **81.33** | **72.00** | **38.00** | 21.33 | **70.67** | 25.33 | 30.00 | **26.00** | 82.67 | 6.67 | 16.67 | 12.00 | **42.38** |

and excels on *ButtonUnmask* and *PickHighlight*, showing the benefit of combining memory representations, where gains arise from maintaining keyframes to preserve long visual histories while using symbolic subgoals.

**Q2: Is high-level symbolic reasoning alone sufficient for memory-augmented manipulation?** It depends on the specific task. As an upper bound of symbolic memory, GROUNDSG+`Oracle` successfully solves many tasks, confirming that language provides strong representational capacity for high-level reasoning. However, performance still degrades on manipulation-intensive tasks such as *StopCube* and *InsertPeg*, where language offers limited guidance and precise visuomotor control becomes the primary bottleneck. The policy also struggles in cluttered scenes, often manipulating wrong objects and causing unintended collisions in *BinFill* and *PickHighlight* despite unambiguous subgoals.

**Q3: How do humans perform on RoboMME?** We conduct a human study by reformulating each task as an *online VideoQA* problem. Execution videos are revealed incrementally, participants select the next high-level action from a predefined candidate set, and an oracle planner guarantees flawless low-level control (Appendix B.7). Humans achieve 90.5% overall success but still fail to fully solve the benchmark, making consistent errors on long-horizon tasks such as *PatternLock* and time-sensitive tasks like *StopCube*. These results indicate that RoboMME remains inherently challenging and places strong demands on memory, serving as a rigorous testbed for memory-augmented policies.

**Q4: How does the effectiveness of different memory designs depend on task characteristics?** No single memory representation consistently dominates across all tasks. Instead, their strengths are highly task-dependent and complementary. Symbolic memory performs well on several `Counting` tasks, such as *BinFill* and *PickXTimes*. Perceptual memory excels on `Imitation` tasks, including *PatternLock* and *RouteStick*. However, no method fully solves an entire task suite, suggesting that task demands cannot be captured by a simple one-to-one correspondence between cognitive dimensions and memory representations.

To better analyze the effectiveness of memory representations, we group the 16 tasks by their primary functional requirements: *Motion-Centric*, *Time-Sensitive*, *Short-Horizon*

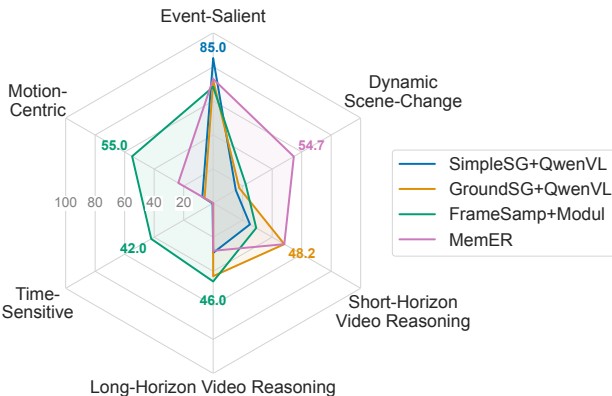

*Figure 3.* Performance comparison across task characteristics.

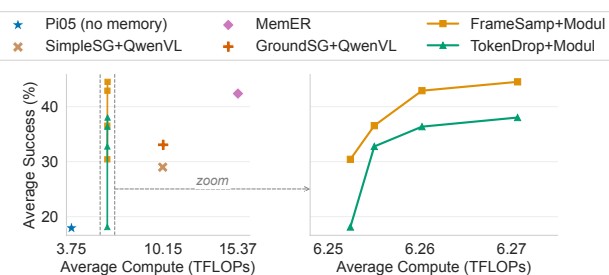

*Figure 4.* Efficiency-performance comparison among models.

*Table 4.* Real-world experiment results.

| Method | Put Fruits | Track Cube | Repick Block | Draw Pattern | Total |
|---|---|---|---|---|---|
| $\pi_{0.5}$ | 2/10 | 1/10 | 1/10 | 0/10 | 4/40 |
| GROUNDSG + QwenVL | **9/10** | 3/10 | 5/10 | 2/10 | 19/40 |
| FRAMESAMP + Modul | 6/10 | **5/10** | **6/10** | **8/10** | **25/40** |

contrast, methods that rely on external VLM inference introduce substantially higher overhead: GROUNDSG+QwenVL requires roughly $3\times$ the computation of $\pi_{0.5}$, while MemER nearly $5\times$. In practice, these costs can be reduced through *caching*, e.g., by reusing subgoals with less frequent VLM inference or reusing visual tokens to avoid redundancy.

**Q6: Do the trends observed on RoboMME transfer to real-world robotic manipulation?** Yes. We evaluate four real-world tasks, *PutFruits*, *TrackCube*, *RepickBlock*, and *DrawPattern*, designed to mirror the *BinFill*, *VideoUnmask(Swap)*, *VideoRepick* and *PatternLock* tasks in simulation. We collect a total of 350 demonstrations and test our strongest symbolic (GROUNDSG+QwenVL) and perceptual (FRAMESAMP+Modul) models to assess whether the trends can transfer to physical settings. As shown in Table 4, the results exhibit similar patterns. Symbolic memory performs best on the counting task *PutFruits*, whereas perceptual memory excels on motion-centric tasks such as *DrawPattern*. On the remaining two tasks, both approaches achieve comparable performance, suggesting room for further improvement. Appendix D provides more details.

## 6. Conclusion and Future Work

This work introduces RoboMME, a unified benchmark for systematically evaluating memory-augmented robotic manipulation across four cognitive dimensions: temporal, spatial, object, and procedural memory. We further develop a family of vision-language-action (VLA) models and conduct controlled comparisons of symbolic, perceptual, and recurrent memory representations with multiple integration mechanisms. Results show that no single design consistently dominates: symbolic memory excels at counting and short-horizon reasoning, while perceptual memory is crucial for motion-centric and time-sensitive behaviors.

Despite its breadth, RoboMME focuses on tabletop manipulation with a fixed set of assets and is mainly evaluated on the $\pi_{0.5}$ backbone, leaving mobile manipulation and alternative architectures, such as memory-bank methods and additional VLA backbones, for future study. Our results further suggest that memory representations are complementary rather than exclusive, motivating unified frameworks that combine multiple forms of memory. We hope RoboMME serves as a foundational and lasting testbed for developing reliable, memory-augmented robotic generalist policies.

*Video Reasoning*, *Long-Horizon Video Reasoning*, *Dynamic Scene-Change* and *Event-Salient*. More detailed descriptions are provided in Appendix C.3. As shown in Figure 3, perceptual memory (FRAMESAMP + Modul) is most effective for motion-centric, time-sensitive, and long-horizon tasks, where access to extended visual history and low-level motions is critical. In contrast, symbolic memory excels on short-horizon and event-salient tasks by providing clear subgoal guidance. MemER performs best on dynamic scene-change tasks, likely because it retains keyframe images to keep necessary details during online execution. Overall, these results indicate that different memory designs provide complementary strengths, suggesting that combining them synergistically could further improve performance.

**Q5: How does memory affect the efficiency-performance balance of memory-augmented policies?** Memory improves history dependence but inevitably introduces additional computational cost. To quantify this trade-off, we vary the memory budget (64-512 tokens) and compare efficiency-performance scaling across models. Computational cost is estimated from the TFLOPs of a single forward pass based on the average episode length. As shown in Figure 4, perceptual memory achieves the most favorable balance. FRAMESAMP+Modul delivers consistent performance gains as the memory budget increases while incurring only modest additional cost, since most computation arises from processing visual tokens rather than from memory integration itself. In

## Acknowledgments

This work was supported in part by NSF IIS-1949634, NSF SES-2128623, NSF NAIRR250085, NSF CAREER #2337870, and NSF NRI #2220876. This research is also supported by the National Artificial Intelligence Research Resource (NAIRR) Pilot and the Delta advanced computing and data resource which is supported by the National Science Foundation (award NSF-OAC 2005572).

## Impact Statement

This work advances benchmarking and evaluation for memory-augmented robotic manipulation, contributing to the development of more reliable and capable robotic systems in assistive and industrial settings. As with other autonomous technologies, such systems should be developed and deployed responsibly with appropriate safety safeguards and human oversight. We do not anticipate societal risks beyond those generally associated with robotics and machine learning research.

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

## Appendix Outline

# A. Model Architectures in the MME-VLA Suite

In the MME-VLA suite, we develop a family of vision-language-action (VLA) policies based on a shared $\pi_{0.5}$ backbone. We first introduce the notation for the base architecture, then describe how different memory *representations* instantiate memory, and finally explain how different *integration mechanisms* incorporate memory into the $\pi_{0.5}$ policy.

## A.1. Basic $\pi_{0.5}$ Architecture

**Tokenization.** At timestep $t$, the policy receives (i) **language tokens** encoding the task instruction $l$, and (ii) **image tokens** encoding the current multi-view RGB observations $I_t = (I_t^1, I_t^2, \dots)$:

$$\boldsymbol{\ell} = \mathrm{Tok}_{\text{text}}(l) \in \mathbb{R}^{N_\ell \times d}, \tag{1}$$

$$\mathbf{o}_t = \mathrm{Tok}_{\text{img}}(I_t) \in \mathbb{R}^{N_o \times d}. \tag{2}$$

Here $d$ is the token embedding dimension, $N_l$ and $N_o$ denote the total number of language and visual tokens, respectively.

**Experts.** $\pi_{0.5}$ (Black et al., 2025) follows the $\pi_0$ design (Black et al., 2024) and adopts a multi-expert transformer architecture composed of:

- a **VLM expert** $E_{\text{vlm}}$ that fuses language and vision, and

- an **action expert** $E_{\text{act}}$ that predicts actions conditioned on the VLM features and the denoising timestep.

Both experts consist of $L$ standard transformer layers with blockwise causal attention and MLPs.

**Blockwise causal attention.** We define a general blockwise attention operator

$$\mathbf{y} = \mathrm{BlockAttn}(\mathbf{x}_1, \mathbf{x}_2, \dots, \mathbf{x}_G), \tag{3}$$

where each $\mathbf{x}_i \in \mathbb{R}^{n_i \times d}$ is treated as a block. Attention is bidirectional within each block and causal across blocks, i.e., tokens in block $i$ may attend to blocks $1{:}i$ but not to future blocks. The operator returns the updated representation of the final block, $\mathbf{y} \in \mathbb{R}^{n_G \times d}$.

**VLM expert.** A transformer encoder is used to extract joint visual-language features. The input consists of visual tokens $\mathbf{o}_t$ and language tokens $\boldsymbol{\ell}$:

$$\mathbf{u}_t^0 = [\mathbf{o}_t; \boldsymbol{\ell}]. \tag{4}$$

where concatenation along the sequence dimension is denoted by $[\cdot;\cdot]$.

Each layer applies blockwise causal attention followed by an MLP:

$$\tilde{\mathbf{u}}_t^k = \mathbf{u}_t^{k-1} + \mathrm{BlockAttn}^k\big(\mathrm{Norm}_{\text{vlm}}^k(\mathbf{u}_t^{k-1})\big), \tag{5}$$

$$\mathbf{u}_t^k = \tilde{\mathbf{u}}_t^k + \mathrm{MLP}_{\text{vlm}}^k\big(\mathrm{Norm}_{\text{vlm}}^k(\tilde{\mathbf{u}}_t^k)\big), \tag{6}$$

for layer $k = 1, \dots, L$, producing $\mathbf{u}_t = \mathbf{u}_t^L$. Here, there is only a single input block. Norm is implemented as RMSNorm. Unless otherwise specified, the normalization layers used in the attention and MLP modules are separately parameterized; we use the same notation here for simplicity.

Unlike the original $\pi_{0.5}$, which also generates low-level commands directly from the VLM expert for task decomposition, we use the VLM solely as a feature extractor, similar to $\pi_0$.

**Action expert.** The action expert is a lightweight transformer that predicts actions by attending to the VLM features. Let $\mathbf{s}_t^0 \in \mathbb{R}^{N_s \times d}$ denote the action input injected with noise. Then for each layer:

$$\tilde{\mathbf{s}}_t^k = \mathbf{s}_t^{k-1} + g_\tau \odot \mathrm{BlockAttn}^k\big(\mathrm{Norm}_{\text{vlm}}^k(\mathbf{u}_t^{k-1}), \mathrm{Norm}_{\text{act}}^k(\mathbf{s}_t^{k-1}, \tau)\big), \tag{7}$$

$$\mathbf{s}_t^k = \tilde{\mathbf{s}}_t^k + \tilde{g}_\tau \odot \mathrm{MLP}_{\text{act}}^k\big(\mathrm{Norm}_{\text{act}}^k(\tilde{\mathbf{s}}_t^k, \tau)\big). \tag{8}$$

Here, $\text{Norm}(\cdot, \tau)$ is implemented as RMSNorm conditioned on the embedding of the denoising timestep $\tau$. $g^\tau$ and $\tilde{g}^\tau$ are gating functions conditioned on $\tau$, and $\odot$ denotes feature-wise multiplication. This mechanism corresponds to AdaLN-Zero (Peebles & Xie, 2023). The policy then predicts the next action $a_t \in \mathbb{R}^D$ (or an action chunk) via an output projection:

$$a_t \sim \pi(a_t \mid I_t, l) = \text{Proj}(\mathbf{s}_t^L). \tag{9}$$

**Flow-matching objective.** At history timestep $t$, the policy predicts an $H$-step action chunk representing future actions:

$$\mathbf{A}_t = [a_t, a_{t+1}, \ldots, a_{t+H-1}] \in \mathbb{R}^{H \times D}. \tag{10}$$

Let $\mathbf{h}_t$ denote the context features. Following conditional flow matching, we learn a velocity field that transports Gaussian noise to the data action distribution.

We sample a ground-truth chunk $\mathbf{A}_t^{\text{data}} \sim p_{\text{data}}$ and noise $\mathbf{A}_t^{\text{noise}} \sim \mathcal{N}(\mathbf{0}, \mathbf{I})$, and draw a denoising timestep $\tau \sim \mathcal{U}(0, 1)$. We construct the linear interpolation

$$\mathbf{A}_t^\tau = (1 - \tau)\mathbf{A}_t^{\text{data}} + \tau \mathbf{A}_t^{\text{noise}}. \tag{11}$$

The time derivative of this path is constant, $\frac{d}{d\tau}\mathbf{A}_t^\tau = \mathbf{A}_t^{\text{noise}} - \mathbf{A}_t^{\text{data}}$. The action expert parameterizes a conditional velocity field

$$\mathbf{v}_\theta(\mathbf{A}_t^\tau, \tau, \mathbf{h}_t), \tag{12}$$

which is trained to match this transport direction via

$$\mathcal{L}_{\text{FM}} = \mathbb{E}\left[\left\|\mathbf{v}_\theta(\mathbf{A}_t^\tau, \tau, \mathbf{h}_t) - \left(\mathbf{A}_t^{\text{noise}} - \mathbf{A}_t^{\text{data}}\right)\right\|_2^2\right]. \tag{13}$$

For more comprehensive understanding about $\pi_0$ and $\pi_{0.5}$ models, please refer to the original papers (Black et al., 2025; 2024).

### A.2. Memory Representations

To enable history-dependent reasoning, we augment the policy with an explicit memory state $\mathbf{M}_t$:

$$a_t \sim \pi(\cdot \mid l, \mathbf{o}_t, \mathbf{M}_t),$$

where $\mathbf{M}_t$ summarizes past observations and actions.

For fair comparison across designs, we allocate a fixed memory token budget $B$ for all memory-based models. Each representation produces at most $B$ tokens; if fewer are available, the sequence is zero-padded. Thus all memory variants share the same interface:

$$\mathbf{M}_t \in \mathbb{R}^{B \times d}.$$

We consider three classes of memory representations: *symbolic*, *perceptual*, and *recurrent* memory.

#### A.2.1. SYMBOLIC MEMORY

Symbolic memory summarizes history using discrete abstractions (e.g., language subgoals). Let

$$\mathcal{H}_t = \{(I_\tau, a_\tau)\}_{\tau < t}$$

denote the trajectory history. A subgoal generator $g(\cdot)$ (implemented by a VLM) produces a text summary, which is tokenized into memory tokens:

$$\mathbf{M}_t^{\text{sym}} = \text{Tok}_{\text{text}}(g(\mathcal{H}_t)) \in \mathbb{R}^{B \times d}.$$

This representation compresses the trajectory into compact semantic descriptions, enabling high-level reasoning and structured planning.[2]

---

[2]Although the original $\pi_{0.5}$ model is capable of generating language subgoals, we treat the VLA primarily as a language-conditioned policy in this study. To control for the source and quality of subgoal generation, we instead employ external models to produce language

### A.2.2. PERCEPTUAL MEMORY

Perceptual memory retains selected visual tokens[3] from past observations:

$$\mathbf{M}_t^{\text{perc}} = \text{Select}\big(\{\mathbf{o}_\tau\}_{\tau \leq t}, B\big).$$

Where $\text{Select}(\cdot)$ is an image token selection operator. Unlike symbolic memory, this representation preserves differentiable visual features and enables end-to-end optimization.

We study two instantiations.

**Frame Sampling (FRAMESAMP).** We select a subset of history timesteps $\mathcal{S}_t \subseteq \{0, \ldots, t\}$ with $|\mathcal{S}_t| = N$ (e.g., evenly spaced) and retain pooled tokens from each frame:

$$\mathcal{S}_t = \text{EvenSample}(\{0, \ldots, t\}; N), \tag{14}$$

$$\mathbf{M}_t^{\text{perc}} = \big[\text{MaxPool}(\mathbf{o}_\tau)\big]_{\tau \in \mathcal{S}_t}. \tag{15}$$

Here $\text{MaxPool}$ reduces each frame from original $16 \times 16$ tokens (extracted by SigLIP (Tschannen et al., 2025), the vision encoder of $\pi_{0.5}$) to $P$ tokens per view (e.g, $4 \times 4$, $8 \times 8$). With $V$ views, $N = B/PV$.

**Token Dropping (TOKENDROP).** Rather than sampling whole frames, we select the most informative spatial regions (i.e., image patches) across time. To reduce computation and noise, we also first spatially pool each frame's tokens from original $16 \times 16$ to a $P$ tokens:

$$\tilde{\mathbf{o}}_\tau = \text{MaxPool}(\mathbf{o}_\tau),$$

We then score each pooled token $\tilde{\mathbf{o}}_{\tau,i}$ with an importance function comparing with previous $K$-th frame tokens at the same patch.

$$s_{\tau,i} = J(\tilde{\mathbf{o}}_{\tau,i}, \tilde{\mathbf{o}}_{\tau-K,i}), \tag{16}$$

where $i$ indexes pooled spatial locations, $K$ is a fixed stride. We keep the top $B$ tokens across all timesteps:

$$\mathcal{K}_t = \text{TopK}\Big(\{(\tau, i)\}, B; s_{\tau,i}\Big), \tag{17}$$

$$\mathbf{M}_t^{\text{perc}} = \big[\tilde{\mathbf{o}}_{\tau,i}\big]_{(\tau,i) \in \mathcal{K}_t}. \tag{18}$$

In practice, $J(\cdot, \cdot)$ measures temporal saliency via average RGB patch differences across frames, prioritizing regions with significant motion or appearance changes while keeping the memory budget fixed. To preserve spatiotemporal structure, we adopt Multi-modal Rotary Position Embedding (M-ROPE) similar to (Yao et al., 2025), assigning each video token a 3D index over time, height, and width. Tokens from the first frame are fully preserved to provide complete context.

### A.2.3. RECURRENT MEMORY

Recurrent memory compresses history into a fixed-size latent state updated online:

$$\mathbf{S}_t = f_{\text{mem}}(\mathbf{S}_{t-1}, \mathbf{o}_t).$$

The resulting state is exposed as memory tokens $\mathbf{M}_t^{\text{recu}}$.

We consider two instantiations.

**Recurrent Memory Transformer (RMT).** RMT maintains $B$ persistent memory slots $\mathbf{S}_{\text{init}} \in \mathbb{R}^{B \times d}$. At each timestep, observation tokens are processed sequentially in $M$ chunks:

$$\mathbf{o}_t = [\mathbf{o}_t^{(1)}, \ldots, \mathbf{o}_t^{(M)}], \qquad \mathbf{o}_t^{(m)} \in \mathbb{R}^{N_C \times d}.$$

---

subgoals.

[3]We also experimented with incorporating historical proprioceptive states, but observed no consistent performance improvement. Therefore, we restrict perceptual memory to visual tokens only.

Memory slots attend to each chunk and are updated recurrently:

$$\mathbf{S}_t^{(m)} = \mathrm{Transformer}\big(Q = \mathbf{S}_t^{(m-1)}, , K = [\mathbf{o}_t^{(m)}; \mathbf{S}_t^{(m-1)}], , V = [\mathbf{o}_t^{(m)}; \mathbf{S}_t^{(m-1)}]\big), \tag{19}$$

initialized with $\mathbf{S}_t^{(0)} = \mathbf{S}_{\mathrm{init}}$. Here, $\mathrm{Transformer}(Q, K, V)$ denotes a standard transformer layer that uses $Q$ as queries and $[K, V]$ as the key-value sequence, returning the updated representations corresponding to the query tokens. After all chunks,

$$\mathbf{M}_t^{\mathrm{recu}} = \mathbf{S}_t^{(M)}.$$

Thus, RMT stores history explicitly in token space through persistent memory slots. In our implementation, we only use the attention block without the MLP block for simplicity.

**Test-Time Training (TTT).** TTT stores memory implicitly in a small set of fast weights rather than explicit tokens. Let $W_t$ denote fast weights associated with a lightweight feed-forward network. At each timestep, the weights are updated online using a local gradient step:

$$W_t = W_{t-1} - \eta \nabla_W \ell_{\mathrm{aux}}(W_{t-1}; \mathbf{o}_t), \tag{20}$$

where $\ell_{\mathrm{aux}}$ is a self-supervised objective. The updated weights are immediately used for prediction:

$$y_t = f(\mathbf{o}_t; W_t).$$

Thus, the output $y_t$ implicitly contains the history information. We use the last $B$ tokens from $y_{-B:}$ as $\mathbf{M}_t^{\mathrm{recu}}$. TTT therefore stores history in parameter space rather than explicit memory tokens.

## A.3. Memory Integration Mechanisms

We study how memory interacts with the backbone.

### A.3.1. MEMORY-AS-CONTEXT

Append memory tokens directly:

$$\mathbf{u}_t = E_{\mathrm{vlm}}([\mathbf{M}_t; \mathbf{o}_t; \boldsymbol{\ell}]).$$

### A.3.2. MEMORY-AS-MODULATOR

Inject memory through *feature-wise modulation* inside the action expert. For each MLP sublayer of the action expert, Eq. 8 becomes

$$\mathbf{r}_t^k = \mathrm{Attn}_{\mathrm{mod}}^k(Q = \tilde{\mathbf{s}}_t^k, K = \mathbf{M}_t, V = \mathbf{M}_t), \tag{21}$$

$$(\gamma_t^k, \beta_t^k) = \mathrm{MLP}_{\mathrm{mod}}^k(\mathbf{r}_t^k), \tag{22}$$

$$\widehat{\mathbf{s}}_t^k = \gamma_t^k \odot \mathrm{Norm}_{\mathrm{mod}}^k(\tilde{\mathbf{s}}_t^k) + \beta_t^k, \tag{23}$$

$$\mathbf{s}_t^k = \tilde{\mathbf{s}}_t^k + \tilde{g}_\tau \odot \mathrm{MLP}_{\mathrm{act}}^k\Big(\mathrm{Norm}_{\mathrm{act}}^k(\widehat{\mathbf{s}}_t^k, \tau)\Big). \tag{24}$$

Here, $\mathbf{r}_t^k$ denotes the layer-wise modulation feature. The parameters of $\mathrm{MLP}_{\mathrm{mod}}^k$ are initialized such that $\gamma_t^k = 1$ and $\beta_t^k = 0$, yielding an identity modulation at initialization of fine-tuning.

### A.3.3. MEMORY-AS-EXPERT

We allocate a dedicated memory expert $E_{\mathrm{mem}}$ to process memory tokens:

$$\tilde{\mathbf{v}}_t^k = \mathbf{v}_t^{k-1} + \mathrm{BlockAttn}^k\big(\mathrm{Norm}_{\mathrm{mem}}^k(\mathbf{v}_t^{k-1})\big), \tag{25}$$

$$\mathbf{v}_t^k = \tilde{\mathbf{v}}_t^k + \mathrm{MLP}_{\mathrm{mem}}^k\big(\mathrm{Norm}_{\mathrm{mem}}^k(\tilde{\mathbf{v}}_t^k)\big), \tag{26}$$

where $\mathbf{v}_t^0 = \mathbf{M}_t$ denotes the input memory tokens.

The resulting memory features are then used as additional context in the action attention. Accordingly, Eq. 7 becomes

$$\tilde{\mathbf{s}}_t^k = \mathbf{s}_t^{k-1} + g_\tau \odot \text{BlockAttn}^k\big(\text{Norm}_{\text{mem}}^k(\mathbf{v}_t^{k-1}), \text{Norm}_{\text{vlm}}^k(\mathbf{u}_t^{k-1}), \text{Norm}_{\text{act}}^k(\mathbf{s}_t^{k-1}, \tau)\big).$$

This design preserves memory as explicit tokens while enabling higher-capacity processing via a separate expert. *To avoid interfering with the VLM representations, the VLM and memory experts use self-attention only, and only the action expert attends to both as context.*

# B. Experiment Implementation Details on RoboMME

## B.1. VLM Subgoal Prediction and Prompts

Language subgoals help decompose complicated tasks into a sequence of subtasks, which has been shown to be helpful in many robotic tasks (Dai et al., 2025a; Lin et al., 2026; Dai et al., 2024). We train all VLM-based subgoal predictors using **Qwen3-VL-4B-Instruct**, which provides strong visual reasoning and grounding capabilities while remaining efficient to fine-tune. In preliminary experiments, it achieves better performance than Qwen2.5-VL-7B with significantly less computational cost.

We fine-tune all models using `Swift`[4] with LoRA adaptation. The training configuration is shown as below, the remaining hyperparameters use the default settings provided by `Swift`.

*Table 5.* Training configuration for VLM subgoal predictor.

| Hyperparameter | Value |
| --- | --- |
| Batch Size | 48 |
| Warmup Ratio | 0.05 |
| LR | $1 \times 10^{-4}$ |
| Train Epochs | 2 |
| LoRA | rank 16, alpha 32 |
| DeepSpeed | ZeRO-2 |

Figures 5 and 6 illustrate the training prompts for simple and grounded subgoal prediction, respectively. Since grounded subgoals consistently yield better performance in our experiments, we also modify the MemER (Sridhar et al., 2026) method to predict grounded subgoals using their prompts, as shown in Figure 7. For each VLM variant, we use one prompt for multi-task training.

---

[4]`https://github.com/modelscope/ms-swift`

**Example Prompt for Simple Subgoal Prediction**

```
{
    "messages":[
        {
            "role":"system",
            "content":"You are a helpful assistant to help guide the robot to complete
    the task by predicting a sequence of language subgoals"
        },
        {
            "role":"user",
            "content":"<video>The task goal is: watch the video carefully, then
    repeatedly pick up and put down the same block that was previously picked up for
    two times, finally press the button to stop\nThe history of previous predicted
    language subgoals are: 1. pick up the correct cube for the first time; 2. put it
    down; 3. pick up the correct cube for the second time; 4. put it down; 5. press
    the button to finish\n<image>What's the next language subgoal based on current
    observation?"
        },
        {
            "role":"assistant",
            "content":"press the button to finish"
        }
    ],
    "images":[
        "<path_to_current_image.png>",
    ],
    "videos":[
        "<path_to_task_input_video.mp4>"
    ]
}
```

*Figure 5.* Example Prompt for VLM Simple Subgoal Prediction. This is used to generate simple subgoals for the SIMPLESG VLA policy.

**Example Prompt for Grounded Subgoal Prediction**

```
{
    "messages":[
        {
            "role":"system",
            "content":"You are a helpful assistant to help guide the robot to complete
        the task by predicting a sequence of grounded language subgoals"
        },
        {
            "role":"user",
            "content":"<video>The task goal is: watch the video carefully, then
        repeatedly pick up and put down the same block that was previously picked up for
        two times, finally press the button to stop\nThe history of previous predicted
        grounded language subgoals are: 1. pick up the correct cube at <bbox> for the
        first time; 2. put it down; 3. pick up the correct cube at <bbox> for the second
        time; 4. put it down; 5. press the button at <bbox> to finish\n<image>What's the
        next grounded language subgoal based on current observation?"
        },
        {
            "role":"assistant",
            "content":"press the button at <bbox> to finish"
        }
    ],
     "objects":{
        "ref":[],
        "bbox":[[390, 706], [394, 706], [261, 518], [261, 518]]
    },
    "images":[
        "<path_to_current_image.png>",
    ],
    "videos":[
        "<path_to_task_input_video.mp4>"
    ]
}
```

*Figure 6.* Example Prompt for VLM Grounded Subgoal Prediction. This is used to generate grounded subgoals for the GROUNDSG VLA policy.

**Example Prompt for MemER**

```
{
    "messages":[
        {
            "role":"system",
            "content":"You are a robot program that predicts actions. The current input
    images from the front-view camera shows the most recent actions the robot has
    executed. The past keyframes are selected frames of particular importance from
    all the actions the robot has executed so far. Based on these, output the current
     subtask the robot should execute and nothing else. Some tasks may have a video
    input for initial setup, some may not.\n\nReturn a JSON with:\n- current_subtask:
     the action that should be executed at the current timestep\n- keyframe_positions:
     list of frame positions (1-indexed) from the current input images where actions
    change"
        },
        {
            "role":"user",
            "content":"The task has a video input for initial setup: <video>\nThe task
    goal is: watch the video carefully, then repeatedly pick up and put down the same
     block that was previously picked up three times, finally press the button to
    stop\nHere are the selected frames from the entirety of the full execution that
    are of particular importance:[<image>, <image>, <image>, <image>]\nHere is
    current input image list from the front-view camera: [<image>, <image>, <image>,
    <image>, <image>, <image>, <image>, <image>]\n\nWhat subtask should the robot
    execute and what is the keyframe position?"
        },
        {
            "role":"assistant",
            "content":"{\"current_subtask\": \"pick up the correct cube at <bbox> for
    the third time\", \"keyframe_positions\": [4]}"
        }
    ],
    "objects":{
        "ref":[],
        "bbox":[[207, 417]]
    },
    "images":[
        "<path_to_past_keyframe_1.png>",
        "<path_to_past_keyframe_2.png>",
        "<path_to_past_keyframe_3.png>",
        "<path_to_past_keyframe_4.png>",
        "<path_to_current_input_1.png>",
        "<path_to_current_input_2.png>",
        "<path_to_current_input_3.png>",
        "<path_to_current_input_4.png>",
        "<path_to_current_input_5.png>",
        "<path_to_current_input_6.png>",
        "<path_to_current_input_7.png>",
        "<path_to_current_input_8.png>"
    ],
    "videos":[
        "<path_to_task_input_video.mp4>"
    ]
}
```

*Figure 7.* Example Prompt for MemER Baseline. We use grounded subgoals for MemER because we found that grounded subgoals perform better.

## B.2. Training Details for MME-VLA Suite

### B.2.1. PARAMETERS FOR DIFFERENT MEMORY REPRESENTATIONS.

All VLA models share a fixed memory budget of $B = 512$ tokens. Unless otherwise specified, we use only the front-view image (i.e., $V = 1$) as the source for either memory tokens or subgoal prediction.

For **symbolic memory**, we simply increase the maximum number of language tokens to 512 and extend the original $\pi_{0.5}$ prompt as shown below, while keeping the rest of the architecture unchanged:

```
Task: {task_goal}\nCurrent Subgoal: {language_subgoal}\nAction:
```

For **neural memory (perceptual and recurrent)**, we first encode all observations using the $\pi_{0.5}$ vision encoder SigLIP2. The tokens are then passed through a lightweight MLP feature encoder that projects the feature dimension from 2048 (SigLIP2 output dimension) to 1024 to match the internal $\pi_{0.5}$ width. The projected features serve as inputs to the memory modules described below.

For **perceptual memory**, we reduce memory and computation by spatially downsampling each frame via max pooling. FRAMESAMP pools each frame to $P = 4 \times 4 = 16$ tokens and uniformly samples $N = 32$ frames. TOKENDROP instead uses a finer $8 \times 8$ grid and selects tokens with a temporal stride of $K = 8$, based on an average RGB-difference threshold of $10^{-4}$ within each patch. All selected tokens are concatenated with M-RoPE positional embeddings (768 dimension) and linearly projected into position-aware visual features to match the input feature width of different integration mechanisms: 2048 for memory-as-context and 1024 for the other two mechanisms.

For **recurrent memory**, we use only cached front-view visual tokens. At each sampled timestep, we construct a history of up to 64 frames, with the last frame always corresponding to the current timestep. For non-video-conditioned tasks, frames are sampled from the execution history every $K = 16$ steps with a random offset for data augmentation; for video-conditioned tasks, we first sample the demonstration video using the same stride rule or uniformly up to 40 frames, then concatenate the sampled execution history. If fewer than 64 frames are available, we left-pad the sequence and apply a validity mask. Each frame is pooled to an $8 \times 8$ grid, yielding 64 visual tokens per frame. These tokens are augmented with M-RoPE positional embeddings and fed sequentially into the recurrent module. RMT maintains 512 learnable memory queries and updates them through grouped-query cross-attention over each frame's tokens, using eight heads with one key-value head. TTT maintains $512 \times 512$ fast weights for a linear MLP, with two TTT heads, learning rate $\eta = 0.01$, query-key normalization, and gradient clipping at 5.0 for stability. After recurrent compression, both RMT and TTT output 512 memory tokens with a validity mask, which are then integrated into the $\pi_{0.5}$ backbone through the three integration mechanisms. During training, each sample is a randomly selected timestep from a randomly selected episode; gradients are backpropagated through the 64 recurrent steps, but supervision is applied only at the final step for RMT or the last eight steps for TTT. During inference, we maintain a cumulative history buffer, append one frame every $K = 16$ execution steps, and keep the most recent 64 frames. Since TTT, RMT, and the feature encoder are lightweight, the total parameter overhead is under 10M.

### B.2.2. PARAMETERS FOR DIFFERENT MEMORY INTEGRATION MECHANISMS.

The base $\pi_{0.5}$ model processes 576 observation tokens (512 visual tokens from two images and 64 language tokens) and contains 3.2B parameters.

For **symbolic memory**, increasing the language budget to $B = 512$ results in 1024 total input tokens.

For **memory-as-context**, we concatenate $B = 512$ memory tokens with the observation tokens, yielding 1088 tokens in total. This approach introduces no additional parameters.

For **memory-as-modulator**, we inject memory-conditioned modulation into every layer of the action expert using a lightweight attention module with one key-value head, head dimension 256, and width 1024, adding approximately 80M parameters.

For **memory-as-expert**, we introduce a dedicated 18-layer transformer expert (width 1024, MLP dimension 2048, eight attention heads with one key-value head, head dimension 256), which increases the model size by approximately 190M parameters.

### B.2.3. TRAINING CONFIGURATION

We largely follow a unified training recipe across all MME-VLA variants to ensure fair comparison. Specifically, we adopt the released $\pi_{0.5}$ configuration from the LIBERO benchmark and *exclude* proprioceptive states from the inputs since we found that they do not bring benefits. To accelerate training, we freeze the SigLIP2 vision backbone and precompute and cache all visual tokens as in (Torne et al., 2025), fine-tuning only the VLM expert, the action expert, and any parameters introduced by memory.

All models are trained for 80K steps. Symbolic and perceptual memory variants are trained with batch size 64, while recurrent memory variants are trained with batch size 16. Most models are trained on 4×A40 GPUs, with cached data stored on the Turbo storage system. Under this setting, symbolic-memory models take approximately 3 days, perceptual-memory models take approximately 3-4 days, and TTT-based recurrent models take approximately 5-6 days. Due to higher GPU memory requirements, RMT-based recurrent models are trained on 2×H100 GPUs, taking approximately 2-3 days. Detailed hyperparameters are provided in Table 6.

Our code for MME-VLA is available at `https://github.com/RoboMME/robomme_policy_learning`.

*Table 6.* Training hyperparameters for MME-VLA models.

| Hyperparameter | Value |
|---|---|
| Base model | $\pi_{0.5}$ base |
| Batch size | 64 for symbolic/perceptual, 16 for recurrent |
| Action horizon | 20 |
| Action space | Joint-space |
| Proprioception | False |
| Total steps | 80,000 |
| Warmup steps | 10,000 |
| LR | $5 \times 10^{-5}$ (constant) |
| LR schedule | Constant |
| Optimizer | AdamW ($\beta_1$=0.9, $\beta_2$=0.95, weight decay 0) |
| Gradient clip norm | 1.0 |
| EMA decay | 0.999 |
| Frozen part | SigLIP ViT |
| Parallelism | FSDP 4xA40 (2xH100 for RMT runs) |

### B.3. Training Details for SAM2Act+

We implement the SAM2Act+ architecture following the two-stage training pipeline described in the original paper (Fang et al., 2025). Specifically, we first pretrain the SAM2 backbone without temporal connections, and then fine-tune the memory attention module. Each stage is trained for 40k steps on our dataset. Training is conducted on four NVIDIA A40 GPUs for approximately 2.5 days per stage. The detailed training configurations and hyperparameters are provided in Table 7.

**Discrete Actions.** A core design choice in SAM2Act+ is to operate in a **discrete waypoint-based** action space rather than predicting dense continuous actions. Our dataset also provides waypoint-based actions at keyframes, which are compatible with this formulation. However, due to the limitations of discrete action parameterization, performance degrades on tasks requiring precise continuous control, such as *StopCube* and *InsertPeg*. For the remaining tasks, the discrete actions are sufficient. Once the model predicts a waypoint, the ManiSkill simulator will invoke an internal motion planner to execute the action.

**Video-based Observation Input.** The original SAM2Act+ architecture does not include a dedicated mechanism for ingesting demonstration videos as conditional input. To support video-conditioned tasks, we treat the demonstration video as part of the action prediction process during training. At test time, we first roll out the demonstration video through SAM2Act+ to prefill its memory bank without interacting with the simulator. After memory initialization, the agent begins execution using the current observation, following the same protocol adopted in (Fang et al., 2025).

Our code for adapting SAM2Act+ can be found at `https://github.com/RoboMME/SAM2Act`

*Table 7.* Training hyperparameters for SAM2Act and SAM2Act+.

| Hyperparameters | SAM2Act Training | SAM2Act+ Training |
|---|---|---|
| Batch size | 48 | 80 |
| LR | $6.0 \times 10^{-4}$ | $1.0 \times 10^{-4}$ |
| Optimizer | LAMB | LAMB |
| LR schedule | warmup + cosine | warmup + cosine |
| Weight decay | $1 \times 10^{-4}$ | $1 \times 10^{-4}$ |
| Warmup steps | 2000 | 2000 |
| Total steps | 40K | 40K |
| LoRA rank | 16 | 16 |
| Parallelism | DDP 4xA40 | DDP 4xA40 |

## B.4. Training Details for OpenVLA-OFT

OpenVLA-OFT (Kim et al., 2025a) provides an effective recipe for fine-tuning OpenVLA (Kim et al., 2025b). We adapt our memory representations to OpenVLA. Unlike $\pi_{0.5}$, OpenVLA is a single dense transformer in which vision, language, proprioception, and action tokens share the same LLaMA backbone. Therefore, memory integration mechanisms that rely on a dedicated action-expert pathway are less suitable.

Memory-as-expert is not directly applicable because OpenVLA does not support multiple experts or cross-attention among them. Memory-as-modulation is also less stable, since modulating action-token representations can perturb the shared attention stream and interfere with the frozen backbone. In our pilot study, we found that modulation can introduce loss spikes. In contrast, memory-as-context is more stable and architecture-compatible: symbolic memory can be appended as language tokens, while perceptual memory can be projected into the token embedding space and appended as additional context tokens.

Therefore, we evaluate symbolic memory and perceptual memory-as-context on OpenVLA-OFT. For symbolic memory, we use the same `QwenVL` subgoal predictors as in MME-VLA. For perceptual memory, we use fused DINOv2 and SigLIP features and evaluate both FRAMESAMP and TOKENDROP. All models start from OpenVLA-7B. We fine-tune the LLaMA backbone with LoRA, while fully training the action head and memory projection layers.

Since we use an L1 regression loss to train the action head, evaluation results do not vary across model seeds, unlike other compared diffusion-based or flow-matching-based models.

Our code for adapting OpenVLA-OFT is available at `https://github.com/RoboMME/OpenVLA-OFT`.

*Table 8.* Training hyperparameters for OpenVLA-OFT.

| Hyperparameter | Value |
| --- | --- |
| Base model | OpenVLA-7B (LoRA, rank 32) |
| Batch size | 32 (symbolic), 24 (perceptual) |
| Total steps | 110,000 |
| Action head | L1 regression, parallel decoding |
| Action horizon | 20 |
| Action space | Joint-space |
| Proprioception | True |
| Optimizer | AdamW ($\beta_1$=0.9, $\beta_2$=0.999) |
| Learning rate | $5 \times 10^{-4}$ |
| LR schedule | Multi-step, $10\times$ decay at 100k steps |
| Frozen parts | Base LLaMA-2-7B (+ inner ViT for perceptual memory) |
| Parallelism | DDP, $2\times$H200 |

## B.5. Training Details for MemoryVLA

MemoryVLA (Peller-Konrad et al., 2023) augments a CogACT-style diffusion policy (Li et al., 2024a) with an explicit episodic memory; we adapt it to the RoboMME benchmark. Its backbone is a Prismatic VLM (Karamcheti et al., 2024) and a Diffusion Transformer (DiT) action head. Because MemoryVLA already contains dedicated memory-augmented modules, we keep its native design rather than using our proposed memory integration mechanisms.

We keep all hyperparameters the same as in their LIBERO training and testing setups, while only changing the batch size to 64 and total training steps to 160k. We adopt the EEF action space for training as in their official implementation.

Our code is available at `https://github.com/RoboMME/MemoryVLA`.

*Table 9.* Training hyperparameters for MemoryVLA.

| Hyperparameter | Value |
| --- | --- |
| Base VLM | Prismatic VLM |
| Batch size | 64 |
| Total steps | 160,000 |
| Action head | DiT-L diffusion (depth 24, width 1024, 16 heads) |
| Action horizon | 16 (1 current $+$ 15 future) |
| Action space | EEF |
| Proprioception | False |
| Optimizer | AdamW, weight decay 0 |
| Gradient clip norm | 1.0 |
| LR | $2 \times 10^{-5}$, constant (no warmup/decay) |
| Parallelism | FSDP 2xH200 |

### B.6. Training Details for Diffusion Policy

As a non-VLA reference point, we train a standard Diffusion Policy (Chi et al., 2025a) (DP) baseline on RoboMME. DP is a lightweight from-scratch policy with no large-scale pretraining and no explicit memory: it conditions only on a short observation window. Per camera (front and wrist, $256 \times 256$), a ResNet-18 encoder with GroupNorm and a SpatialSoftmax keypoint head (32 keypoints) produces a compact feature; these are concatenated with the 8-D robot state and a projected CLIP (ViT-B/32, frozen) language embedding to form the global conditioning vector. A FiLM-modulated 1D temporal U-Net denoises the action chunk under a DDPM schedule. We use min-max action normalization as in the original implementation, while all other methods in our experiments use quantile normalization (q1 and q99).

Our code is available at `https://github.com/RoboMME/DP`.

*Table 10.* Training hyperparameters for Diffusion Policy on RoboMME.

| Hyperparameter | Value |
| --- | --- |
| Model | 1D U-Net (down dims $[256, 512, 1024]$) |
| Visual encoder | ResNet-18 + SpatialSoftmax |
| Batch size | 128, single GPU |
| Total steps | 200,000 |
| Conditioning | 2 cameras + 8-D state + CLIP ViT-B/32 text (frozen) |
| Diffusion | DDPM, 100 steps, cosine, $\epsilon$-pred, clip sample |
| Action horizon | 16 |
| Action space | Joint space |
| Proprioception | True |
| Optimizer | AdamW ($\beta = [0.95, 0.999]$, weight decay $1 \times 10^{-6}$) |
| LR | $1 \times 10^{-4}$, cosine, 500 warmup steps |
| EMA | 0.9999 |
| Frozen components | CLIP text encoder |

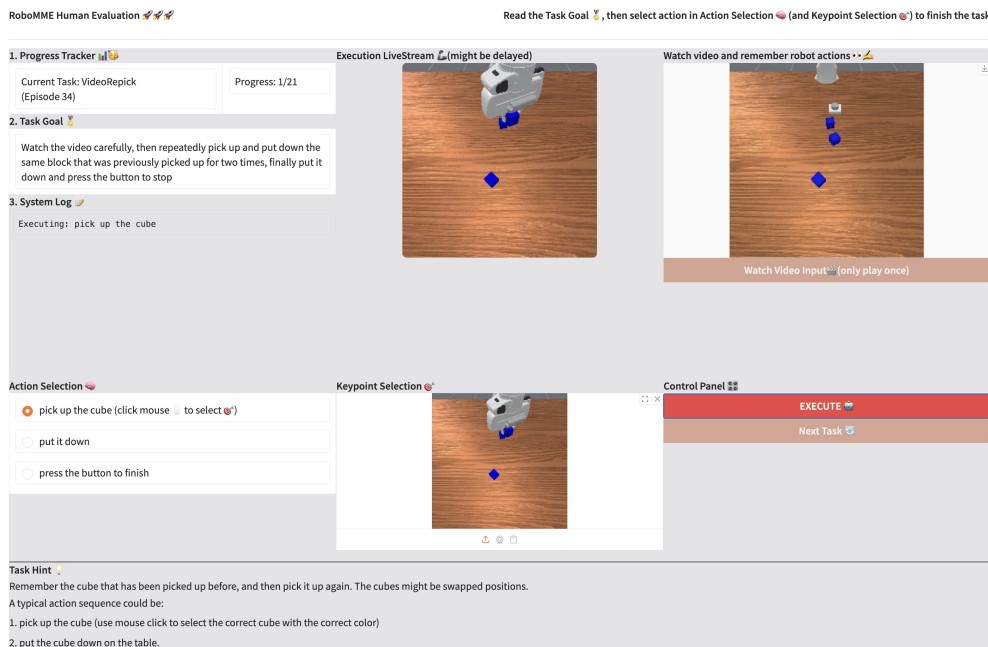

*Figure 8.* Human Study GUI.

### B.7. Oracle Planner and Human Study

To estimate an approximate upper bound on task performance, we conduct two forms of oracle evaluation. First, we evaluate policies using ground-truth subgoals. Second, we reformulate each task as a VideoQA-style decision problem: video clips are revealed incrementally, and humans or models select the correct high-level action from a predefined action set with corresponding grounding information by clicking. An oracle planner then executes the selected actions to complete the task. Note that our oracle planner always guarantees successful low-level motion planning as long as a high-level action is chosen, e.g., `pick up the cube (u,v)` means picking up any pickable cubes nearest to the coordinate (u, v) in the image.

We evaluate several proprietary foundation models under this protocol and recruit 18 human participants to complete a total of 800 evaluation episodes. A screenshot of the graphical user interface is shown in Figure 8.

# C. Full Results on RoboMME

## C.1. Results Using the Oracle Planner

Results using the oracle planner are reported in Table 11. For simplicity, we evaluate each episode with a single run.

We observe that simply prompting existing video foundation models yields limited performance, suggesting a substantial domain shift between our environment and their pretraining data. This observation is consistent with findings reported in MemER (Sridhar et al., 2026). In particular, we find that these models struggle to translate visual understanding into reliable sequential decision-making without task-specific adaptation or fine-tuning.

Human participants, in contrast, perform well on most tasks. However, failures still occur on tasks with strong memory demands. In particular, `Permanence` tasks require sustained attention over long horizons, leading to noticeable performance degradation. Tasks such as *DrawPattern* and *RouteStick* also exhibit lower success rates, primarily because participants tend to forget earlier trajectory details or intermediate goals. *SwingXTimes* and *StopCube* show similarly low performance, as the video clips are revealed incrementally to mimic online execution. After selecting several correct high-level actions, participants often lose track of how many actions they have already taken.

These results indicate that RoboMME imposes substantial long-term memory demands even for humans.

*Table 11.* Performance using the oracle planner for both foundation models and human participants.

| Method | Counting | | | | Permanence | | | | Reference | | | | Imitation | | | | AVG |
|---|---|---|---|---|---|---|---|---|---|---|---|---|---|---|---|---|---|
| | Bin Fill | Pick Xtimes | Swing Xtimes | Stop Cube | Video Umsk | Button Umsk | Video UmskS | Button UmskS | Pick HighL | Video Repick | Video PlcBtn | Video PlcOrd | Move Cube | Insert Peg | Draw Pattern | Route Stick | |
| HUMAN | **96.00** | **100.0** | **80.00** | **78.00** | **90.00** | **92.00** | **92.00** | **90.00** | **92.00** | **92.00** | **98.00** | **90.00** | **90.00** | **98.00** | **84.00** | **86.00** | **90.50** |
| Gemini-2.5-Pro | 78.00 | 80.00 | 58.00 | 32.00 | 86.00 | 82.00 | 18.00 | 12.00 | 78.00 | 28.00 | 50.00 | 60.00 | 66.00 | 32.00 | 4.00 | 2.00 | 47.88 |
| GPT4o | 50.00 | 58.00 | 32.00 | 18.00 | 32.00 | 60.00 | 28.00 | 34.00 | 30.00 | 14.00 | 18.00 | 20.00 | 70.00 | 18.00 | 6.00 | 2.00 | 30.63 |
| Qwen3-VL-32B | 54.00 | 60.00 | 54.00 | 32.00 | 26.00 | 28.00 | 22.00 | 14.00 | 14.00 | 32.00 | 26.00 | 14.00 | 66.00 | 2.00 | 2.00 | 0.00 | 27.88 |

## C.2. Results Based on Each Task Suite

We further provide detailed comparisons of all models in the MME-VLA suite and two selected prior methods, SAM2Act and MemER, on each task suite, as shown in Table 12. All our MME-VLA models are evaluated over nine runs, while the prior methods are evaluated over three runs for simplicity.

*Table 12.* Comparison over MME-VLA suite and prior methods per task suite. **Red** marks the best per section; ☐ marks the overall best for non-oracle models.

| Representation | Policy | Integration/VLM | Counting | Permanence | Reference | Imitation | AVG |
|---|---|---|---|---|---|---|---|
| No Memory ($\pi_{0.5}$) | – | – | 28.78 | 17.00 | 17.17 | 8.78 | $17.93_{\pm 0.61}$ |
| *Symbolic* | SIMPLESG | Oracle | 82.56 | 21.56 | 32.28 | 61.94 | $49.58_{\pm 1.33}$ |
| | GROUNDSG | Oracle | **83.86** | **93.31** | **95.17** | **63.98** | $\mathbf{84.08}_{\pm 1.86}$ |
| | SIMPLESG | Gemini | 39.00 | 13.50 | 22.50 | 18.00 | $23.25_{\pm 0.35}$ |
| | | QwenVL | **44.61** | 19.61 | 25.22 | **26.56** | $29.00_{\pm 0.39}$ |
| | GROUNDSG | Gemini | 12.75 | 15.75 | 11.25 | 6.5 | $11.56_{\pm 0.97}$ |
| | | QwenVL | 38.00 | **39.34** | **31.56** | 21.89 | $\mathbf{32.70}_{\pm 0.61}$ |
| *Perceptual* | TOKENDROP | Context | 57.89 | **26.89** | 23.72 | 29.43 | $34.50_{\pm 0.89}$ |
| | | Modul | 52.33 | 26.83 | 34.72 | 38.28 | $38.04_{\pm 0.82}$ |
| | | Expert | 59.44 | 23.61 | 25.33 | 31.06 | $34.86_{\pm 0.93}$ |
| | FRAMESAMP | Context | 50.14 | 23.31 | 20.95 | 28.33 | $30.68_{\pm 1.17}$ |
| | | Modul | 65.22 | 25.11 | **36.33** | 51.39 | $\mathbf{44.51}_{\pm 0.77}$ |
| | | Expert | 66.78 | 25.17 | 24.11 | 28.94 | $36.25_{\pm 0.68}$ |
| *Recurrent* | TTT | Context | **36.06** | 21.39 | **22.00** | 9.67 | $22.28_{\pm 1.57}$ |
| | | Modul | 34.53 | 22.17 | 20.42 | 10.72 | $21.96_{\pm 2.39}$ |
| | | Expert | 36.00 | **22.95** | 19.67 | **10.78** | $\mathbf{22.35}_{\pm 1.74}$ |
| | RMT | Context | 32.17 | 15.39 | 19.72 | 10.56 | $19.46_{\pm 2.73}$ |
| | | Modul | 34.14 | 15.78 | 21.11 | 9.67 | $20.17_{\pm 1.27}$ |
| | | Expert | 34.39 | 15.72 | 14.34 | 7.75 | $18.05_{\pm 1.66}$ |
| SAM2Act+ (Sridhar et al., 2026) | | | 35.33 | 26.00 | 16.83 | 7.33 | $21.37_{\pm 1.82}$ |
| MemER (Sridhar et al., 2026) | | | **48.83** | 53.17 | 38.00 | **29.50** | $42.38_{\pm 0.33}$ |

## C.3. Results Based on Task Functional Requirements

We observe that no single method fully solves an entire task suite, suggesting that task demands cannot be explained by a simple one-to-one correspondence between cognitive memory types and model memory representations. To better analyze task differences, we regroup tasks according to their functional characteristics, as summarized in Table 13. The regrouped results are given in Tab 14, where we compare our MME-VLA variants trained with symbolic and perceptual memory, as well as the strongest prior method MemER.

*Table 13.* Task categorization based on functional characteristics.

| Category | Functional Requirement | Tasks |
|---|---|---|
| **Motion-Centric** | Sustained trajectory tracking and structured continuous motions (e.g., linear, circular, insertion, swinging) | *PatternLock, RouteStick, SwingXtimes, InsertPeg* |
| **Time-Sensitive** | Precise temporal coordination; success depends critically on acting at the correct moment | *StopCube* |
| **Short-Horizon Video Reasoning** | Grounding objects from short video history in mostly static or mildly dynamic scenes | *VideoUnmask, VideoRepick, VideoUnmaskSwap* |
| **Long-Horizon Video Reasoning** | Integrating information across extended temporal sequences (often > 1000 steps) before acting | *VideoPlaceButton, VideoPlaceOrder* |
| **Dynamic Online Scene-Change** | Continual belief updates and online adaptation under concurrent environmental changes during execution | *ButtonUnmask, ButtonUnmaskSwap, PickHighlight* |
| **Event-Salient** | progress is marked by visually salient, discrete events that clearly signal subtask completion or manipulation intent | *PickXtimes, BinFill, MoveCube* |

*Table 14.* Performance grouped by task functional requirements. Results are averaged within each category.

| Representation | Policy | Integration/ VLM Type | Motion-Centric | Time-Sensitive | Short-Horizon | Long-Horizon | Scene-Change | Event-Salient |
|---|---|---|---|---|---|---|---|---|
| **No Memory** ($\pi_0$) | | | 11.17 | 6.67 | 16.39 | 28.45 | 13.41 | 32.96 |
| **Symbolic** | SIMPLESG | Gemini | **14.00** | 2.00 | 19.33 | 27.50 | 10.33 | 56.67 |
| | | QwenVL | 7.34 | 0.44 | 24.96 | 29.22 | 15.33 | **84.96** |
| | GROUNDSG | Gemini | 3.25 | **3.00** | 22.00 | 9.50 | 7.67 | 20.33 |
| | | QwenVL | 5.83 | 0.00 | **48.22** | **42.89** | **17.70** | 72.08 |
| **Perceptual** | TOKENDROP | Context | 32.84 | 3.11 | 25.93 | 28.22 | 22.74 | 71.70 |
| | | Modul | 44.28 | 5.33 | 26.22 | **47.78** | **24.00** | 60.00 |
| | | Expert | 32.11 | 4.22 | 22.00 | 30.56 | 22.89 | **76.45** |
| | FRAMESAMP | Context | 27.45 | 13.67 | 21.00 | 25.45 | 21.04 | 63.48 |
| | | Modul | **54.95** | **42.00** | **29.18** | 46.00 | 22.07 | 68.22 |
| | | Expert | 31.84 | 28.89 | 25.93 | 27.11 | 21.70 | 75.55 |
| SAM2Act+ (Fang et al., 2025) | | | 6.33 | 0.00 | 16.89 | 22.34 | 25.33 | 48.44 |
| MemER (Sridhar et al., 2026) | | | 23.67 | 0.00 | **48.22** | 28.00 | **54.67** | 72.89 |

## C.4. Full Results on All Tasks

We provide complete per-task results for all MME-VLA models and additional baselines in Table 15. We train a U-Net-based Diffusion Policy (Chi et al., 2025a) and MemoryVLA (Shi et al., 2026); both achieve an overall success rate below 10%.

We also adapt our memory representations and integration strategies to the OpenVLA-OFT (Kim et al., 2025a) backbone, but find that it is less effective than the $\pi_{0.5}$ backbone in our setting. Specifically, OpenVLA-OFT with SIMPLESG+QwenVL achieves 21.6%, OpenVLA-OFT with GROUNDSG+QwenVL achieves 19.5%, OpenVLA-OFT with TOKENDROP+Context achieves 9.1%, and OpenVLA-OFT with FRAMESAMP+Context achieves 9.6%.

We observe that OpenVLA has more difficulty leveraging grounded symbolic information than $\pi_{0.5}$. One likely reason is that $\pi_{0.5}$ is co-trained on large-scale visual grounding data and can better consume or predict pixel-level locations, whereas OpenVLA does not have the same grounding capability. We also find that perceptual memory is less effective for OpenVLA-OFT, bringing limited gains compared with symbolic memory. In contrast, the $\pi_{0.5}$ backbone yields more stable and consistent gains when incorporating various memory representations.

We will continue improving these models and update the results on our leaderboard: robomme.github.io/leaderboard.html.

*Table 15.* Full results for different models. **Red** marks the best per section; ☐ marks the overall best for non-oracle models.

| Representation | Policy | Integration/ VLM Type | Counting | | | | Permanence | | | |
|---|---|---|---|---|---|---|---|---|---|---|
| | | | **BinFill** | **PickXTimes** | **SwingXTimes** | **StopCube** | **VideoUnmask** | **ButtonUnmask** | **VideoUnmaskSwap** | **ButtonUnmaskSwap** |
| **Symbolic** | SIMPLESG | Oracle | 85.78 ± 2.91 | 99.78 ± 0.67 | **100.0 ± 0.00** | 44.67 ± 22.25 | 33.11 ± 2.67 | 22.00 ± 3.06 | 15.56 ± 2.60 | 15.56 ± 3.84 |
| | GROUNDSG | Oracle | **85.78 ± 1.67** | **100.00 ± 0.00** | **100.0 ± 0.00** | **49.67 ± 20.02** | **98.78 ± 1.04** | **95.00 ± 2.83** | **99.22 ± 1.04** | **80.22 ± 15.69** |
| **Symbolic** | SIMPLESG | Gemini | 46.00 ± 7.14 | 63.00 ± 7.07 | **45.00 ± 4.24** | 2.00 ± 0.00 | 29.00 ± 1.41 | 9.00 ± 7.07 | 14.00 ± 2.83 | 2.00 ± 2.83 |
| | SIMPLESG | QwenVL | 77.56 ± 2.19 | 95.33 ± 2.00 | 5.11 ± 1.45 | 0.44 ± 0.88 | 34.22 ± 3.53 | 19.33 ± 8.31 | 15.33 ± 2.83 | 9.56 ± 4.56 |
| | GROUNDSG | Gemini | 26.00 ± 8.49 | 18.00 ± 0.00 | 4.00 ± 0.00 | **3.00 ± 1.41** | 36.00 ± 0.00 | 14.00 ± 2.83 | 13.00 ± 1.41 | 0.00 ± 0.00 |
| | GROUNDSG | QwenVL | 52.00 ± 2.00 | 92.67 ± 3.06 | 7.33 ± 6.11 | 0.00 ± 0.00 | 88.67 ± 1.15 | **24.00 ± 2.00** | 30.67 ± 1.15 | **14.00 ± 2.00** |
| **Perceptual** | TOKENDROP | Context | 48.67 ± 4.58 | 85.11 ± 4.14 | 94.67 ± 3.00 | 3.11 ± 3.89 | **33.78 ± 5.33** | **31.56 ± 3.43** | 26.22 ± 2.11 | 16.00 ± 5.92 |
| | TOKENDROP | Modul | 34.44 ± 3.43 | 83.56 ± 8.59 | 86.00 ± 5.92 | 5.33 ± 4.24 | 28.22 ± 2.91 | 29.33 ± 3.00 | **28.44 ± 3.71** | **21.33 ± 5.29** |
| | TOKENDROP | Expert | 54.22 ± 3.07 | **87.56 ± 3.84** | 91.78 ± 3.07 | 4.22 ± 5.70 | 26.67 ± 2.65 | 30.44 ± 4.45 | 18.44 ± 3.13 | 18.89 ± 4.48 |
| | FRAMESAMP | Context | 41.22 ± 4.27 | 72.00 ± 3.74 | 73.67 ± 11.08 | 13.67 ± 8.82 | 26.89 ± 5.28 | 30.22 ± 3.52 | 20.89 ± 2.94 | 15.22 ± 3.93 |
| | FRAMESAMP | Modul | 39.56 ± 5.27 | 87.33 ± 2.45 | 92.00 ± 2.24 | 42.00 ± 11.36 | 32.67 ± 3.16 | 25.11 ± 3.18 | 24.44 ± 6.06 | 18.22 ± 3.80 |
| | FRAMESAMP | Expert | **57.33 ± 4.90** | 86.22 ± 4.52 | **94.67 ± 2.45** | 28.89 ± 9.33 | 31.78 ± 5.43 | 25.78 ± 3.38 | 22.89 ± 2.26 | 20.22 ± 3.07 |
| **Recurrent** | TTT | Context | **35.56 ± 5.64** | 62.89 ± 8.01 | **42.44 ± 7.44** | 3.33 ± 2.65 | 29.78 ± 4.06 | **22.89 ± 3.89** | 18.44 ± 3.28 | 14.44 ± 5.90 |
| | TTT | Modul | 34.22 ± 8.49 | **65.11 ± 5.41** | 36.67 ± 8.97 | 2.11 ± 2.83 | 27.22 ± 9.90 | 22.11 ± 1.41 | **25.00 ± 12.73** | 14.11 ± 2.83 |
| | TTT | Expert | 34.89 ± 6.57 | 63.78 ± 6.12 | 41.33 ± 9.95 | 4.00 ± 3.46 | **31.78 ± 6.12** | 22.44 ± 3.13 | 19.56 ± 6.98 | **18.00 ± 6.86** |
| | RMT | Context | 32.44 ± 7.77 | 56.89 ± 9.12 | 33.56 ± 7.99 | **5.78 ± 5.52** | 31.33 ± 7.87 | 10.89 ± 9.06 | 17.33 ± 5.48 | 2.00 ± 2.65 |
| | RMT | Modul | 33.33 ± 5.00 | 60.78 ± 7.45 | 37.78 ± 6.20 | 4.67 ± 5.48 | 31.11 ± 5.21 | 11.78 ± 9.92 | 17.78 ± 3.38 | 2.44 ± 2.79 |
| | RMT | Expert | 35.78 ± 7.90 | 60.22 ± 8.27 | 36.00 ± 8.34 | 5.56 ± 5.81 | 28.00 ± 3.16 | 17.11 ± 9.79 | 15.78 ± 5.52 | 2.00 ± 2.65 |
| DP-Unet (Chi et al., 2025a) | | | 32.00 ± 2.67 | 2.67 ± 3.56 | 6.67 ± 1.78 | 0.00 ± 0.00 | 16.00 ± 8.00 | 4.00 ± 2.67 | 4.00 ± 0.00 | 0.00 ± 0.00 |
| MemoryVLA (Shi et al., 2026) | | | 10.00 ± 2.00 | 17.33 ± 9.87 | 1.33 ± 1.15 | 0.00 ± 0.00 | 16.00 ± 5.29 | 8.00 ± 5.29 | 4.67 ± 1.15 | 4.67 ± 1.15 |
| OpenVLA-OFT (Kim et al., 2025a) | | | 8.00 ± 0.00 | 8.00 ± 0.00 | 2.00 ± 0.00 | 2.00 ± 0.00 | 16.00 ± 0.00 | 0.00 ± 0.00 | 18.00 ± 0.00 | 0.00 ± 0.00 |
| OpenVLA-OFT (w/ SIMPLESG+QwenVL) | | | 48.00 ± 0.00 | 50.00 ± 0.00 | 20.00 ± 0.00 | 2.00 ± 0.00 | 22.00 ± 0.00 | 0.00 ± 0.00 | 24.00 ± 0.00 | 2.00 ± 0.00 |
| OpenVLA-OFT (w/ GROUNDSG+QwenVL) | | | 44.00 ± 0.00 | 40.00 ± 0.00 | 16.00 ± 0.00 | 0.00 ± 0.00 | 46.00 ± 0.00 | 2.00 ± 0.00 | 22.00 ± 0.00 | 0.00 ± 0.00 |
| OpenVLA-OFT (w/ TOKENDROP+Context) | | | 16.00 ± 0.00 | 14.00 ± 0.00 | 6.00 ± 0.00 | 0.00 ± 0.00 | 12.00 ± 0.00 | 0.00 ± 0.00 | 8.00 ± 0.00 | 0.00 ± 0.00 |
| OpenVLA-OFT (w/ FRAMESAMP+Context) | | | 6.00 ± 0.00 | 10.00 ± 0.00 | 2.00 ± 0.00 | 0.00 ± 0.00 | 6.00 ± 0.00 | 0.00 ± 0.00 | 8.00 ± 0.00 | 0.00 ± 0.00 |
| $\pi_{0.5}$ (Black et al., 2025) | | | 30.00 ± 6.48 | 42.89 ± 6.41 | 35.56 ± 5.27 | **6.67 ± 5.20** | 20.44 ± 2.96 | 22.22 ± 10.22 | 18.67 ± 4.24 | 6.67 ± 4.12 |
| $\pi_{0.5}$ w/ past actions (Bu et al., 2025) | | | 26.67 ± 4.84 | 58.33 ± 6.20 | 26.67 ± 4.15 | 4.67 ± 4.41 | 30.67 ± 5.72 | 23.67 ± 3.29 | 20.67 ± 3.01 | 16.00 ± 2.42 |
| SAM2Act+ (Fang et al., 2025) | | | 40.00 ± 5.29 | 76.00 ± 2.00 | 25.33 ± 1.15 | 0.00 ± 0.00 | 27.33 ± 1.15 | 32.00 ± 0.00 | 18.00 ± 5.29 | **26.67 ± 1.15** |
| MemER (Sridhar et al., 2026) | | | **56.67 ± 3.06** | **79.33 ± 4.16** | **59.33 ± 8.33** | 0.00 ± 0.00 | **81.33 ± 3.06** | **72.00 ± 3.46** | **38.00 ± 0.00** | 21.33 ± 3.06 |

| Representation | Policy | Integration/ VLM Type | Reference | | | | Imitation | | | |
|---|---|---|---|---|---|---|---|---|---|---|
| | | | **PickHighlight** | **VideoRepick** | **VideoPlaceButton** | **VideoPlaceOrder** | **MoveCube** | **InsertPeg** | **PatternLock** | **RouteStick** |
| **Symbolic** | SIMPLESG | Oracle | 44.00 ± 4.69 | 27.78 ± 2.33 | 31.33 ± 4.90 | 26.00 ± 3.00 | 87.33 ± 3.46 | 10.00 ± 3.32 | 95.33 ± 2.24 | 55.11 ± 5.49 |
| | GROUNDSG | Oracle | **83.33 ± 5.99** | **97.33 ± 1.04** | **100.00 ± 0.00** | **100.00 ± 3.00** | **87.78 ± 4.83** | **15.56 ± 3.66** | **97.00 ± 2.62** | **55.56 ± 6.02** |
| **Symbolic** | SIMPLESG | Gemini | **20.00 ± 2.83** | 15.00 ± 1.41 | 26.00 ± 8.49 | 29.00 ± 7.07 | 61.00 ± 1.41 | **4.00 ± 0.00** | 7.00 ± 1.41 | 0.00 ± 0.00 |
| | SIMPLESG | QwenVL | 17.11 ± 3.18 | 25.33 ± 3.74 | 33.33 ± 3.46 | 25.11 ± 4.81 | **82.00 ± 6.78** | 3.78 ± 1.56 | **12.67 ± 2.00** | **7.78 ± 3.07** |
| | GROUNDSG | Gemini | 9.00 ± 4.24 | 17.00 ± 1.41 | 12.00 ± 5.66 | 7.00 ± 1.41 | 17.00 ± 4.24 | 0.00 ± 0.00 | 7.00 ± 4.24 | 2.00 ± 0.00 |
| | GROUNDSG | QwenVL | 15.11 ± 1.15 | **25.33 ± 1.15** | **54.00 ± 4.00** | **31.78 ± 1.15** | 71.56 ± 1.15 | 3.33 ± 1.15 | 6.67 ± 2.31 | 6.00 ± 2.00 |
| **Perceptual** | TOKENDROP | Context | 20.67 ± 4.24 | 17.78 ± 2.11 | 31.11 ± 3.48 | 25.33 ± 5.20 | 81.33 ± 4.12 | 4.00 ± 3.32 | 12.67 ± 4.00 | 20.00 ± 3.16 |
| | TOKENDROP | Modul | 21.33 ± 5.83 | 22.00 ± 3.16 | 59.56 ± 2.40 | **36.00 ± 3.61** | 62.00 ± 3.46 | 7.11 ± 3.33 | 32.44 ± 7.40 | 51.56 ± 6.62 |
| | TOKENDROP | Expert | 19.33 ± 3.00 | 20.89 ± 4.37 | 36.44 ± 2.96 | 24.67 ± 3.46 | **87.56 ± 6.77** | 2.22 ± 1.86 | 16.22 ± 3.07 | 18.22 ± 4.94 |
| | FRAMSAMP | Context | 17.67 ± 2.83 | 15.22 ± 4.08 | 30.00 ± 4.68 | 20.89 ± 6.28 | 77.22 ± 7.87 | 1.22 ± 1.63 | 15.22 ± 2.34 | 19.67 ± 3.67 |
| | FRAMSAMP | Modul | **22.89 ± 3.89** | **30.44 ± 5.81** | **60.00 ± 4.00** | 32.00 ± 3.87 | 77.78 ± 3.80 | **7.56 ± 3.57** | **53.56 ± 4.56** | **66.67 ± 4.12** |
| | FRAMSAMP | Expert | 19.11 ± 4.59 | 23.11 ± 5.75 | 30.00 ± 5.10 | 24.22 ± 4.29 | 83.11 ± 5.21 | 2.00 ± 2.00 | 13.56 ± 2.40 | 17.11 ± 2.26 |
| **Recurrent** | TTT | Context | **20.44 ± 6.15** | **13.11 ± 3.76** | **34.22 ± 8.33** | 20.22 ± 5.04 | 32.44 ± 6.15 | 1.11 ± 1.05 | 1.56 ± 2.60 | 3.56 ± 2.96 |
| | TTT | Modul | 14.56 ± 5.66 | 12.11 ± 2.83 | 32.67 ± 2.83 | 22.33 ± 1.21 | 31.22 ± 4.24 | 1.11 ± 1.41 | 3.56 ± 4.24 | 7.00 ± 1.41 |
| | TTT | Expert | 12.22 ± 3.80 | 9.56 ± 3.43 | 34.00 ± 6.00 | 22.89 ± 4.59 | **33.56 ± 6.06** | 0.89 ± 1.05 | 3.11 ± 2.85 | 5.56 ± 3.28 |
| | RMT | Context | 14.00 ± 5.66 | 3.78 ± 2.11 | 32.00 ± 8.06 | 29.11 ± 8.25 | 25.78 ± 8.60 | 2.00 ± 2.00 | **5.56 ± 4.67** | **8.89 ± 2.85** |
| | RMT | Modul | 17.11 ± 3.33 | 4.22 ± 2.91 | 32.00 ± 4.24 | **31.11 ± 6.72** | 24.67 ± 3.32 | **2.21 ± 2.00** | 3.78 ± 4.18 | 8.00 ± 4.69 |
| | RMT | Expert | 11.78 ± 3.53 | 0.22 ± 0.67 | 24.22 ± 3.80 | 22.67 ± 9.80 | 20.00 ± 5.39 | 1.89 ± 1.41 | 4.22 ± 3.80 | 4.89 ± 3.62 |
| DP-Unet (Chi et al., 2025a) | | | 0.00 ± 0.00 | 0.00 ± 0.00 | 22.00 ± 0.00 | 5.67 ± 3.11 | 8.67 ± 1.78 | 0.00 ± 0.00 | 2.00 ± 2.00 | 10.00 ± 0.00 |
| MemoryVLA (Shi et al., 2026) | | | 7.33 ± 3.06 | 0.00 ± 0.00 | 13.33 ± 2.31 | 4.00 ± 4.00 | 14.00 ± 3.46 | 0.00 ± 0.00 | 12.00 ± 2.00 | 1.33 ± 1.15 |
| OpenVLA-OFT (Kim et al., 2025a) | | | 10.00 ± 0.00 | 0.00 ± 0.00 | 28.00 ± 0.00 | 20.00 ± 0.00 | 18.00 ± 0.00 | 4.00 ± 0.00 | 6.00 ± 0.00 | 8.00 ± 0.00 |
| OpenVLA-OFT (w/ SIMPLESG+QwenVL) | | | 22.00 ± 0.00 | **34.00 ± 0.00** | 30.00 ± 0.00 | 18.00 ± 0.00 | 48.00 ± 0.00 | 2.00 ± 0.00 | 14.00 ± 0.00 | 10.00 ± 0.00 |
| OpenVLA-OFT (w/ GROUNDSG+QwenVL) | | | 18.00 ± 0.00 | 20.00 ± 0.00 | 32.00 ± 0.00 | 24.00 ± 0.00 | 26.00 ± 0.00 | 2.00 ± 0.00 | 12.00 ± 0.00 | 8.00 ± 0.00 |
| OpenVLA-OFT (w/ TOKENDROP+Context) | | | 10.00 ± 0.00 | 8.00 ± 0.00 | 20.00 ± 0.00 | 16.00 ± 0.00 | 18.00 ± 0.00 | 0.00 ± 0.00 | 6.00 ± 0.00 | 12.00 ± 0.00 |
| OpenVLA-OFT (w/ FRAMESAMP+Context) | | | 10.00 ± 0.00 | 4.00 ± 0.00 | 20.00 ± 0.00 | 18.00 ± 0.00 | 26.00 ± 0.00 | 0.00 ± 0.00 | 14.00 ± 0.00 | **28.00 ± 0.00** |
| $\pi_{0.5}$ (Black et al., 2025) | | | 11.33 ± 4.90 | 0.44 ± 0.88 | **31.11 ± 4.01** | 25.78 ± 1.86 | 26.00 ± 3.61 | 1.56 ± 1.94 | 2.89 ± 2.26 | 4.67 ± 2.45 |
| $\pi_{0.5}$ w/ past actions (Bu et al., 2025) | | | 12.33 ± 4.13 | 8.67 ± 3.72 | 24.00 ± 6.02 | 18.67 ± 6.45 | 34.00 ± 5.93 | 1.00 ± 0.00 | 4.00 ± 2.94 | 5.67 ± 2.34 |
| SAM2Act+ (Fang et al., 2025) | | | 17.33 ± 2.31 | 5.33 ± 2.31 | 24.67 ± 4.16 | 20.00 ± 2.00 | 29.33 ± 2.31 | 0.00 ± 0.00 | 0.00 ± 0.00 | 0.00 ± 0.00 |
| MemER (Sridhar et al., 2026) | | | **70.67 ± 4.16** | 25.33 ± 6.11 | 30.00 ± 3.46 | **26.00 ± 2.00** | **82.67 ± 3.06** | **6.67 ± 1.15** | **16.67 ± 1.15** | 12.00 ± 3.46 |

*Table 16.* Effect of memory token budget on perceptual memory methods. Increasing the memory budget generally improves performance.

| Memory Tokens | FRAMESAMP+Modul | TOKENDROP+Modul |
|:---:|:---:|:---:|
| 64 | 30.42 | 18.11 |
| 128 | 36.54 | 32.76 |
| 256 | 42.90 | 36.36 |
| 512 | 44.51 | 38.02 |
| 1024 | 45.87 | 40.17 |

*Table 17.* Attention allocation among memory, observation, and language tokens. We report mean / max attention probability mass (%) aggregated across layers.

| Method | Memory | Observation | Language | Success |
|:---|:---:|:---:|:---:|:---:|
| FRAMESAMP+Context | 1.5 / 8.9 | **60.2 / 84.9** | 38.3 / 63.9 | 30.68 |
| TOKENDROP+Context | 4.8 / 17.0 | 45.4 / 72.9 | 49.2 / 94.8 | 34.50 |
| TOKENDROP+Expert | 0.7 / 12.7 | 47.5 / 71.9 | 51.8 / **97.8** | 34.86 |
| FRAMESAMP+Expert | 1.3 / **21.8** | 46.0 / 68.2 | **52.7** / 96.4 | 36.25 |

## C.5. Discussion on Memory Token Contribution

To better understand how memory contributes to MME-VLA performance, we conduct two additional analyses using perceptual memory as a representative case study.

**Effect of memory budget.** We first study how performance changes with the memory budget. Specifically, we evaluate FRAMESAMP+Modul and TOKENDROP+Modul with memory budgets ranging from 64 to 1024 tokens. As shown in Table 16, performance consistently improves as the memory budget increases for both methods. For FRAMESAMP+Modul, the success rate increases from 30.42% with 64 memory tokens to 45.87% with 1024 tokens. Similarly, TOKENDROP+Modul improves from 18.11% to 40.17%. These results suggest that additional memory tokens provide useful historical information, although the gains become smaller beyond 512 tokens.

We therefore use 512 memory tokens in our main experiments as a trade-off between performance and computational cost. For higher performance, we can either increase the memory budget further or retain more spatial tokens per selected frame, such as using an $8 \times 8$ or the original $16 \times 16$ tokens, to preserve finer visual details.

**Attention allocation to memory, observation, and language.** We further analyze how much the action tokens attend to memory compared with current observation and language tokens. For memory-as-context and memory-as-expert, we measure the average attention probability mass assigned to memory, observation, and language tokens across all action tokens, and report both the mean and maximum values aggregated across layers. We conduct this analysis on a small-scale evaluation with 10 episodes per task, 160 episodes in total.

As shown in Table 17, memory tokens receive a smaller average attention mass than observation and language tokens. This is expected: most low-level actions can be executed from the current observation, while memory is mainly needed at key decision points where historical information disambiguates the correct behavior. Importantly, however, models with stronger performance often assign higher peak attention to memory or language tokens, suggesting that effective use of memory is sparse but crucial. For example, FRAMESAMP+Expert achieves the best success rate among these variants and also shows the highest maximum memory attention. This indicates that memory tokens may not dominate the entire attention distribution, but can strongly influence action prediction when historical context is needed.

**Effect of memory modulation.** For memory-as-modulator, memory does not enter the model as additional context tokens. Instead, it predicts feature-wise scale and bias terms that directly modulate the action expert. We therefore analyze the distribution of the predicted scale and bias values over all feature dimensions. As shown in Table 18, both TOKENDROP+Modul and FRAMESAMP+Modul produce non-trivial modulation signals, confirming that memory actively changes the action representation rather than being ignored. At the same time, FRAMESAMP+Modul achieves higher success with smaller variance in both scale and bias values, suggesting that stable memory-conditioned modulation is beneficial, whereas overly large variation may perturb the action features and hurt performance.

*Table 18.* Statistics of memory-conditioned modulation. We report mean $\pm$ standard deviation of the predicted scale and bias values.

| Method | Scale | Bias | Success |
|---|---|---|---|
| TokenDrop+Modul | $5.99 \pm 38.67$ | $0.024 \pm 14.19$ | 38.04 |
| FrameSamp+Modul | $3.74 \pm 22.55$ | $0.012 \pm 9.45$ | 44.51 |

Overall, these analyses show that memory contributes to MME-VLA in complementary ways. Increasing the memory budget improves performance, demonstrating that historical information is useful. Attention-based methods use memory sparsely, with memory becoming important mainly at history-dependent decision points. Memory-as-modulator incorporates memory through direct feature-wise conditioning of the action expert, which provides a stronger and more stable mechanism to influence action prediction. These findings help explain why perceptual memory, especially FrameSamp+Modul, achieves the best overall performance in our experiments.

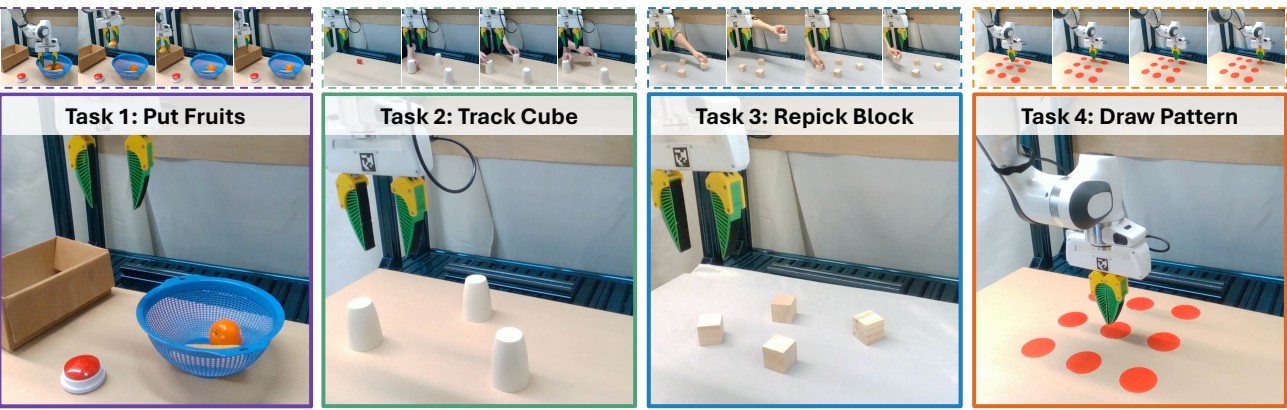

*Figure 9.* We design four history-dependent real-world tasks for policy evaluation. The upper dashed box indicates selected frames from previous video history. The bottom large box indicates current observation for execution.

## D. Real Robot Experiment

We design four real-world tasks (Figure 9), *PutFruits*, *TrackCube*, *RepickBlock*, and *DrawPattern*, to mirror four simulation tasks, *BinFill*, *VideoUnmask(Swap)*, *VideoRepick*, and *PatternLock*, respectively.

- **PutFruits.** The robot transfers a specified number of fruits from a basket to a bin. During execution, a human may intervene by removing placed fruits or adding new ones, preventing the robot from relying solely on immediate perception and requiring it to track progress over time.

- **TrackCube.** The robot selects the correct cup containing a target cube given a pre-recorded video in which a human covers the cubes with multiple cups and optionally swaps or moves their positions.

- **RepickBlock.** Given a pre-recorded video showing a human picking up several blocks, the robot must identify and pick up the same set of blocks again.

- **DrawPattern.** The robot reproduces a movement trajectory demonstrated in a pre-recorded video.

We teleoperate the robot using an Oculus Quest 2 and record demonstrations at 15 Hz. We collect 50 demonstrations for *PutFruits* and 100 demonstrations for each of the other three tasks, as *PutFruits* episodes contain longer trajectories. In total, the dataset comprises 350 demonstrations and 78,400 timesteps.

After data collection, we manually annotate each episode with grounded subgoals at keyframes. For simplicity, we omit grounding annotations for *PutFruits*, as we found that simple goal descriptions are sufficient for counting-based behaviors. Figure 10 illustrates the platform setup. Experiments are conducted using a 7-DoF Franka Emika Panda robot equipped with UMI (Chi et al., 2025b) fin-ray fingers mounted on the default Franka gripper. The robot operates in a tabletop environment with three RGB-D cameras following the DROID setup (Khazatsky et al., 2024): two Intel RealSense D435 cameras mounted on the left and right shoulders and one Intel RealSense D405 mounted on the wrist.

For training, we evaluate our strongest perceptual model (FRAMESAMP+Modul), symbolic model (GROUNDSG + QwenVL), and the $\pi_{0.5}$ baseline for comparison. We adopt a larger memory budget of $B = 2048$, as simulation experiments showed only modest additional computational cost. We use the right-shoulder view image for the VLM subgoal predictor and memory contruction. QwenVL is trained using the same configuration as in simulation. The subgoal annotation for *PutFruits* and *DrawPattern* tasks does not contain grounding information. All other hyperparameters remain unchanged, except that we set the total number of training steps to 50k.

The aggregated results for each task are shown in Table 4. All rollout demonstrations are available on the project website.

We observe similar trends in the real world. Perceptual memory performs best on the motion-centric task *DrawPattern*, whereas symbolic memory performs better on the event-salient task *PutFruits*. Both approaches achieve moderate performance on *RepickBlock*. On *TrackCube*, perceptual memory performs slightly better, as the VLM-based subgoal predictor often fails to produce accurate grounding when dynamic swapping occurs.

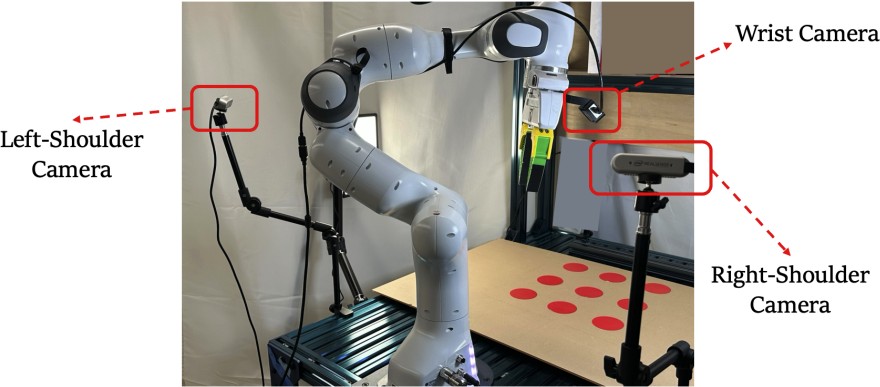

*Figure 10.* Real Robot Experiment Platform Setup.

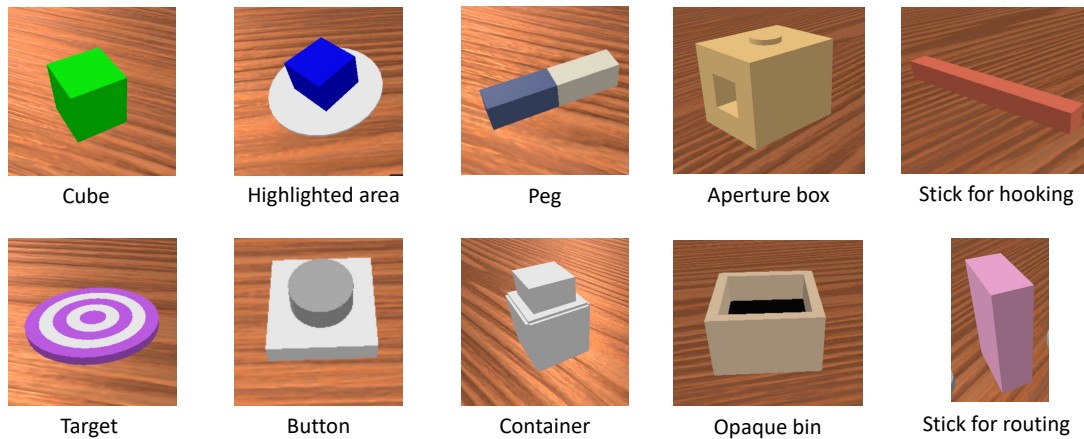

*Figure 11.* Overview of the main assets used in RoboMME.

## E. Task Description

RoboMME is constructed using a carefully designed taxonomy to evaluate four types of memory: temporal, spatial, object, and procedural. These memory types correspond to four task suites, `Counting`, `Permanence`, `Reference`, and `Imitation`, respectively, each comprising four tasks.

Across all tasks, the environments use a fixed set of tabletop objects (see Fig. 11), grouped by their functional roles:

- **Primitive Objects:**

    - **Cubes** (green/blue/red) for basic pick-and-place manipulation.
    - **Pegs** for insertion in the *InsertPeg* task.
    - **Hook stick** for pulling or repositioning objects in the *MoveCube* task.
    - **Routing stick** for navigating around obstacles in the *RouteStick* task.
    - **Highlighted areas**, implemented as white discs beneath objects, that provide temporary visual cues in the *PickHighlight* task.

- **Interactive Objects:**

    - **Targets**, colored discs (purple/red/gray) that change color to indicate state transitions.
    - **Buttons** for press-based interaction, typically serving as termination signals.

- **Specialized Receptacles:**

    - **Containers** that can be picked up to cover and hide cubes, testing spatial memory.
    - **Aperture box** (box with a hole) for peg insertion tasks.
    - **Opaque bin** with a black interior for the *BinFill* task, preventing visual counting and enforcing object-permanence reasoning.

Detailed descriptions of each task are provided below.

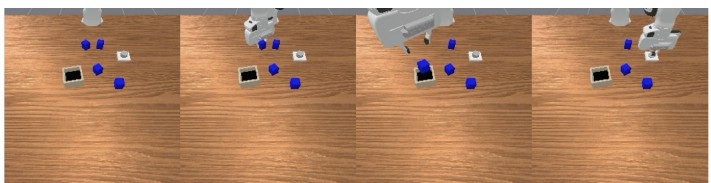

*Figure 12.* **BinFill Task Example.** In this instance, the goal is to place one blue cube into the bin and then press the button to stop.

### E.1. BinFill

*Table 19. BinFill* Task Configuration.

| Difficulty | #Colors | #Total Cubes | #Goal Cubes |
|---|---|---|---|
| Easy | 1 | 4–6 | 1 |
| Medium | 2 | 8–10 | 1–2 |
| Hard | 3 | 10–12 | 2–3 |

The *BinFill* task, illustrated in Fig. 12, evaluates sequential decision-making and temporal memory. The robot must place a specified number of colored cubes into a bin and then press a button to terminate the episode. Success requires accurate counting and timely termination. Task configurations are summarized in Table 19.

**Language Goals.** We define three types of language instructions with increasing compositional complexity:

1. Place **one** cube of a specified color. *e.g., "put one blue cube into the bin and press the button to stop".*

2. Place cubes of **two** specified colors. *e.g., "put one blue cube and two green cubes into the bin and press the button to stop".*

3. Place cubes of **three** specified colors. *e.g., "put one blue cube, one green cube and two red cubes into the bin and press the button to stop".*

**Task Characteristics.** To introduce dynamic and history-dependent behavior, we consider two settings randomly selected at environment initialization:

1. **Static:** all cubes are present at the beginning of the episode.

2. **Streaming:** cubes appear incrementally over time in a random order.

**Success Criteria.**

- **Correct completion:** the button is pressed only after the exact required number of cubes for each specified color has been placed in the bin.

- **Failure:** placing fewer or more cubes than required for any color at evaluation time.

- **Immediate failure:** exceeding the required count for any color will terminate the episode early.

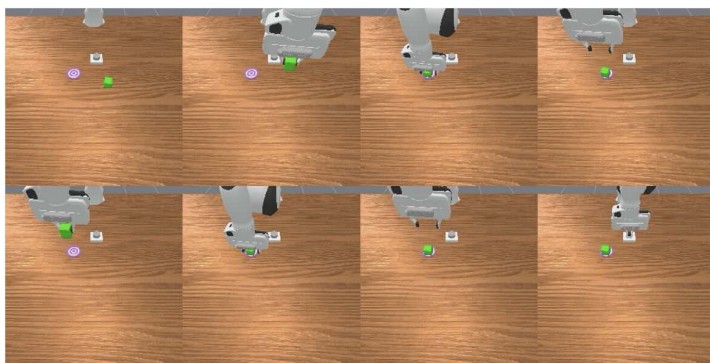

*Figure 13.* **PickXtimes Task Example.** In this instance, the goal is to pick up the green cube and place it on the target twice, then press the button to stop.

## E.2. PickXtimes

*Table 20. PickXtimes* Task Configuration.

| Difficulty | #Total Cubes | #Repetitions |
|---|---|---|
| Easy | 1 | 1–3 |
| Medium | 3 (2 distractors) | 1–3 |
| Hard | 3 (2 distractors) | 4–5 |

The *PickXTimes* task, illustrated in Fig. 13, evaluates the robot's ability to perform iterative pick-and-place actions with temporal memory. The robot repeatedly picks up a cube of a specified color, places it on a target, and completes an exact number of cycles before pressing a button to terminate the task. Task configurations are summarized in Table 20.

**Language Goals.** We define two types of language instructions:

1. Pick and place for **one** time. *e.g., "pick up the blue cube and place it on the target, then press the button to stop"*.

2. Pick and place for **multiple** times. *e.g., "pick up the blue cube and place it on the target, repeating this pick-and-place action three times, then press the button to stop"*.

**Successful Pick-and-Place.** A pick-and-place is considered successful when the robot lifts the cube above a predefined height threshold while maintaining a valid grasp, and then lowers it onto the target surface below a specified placement height.

**Success Criteria.**

- **Correct completion:** the robot completes exactly the specified number of pick-and-place repetitions and then presses the button to terminate the episode.

- **Immediate failure:** The episode terminates early if:
  1. The robot picks up a wrong cube.
  2. The button is pressed before all required repetitions are completed.
  3. The robot performs more pick-and-place repetitions than specified.

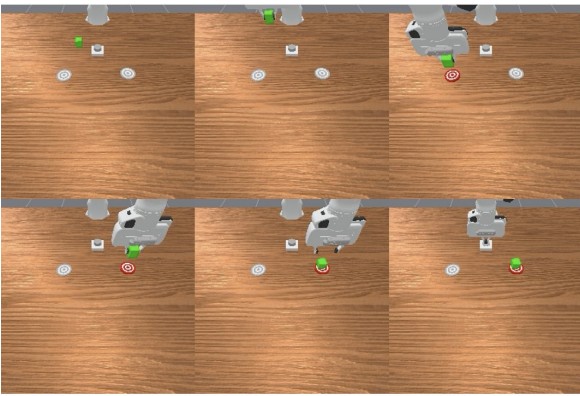

*Figure 14.* **SwingXtimes Task Example.** In this instance, the goal is pick up the green cube, first move it to the right-side target, then place the cube on the left-side target (i.e., swing between targets one time), finally press the button to stop. Spatial directions (e.g., left/right) follow the robot base coordinate frame rather than the front camera frame.

### E.3. SwingXtimes

*Table 21. SwingXtimes Task Configuration.*

| Difficulty | #Total Cubes | #Repetitions |
|---|---|---|
| Easy | 1 | 1–3 |
| Medium | 3 (2 distractors) | 1–2 |
| Hard | 3 (2 distractors) | 3 |

The *SwingXtimes* task is illustrated in Fig. 14. The robot must pick up a specified colored cube, transport it back and forth between two distinct targets (right-side and left-side) for a specified number of repetitions, and then press a button to stop.

**Language Goals.** We define two types of language instructions with increasing compositional complexity:

1. Perform **one** swing cycle. *e.g., "Pick up the green cube, move it to the right-side target and then put down the cube on the left-side target and press the button to stop"*.

2. Perform **multiple** swing cycles. *e.g., "Pick up the green cube, move it to the right-side target and then to the left-side target, repeating this right-to-left swing motion three times, then put down the cube and press the button to stop."*.

**Successful Reach.** A reach is successful when the cube is held nearly upright and positioned within a small tolerance of the target center in both vertical alignment and horizontal offset. Upon success, the target changes from gray to red, indicating that the gripper has reached it.

**Success Criteria.**

• **Correct completion:** The button is pressed only after the robot completes the exact required number of swing motions between the two targets and places the cube on the table.

• **Immediate failure:** The episode terminates early if:

  1. The robot picks up the wrong cube.
  2. The button is pressed before all repetitions are completed.
  3. The robot reaches either target more than the specified number of times (excessive swings).

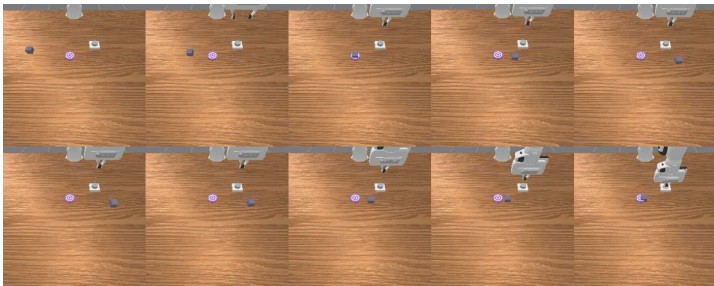

*Figure 15.* **StopCube Task Example.** In this instance, the goal is press the button to stop the cube exactly at the target on its second visit.

### E.4. StopCube

*Table 22. StopCube Task Configuration.*

| Difficulty | #Passes | Move Speed |
|---|---|---|
| Easy | 2–3 | slow/medium/fast |
| Medium | 3–4 | slow/medium/fast |
| Hard | 4–5 | slow/medium/fast |

The *StopCube* task, illustrated in Fig. 15, challenges the robot's temporal memory in a dynamic environment. The robot must monitor a cube that continuously oscillates across a target following a predefined line and press a button to stop *exactly* when the cube reaches the target at a specified ordinal occurrence (e.g., the third visit). Successful completion requires combining continuous visual tracking with discrete event counting to identify the correct occurrence and execute the stopping action with high temporal precision. Detailed Task Configurations are summarized in Table. 22.

**Language Goal.** The instruction specifies the ordinal constraint ($k$-th visit) for the stop condition. e.g., "press the button to stop the cube exactly at the target on its third visit."

**Task Dynamics.** The environment features a dynamic interaction phase defined by continuous movement:

1. **Oscillating Cube:** A cube is moving back and forth between start and end positions along a straight line.

2. **Static Behavior:** To succeed, the robot must position its end-effector over the button and remain static (hovering) until the correct timing.

3. **Immediate Stop:** Pressing the button stops the cube instantly. The robot must press at the exact moment the cube reaches the target zone in the specified cycle, accounting for the motion delay from hover to button.

**Success Criteria.**

• **Precise Synchronization:** The button must be pressed strictly within the time window when the cube overlaps with the target.

• **Correct Ordinality:** The stop must correspond to the specified count when the cube reaches the target.

• **Failure:** The episode terminates early if the button is pressed too early or too late.

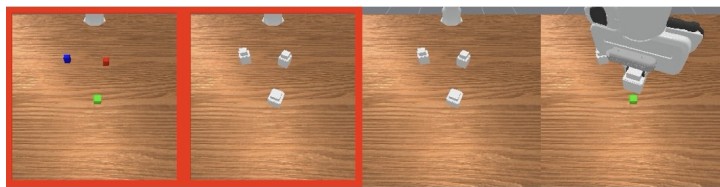

*Figure 16.* **VideoUnmask Task Example.** In this instance, after watching a video in which all cubes are masked, the robot must pick up the container hiding the green cube. Red-bordered frames denote the video-based observation prior to execution.

### E.5. VideoUnmask

*Table 23. VideoUnmask Task Configuration.*

| Difficulty | #Total Containers | #Goal Containers | #Total Cubes |
|---|---|---|---|
| Easy | 3 | 1 | 3 (red, green, blue) |
| Medium | 5 | 1 | 3 (red, green, blue) |
| Hard | 10 | 2 | 3 (red, green, blue) |

The *VideoUnmask* task, illustrated in Fig. 16, evaluates spatial memory from a demonstration video. The robot first watches the video and infers which container hides the specified cube (red, blue, or green), then selects the corresponding container(s) during execution. Successful completion requires identifying the correct container(s) from the video and picking each required one at least once (in the specified order when multiple picks are requested), without selecting any incorrect containers. Task configurations are summarized in Table 23.

**Language Goals.** We use language instructions with increasing temporal and compositional complexity:

1. Pick **one** container hiding a specified cube
   *e.g., "Watch the video carefully, then pick up the container hiding the green cube."*

2. Pick **two** containers sequentially (order matters)
   *e.g., "Watch the video carefully, then pick up the container hiding the green cube, and finally pick up the container hiding the red cube."*

**Task Characteristics.** Each episode consists of a **video phase** followed by an **execution phase**.

- **Video:** Multiple containers are placed on the table. Each cube (red/green/blue) is hidden under a container, and some containers may be empty. The robot remains stationary during the video.

- **Execution:** The robot must pick up the container(s) hiding the specified cube(s) described in the language instruction, using the correspondence inferred from the demonstration.

**Success Criteria.**

- **Correct completion:** All containers hiding the specified cubes are picked up at least once, following the required order when applicable.

- **Immediate failure:** Picking any incorrect container (i.e., one hiding a non-specified cube) immediately terminates the episode.

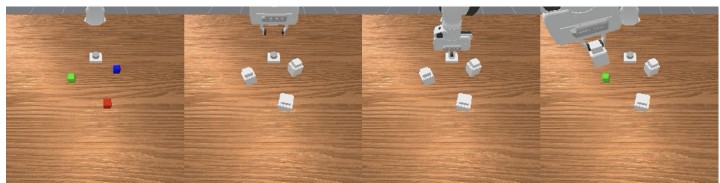

*Figure 17.* **ButtonUnmask Task Example.** In this instance, the robot first presses the button on the table and then selects the container hiding the green cube. During the button press, all cubes are masked.

### E.6. ButtonUnmask

*Table 24. ButtonUnmask Task Configuration.*

| Difficulty | #Total Containers | #Goal Containers | #Total Cubes |
|---|---|---|---|
| Easy | 3 | 1 | 3 (red, green, blue) |
| Medium | 5 | 1 | 3 (red, green, blue) |
| Hard | 15 | 2 | 3 (red, green, blue) |

The *ButtonUnmask* task, illustrated in Fig. 17, evaluates the robot's spatial memory for identifying hidden objects through active interaction. Unlike passive observation tasks, the robot must first press a button to reveal the scene, then determine which container hides the specified cube and pick it up. Task configurations are given in Table 24.

**Language Goals.** We use language instructions with increasing compositional complexity:

1. Pick **one** specified container by cube color
   *e.g., "First press the button on the table, then pick up the container hiding the green cube."*

2. Pick **two** specified containers sequentially (order matters)
   *e.g., "First press the button on the table, then pick up the container hiding the green cube, and finally pick up the container hiding the red cube."*

**Task Characteristics.**

- **Button Pressing:** The robot needs to press the button at the beginning. During the press, multiple containers are concurrently placed on the table. Each cube is hidden under a container, and some containers may be empty. The robot needs to memorize the event that occurred.

- **Unmasking:** The robot identifies and picks up the container(s) hiding the specified cube(s) described in the language instruction. When multiple pickups are required, the robot must release the first container (e.g., place it back on the table) before attempting the next.

**Success Criteria.**

- **Correct completion:** The robot successfully presses the button and picks up all specified containers at least once, following the required order.

- **Immediate failure:** The episode terminates early if:
  1. The robot picks up any container before completing the button-press phase.
  2. The robot picks up an incorrect container (i.e., one hiding a non-specified cube).

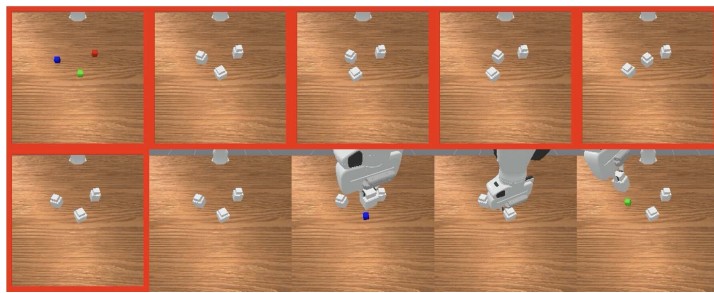

*Figure 18.* **VideoUnmaskSwap Task Example.** In this instance, the robot first watches the video, then picks up the container hiding the blue cube, followed by the one hiding the green cube. The video shows that all cubes are being masked by containers, with some containers swapped between locations. Red-bordered frames denote the video-based initial observations provided before execution.

## E.7. VideoUnmaskSwap

*Table 25. VideoUnmaskSwap Task Configuration.*

| Difficulty | #Total Containers | #Swap Times | #Goal Containers | #Total Cubes |
|---|---|---|---|---|
| Easy | 3 | 1–2 | 1–2 | 3 (red, green, blue) |
| Medium | 4 | 1–2 | 1 | 3 (red, green, blue) |
| Hard | 4 | 2–3 | 2 | 3 (red, green, blue) |

The *VideoUnmaskSwap* task (Fig. 18) evaluates spatial memory from a video demonstration. The robot observes containers swapping positions, tracks the correct containers, and then retrieves them during execution. Task configurations are summarized in Table 25.

**Language Goals.** We use language instructions with increasing temporal and compositional complexity:

1. Pick **one** specified container
   *e.g., "Watch the video carefully, then pick up the container hiding the blue cube."*

2. Pick **two** specified containers sequentially (order matters)
   *e.g., "Watch the video carefully, then pick up the container hiding the blue cube, and finally pick up the container hiding the red cube."*

**Task Characteristics.** Each episode consists of a **video phase** followed by an **execution phase**.

- **Video:** Multiple containers are placed on the table. Each cube (red/green/blue) is hidden under a container, and some containers may be empty. The robot remains stationary while the containers swap positions multiple times according to the difficulty setting.

- **Execution:** The robot must pick up the container(s) hiding the specified cube(s) described in the language instruction, tracking each container's identity across the observed swaps.

**Success Criteria.**

- **Correct completion:** All specified containers are picked up at least once, following the required order when applicable.

- **Immediate failure:** Picking any incorrect container (i.e., one hiding a non-specified cube) immediately terminates the episode.

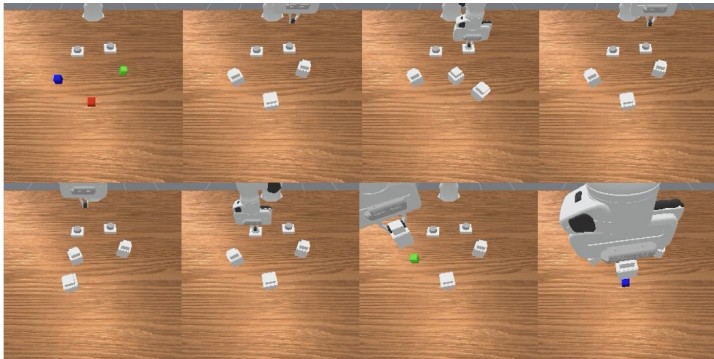

*Figure 19.* **ButtonUnmaskSwap Task Example.** In this instance, the robot first presses both buttons on the table, then picks up the container hiding the green cube, followed by the one hiding the blue cube. During the button presses, all cubes are masked and their locations are swapped.

### E.8. ButtonUnmaskSwap

*Table 26. ButtonUnmaskSwap Task Configuration.*

| Difficulty | #Total Containers | #Goal Containers | #Swap Times | #Total Cubes |
|---|---|---|---|---|
| Easy | 3 | 1–2 | 1–2 | 3 (red, green, blue) |
| Medium | 4 | 1 | 1–2 | 3 (red, green, blue) |
| Hard | 4 | 2 | 2–3 | 3 (red, green, blue) |

The *ButtonUnmaskSwap* task (Fig. 19) evaluates spatial memory under active interaction. The robot must press both buttons during container hiding and swapping, track the container concealing the target cube, and finally lift the correct one. Task configurations are summarized in Table 26.

**Language Goals.** We use language instructions with increasing complexity:

1. Pick **one** specified container
   *e.g., "First press both buttons on the table, then pick up the container hiding the red cube."*

2. Pick **two** specified containers sequentially (order matters)
   *e.g., "First press both buttons on the table, then pick up the container hiding the green cube, and finally pick up the container hiding the red cube."*

**Task Characteristics.**

- **Button Pressing:** The robot needs to press both buttons. During the pressing, multiple containers are placed on the table to enclose the cubes, after which some containers swap positions.

- **Unmasking:** The robot must pick up the container(s) hiding the specified cube color(s) described in the language instruction, following the same order when multiple pickups are required.

**Success Criteria.**

- **Correct completion:** The robot presses both buttons and picks up all specified containers at least once, following the required order.

- **Immediate failure:** The episode terminates early if:

  1. The robot picks up any container before completing the button-press phase.
  2. The robot picks up an incorrect container (i.e., one hiding a non-specified cube).

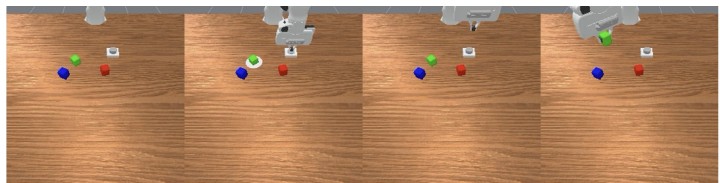

*Figure 20.* **PickHighlight Task Example.** In this instance, the goal is to first press the button, then pick up all cubes *highlighted by a white area*, finally press the button again to stop. During the button pressing, the green cube was highlighted with a white area on the table, the robot needs to capture this information, and pick up the correct cubes.

### E.9. PickHighlight

*Table 27. PickHighlight* Task Configuration.

| Difficulty | #Total Cubes | #Goal Cubes |
|---|---|---|
| Easy | 3 | 1 |
| Medium | 4 | 2 |
| Hard | 6 (cluster) | 3 |

The *PickHighlight* task, shown in Fig. 20, evaluates the robot's short-term object memory. The robot must press a button to briefly highlight specific cubes, memorize the highlighted cubes, and then pick them up after the cues disappear. This tests the robot's ability to maintain a working memory of transient object states. Detailed Task Configurations are summarized in Table 27.

**Language Goals.** The task involves a multi-stage sequential command: press the button, memorize, and pick highlighted objects.
*e.g., "first press the button, then pick up all highlighted cubes, finally press the button again to stop."*

**Task Characteristics.**

• **Button Pressing:** The robot must first press the button. During the press, the specified cubes are briefly highlighted with a white visual indicator for a fixed number of timesteps.

• **Highlighted Cube Picking:** The robot must identify and pick up all cubes highlighted during this window. If multiple cubes are specified, each must be picked up at least once.

**Success Criteria.**

• **Correct completion:** The robot successfully presses the button and picks up all correct cubes at least once.

• **Immediate failure:** The episode terminates early if:

  1. The robot fails to press the button before attempting a pick.
  2. The robot picks up any wrong cubes (i.e., a cube that was not highlighted).

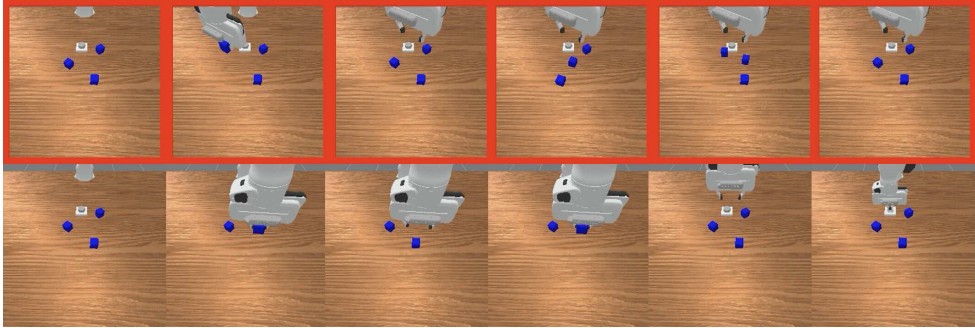

*Figure 21.* **VideoRepick Task Example.** In this instance, the robot first watches a video, then picks up the same previously picked cube twice, and finally presses the button to stop. Red-bordered frames denote the video-based observations shown prior to execution.

### E.10. VideoRepick

*Table 28.* VideoRepick Task Configuration.

| Difficulty | #Cubes | #Swap Times | #Repetition |
|---|---|---|---|
| Easy | 3 (same color) | 1–2 | 1–3 |
| Medium | 3 (same color) | 2–3 | 1–3 |
| Hard | 15 (different colors, clutter scene) | 0 | 1–3 |

The *VideoRepick* task is illustrated in Fig. 21. The robot must watch a demonstration video to identify a specific cube instance, locate the same instance in the current scene (even if the cubes have moved or swapped), and repeat the pick-and-place action exactly $N$ times before pressing a stop button. Task configurations are provided in Table 28.

**Language Goals.**

- Repeat picking the cube that was previously picked in the video for $N$ times, then stop.
  *e.g., "Watch the video carefully, then pick up the same cube that was previously picked twice, and finally press the button to stop."*

**Task Characteristics.** Each episode consists of a **video phase** and an **execution phase**.

- **Video:** the robot picks up a specified cube and places it. Afterward, the robot remains still while the cubes may swap positions, requiring the agent to track the correct cube from the video.

- **Execution:** the robot locates the same previously picked cube and repeats the pick-and-place action for a specific number of cycles specified in the instruction.

**Success Criteria.**

- **Correct completion:** complete exactly $N$ pick-and-place repetitions with the demonstrated cube, then press the button.

- **Immediate failure:** the episode terminates early if:

  1. the robot picks up the wrong cube,
  2. the button is pressed before finishing $N$ repetitions, or
  3. the robot completes more than $N$ repetitions.

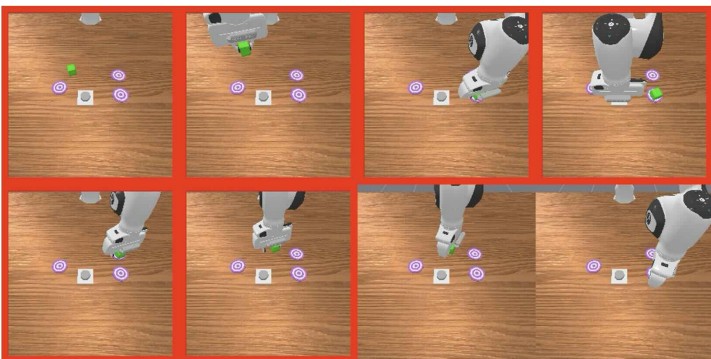

*Figure 22.* **VideoPlaceButton Task Example.** In this example, the robot first observes a video depicting an interleaved sequence of cube placements and button presses, and then places the cube onto the correct target corresponding to its location prior to the button press. Red-bordered frames indicate the video-based observations provided before execution.

### E.11. VideoPlaceButton

*Table 29. VideoPlaceButton Task Configuration.*

| Difficulty | #Cubes | #Targets | Target Swapping |
|------------|--------|----------|-----------------|
| Easy       | 1      | 3        | No              |
| Medium     | 3      | 4        | No              |
| Hard       | 3      | 4        | Yes             |

The *VideoPlaceButton* task, illustrated in Fig. 22, tests long-video temporal reasoning and object grounding. The robot must watch a long video in which a cube is placed on multiple targets, separated by a distinct button press, and then parse a natural language instruction specifying a temporal relation (e.g., *before* or *after* the button press) to identify the correct destination. Task configurations are summarized in Table 29.

**Language Goals.** We use language instruction to specify a cube and the target location relative to the button press event (*before/after*): *e.g., "Watch the video carefully and place the blue cube on the target where it was placed immediately before the button was pressed."*

**Task Characteristics.** Each episode has a **video phase** and an **execution phase**.

- **Video:** the robot picks up a cube and alternates between placing it on different targets and pressing a button. In the *Hard* mode, two targets swap positions after the sequence finishes.

- **Execution:** the robot must identify the correct cube, pick it up, and place it on the correct target according to the specified temporal relation (e.g., before or after the button press).

**Success Criteria.**

- **Correct completion:** successfully pick the correct color cube and place it on the correct target.

- **Immediate failure:** The episode terminates early if: (1) pick a wrong cube; (2) place the cube on the wrong target.

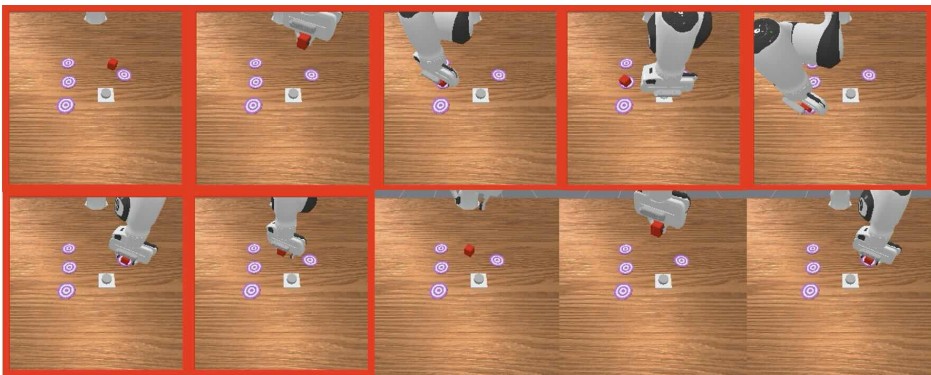

*Figure 23.* **VideoPlaceOrder Task Example.** In this instance, the goal is to place the red cube on the *third* target where it was previously placed in the given video. Red-bordered frames denote the video-based observation prior to execution.

### E.12. VideoPlaceOrder

*Table 30. VideoPlaceOrder Task Configuration.*

| Difficulty | #Cubes | #Targets | Target Swapping |
|---|---|---|---|
| Easy | 1 | 4 | No |
| Medium | 3 | 4 | No |
| Hard | 3 | 4 | Yes |

The *VideoPlaceOrder* task, illustrated in Fig. 23, evaluates long-video temporal reasoning and object grounding. The robot must watch a long video to memorize the sequence of cube placements, then parse a natural language instruction specifying an ordinal position (e.g., "the third target") to identify the correct destination. Task configurations are summarized in Table 30.

**Language Goals.** We use language instructions that specify the cube and the ordinal rank of the target location relative to the video history, *e.g., "Watch the video carefully and place the red cube on the third target where it was placed."*

**Task Characteristics.** Each episode consists of a **video phase** and an **execution phase**.

- **Video:** the robot picks up a cube and places it onto multiple targets in a specific sequence, pressing a button between placements. In the *Hard* mode, two targets swap positions after the sequence finishes.

- **Execution:** the robot identifies and picks up the correct cube, then places it onto the correct target corresponding to the requested ordinal rank (e.g., the third target visited).

**Success Criteria.**

- **Correct completion:** successfully pick up the correct colored cube and place it on the target indicated by the specified temporal order.

- **Immediate failure:** the episode terminates early if: (1) the robot picks up the wrong cube; or (2) the cube is placed on the wrong target.

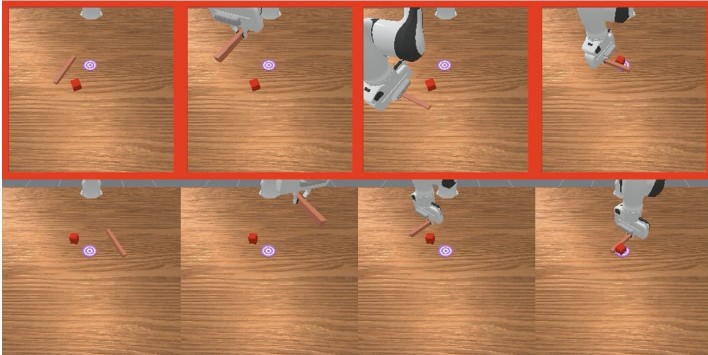

*Figure 24.* **MoveCube Task Example.** In this instance, the robot must watch the video carefully, then move the cube to the target in the same manner as demonstrated. Red-bordered frames denote the video-based observation prior to execution.

### E.13. MoveCube

The *MoveCube* task, illustrated in Fig. 24, assesses the robot's ability to recognize and reproduce distinct action semantics. The task requires the robot to observe a video demonstration where a cube is moved to a target using one of three specific manipulation manners: (1) hooking it with a tool; (2) pushing it with a closed gripper; or (3) performing a standard pick-and-place operation. Success requires correctly inferring the underlying motion strategy from the video and executing the exact same method during the execution phase.

**Language Goal.** *"Watch the video carefully, then move the cube to the target in the same manner as before."*

**Task Characteristics.** Each episode has a **video phase** and an **execution phase**.

- **Video:** The robot moves a cube to a target location using one of three manipulation methods:
  1. **Hooking:** pick up a peg tool and use it to hook the cube to the target.
  2. **Pushing:** close the gripper and use the gripper to push the cube across the surface to the target.
  3. **Pick-and-Place:** grasp the cube directly, lifting it, and placing it on the target.
- **Execution:** After the demonstration, the cube and target are reset to *new random positions*. The robot must identify the method used in the video demonstration and replicate it exactly, adapting it to the new configuration.

**Success Criteria.**

- **Correct completion:** The robot successfully moves the cube to the target using the same manipulation primitive observed in the demonstration.
- **Immediate failure:** The episode terminates immediately if the robot attempts a different method than the one demonstrated (e.g., picking up the cube when a push was demonstrated, or using the tool when a direct grasp was demonstrated).

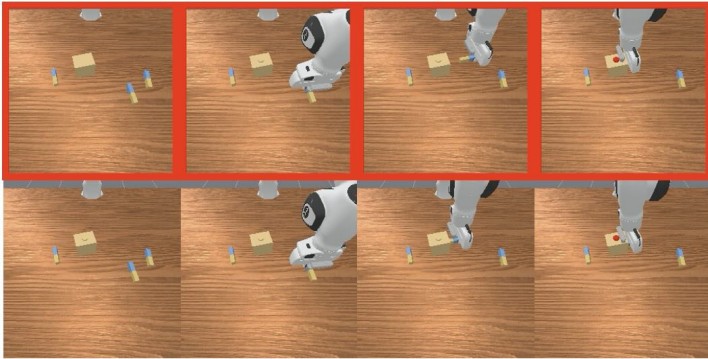

*Figure 25.* **InsertPeg Task Example.** In this instance, the robot must watch the video carefully, then grasp the same peg at the same end and insert it into the same side of the box as demonstrated. Red-bordered frames denote the video-based observation prior to execution.

### E.14. InsertPeg

The *InsertPeg* task, illustrated in Fig. 25, evaluates fine-grained visual perception and procedural/object memory. The robot observes a demonstration to determine which peg to select, which end to grasp, and which side to insert, then reproduces the same manipulation after the scene is reset to the same initial configuration as in the video.

**Language Goal.** *"Watch the video carefully, then grasp the same peg at the same end and insert it into the same side of the box as in the video."*

**Task Characteristics.** Each episode consists of a **video phase** and an **execution phase**.

- **Video:** The robot picks up a peg by grasping a specific end (head or tail) and inserts it into the box from a specific side (left or right). After the demonstration, all pegs are reset to their original positions and orientations without randomness.

- **Execution:** The robot must identify the correct peg based on video demonstrations, then grasp the correct peg at the same end and insert it into the box from the demonstrated side.

**Success Criteria.**

- **Correct completion:** The robot grasps the correct peg by the correct end and inserts it into the box from the correct side.

- **Immediate failure:** The episode terminates early if:

    1. The robot picks up an incorrect peg.
    2. The robot grasps the wrong end of the peg (e.g., grasping the head when the tail was demonstrated, or vice versa).
    3. The robot inserts the peg from the wrong side of the box (e.g., inserting from the left when the right side was demonstrated).

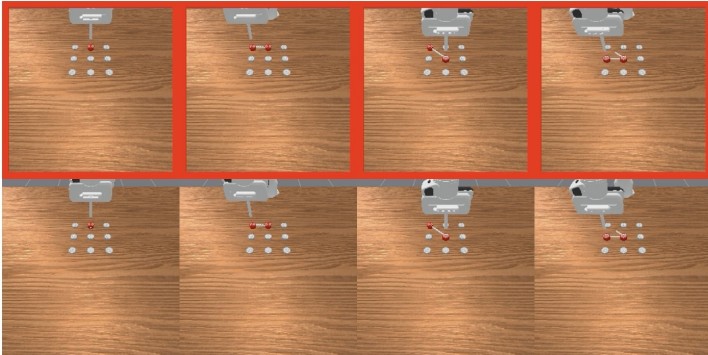

*Figure 26.* **PatternLock Task Example.** In this instance, the goal is to watch the video carefully and then use the stick attached to the robot to retrace the same pattern shown in the video. Red-bordered frames denote the video-based observations prior to execution.

### E.15. PatternLock

*Table 31. PatternLock* Task Configuration

| Difficulty | Grid Size | Pattern Length |
|---|---|---|
| Easy | 3×3 | 2–4 |
| Medium | 4×4 | 3–5 |
| Hard | 5×5 | 4–8 |

The *PatternLock* task, illustrated in Fig. 26, evaluates the robot's long-horizon procedural memory from a video demonstration. The robot observes a stick attached to the end-effector tracing a continuous pattern over an NxN grid of gray targets, and must then reproduce the same trajectory during execution. Success requires following the full demonstrated sequence without skipping or moving to incorrect targets.

**Language Goal.** *"Watch the video carefully, then use the stick attached to the robot to retrace the same pattern shown in the video."*

**Task Characteristics.**

Each episode consists of a **video phase** and an **execution phase**.

- **Demonstration:** The robot uses a stick to trace a continuous pattern over a target grid, defining an ordered sequence of targets.

- **Execution:** After reset, the robot must move the gripper to retrace the targets in the same order continually. The environment provides real-time feedback during tracing:
  - **Target highlighting feedback:** A gray target turns red when the stick is near it, indicating a valid visit.
  - **Trajectory trace feedback:** A line renders the stick-tip trajectory in real time.

**Success Criteria.**

- **Correct completion:** The robot visits all targets in the correct order following the demonstration.

- **Immediate failure:** the episode terminates early if the robot deviates from the demonstrated order.

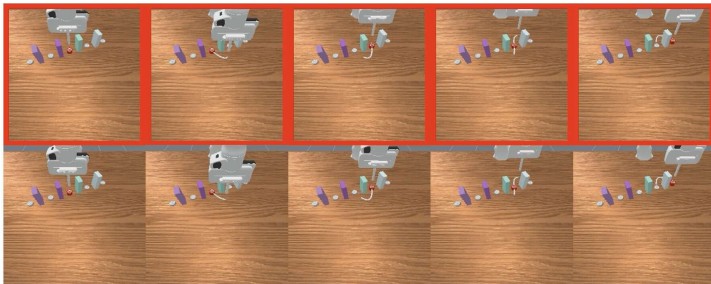

*Figure 27.* **RouteStick Task Example.** In this instance, the robot watches the video and uses the stick attached to the end-effector to navigate around obstacles on the table, reproducing the demonstrated path.

### E.16. RouteStick

*Table 32.* RouteStick Task Configuration

| Difficulty | Path Length | Backtracking |
| --- | --- | --- |
| Easy | 2–3 | No |
| Medium | 4–5 | No |
| Hard | 4–7 | Yes |

The *RouteStick* task, illustrated in Fig. 27, evaluates long-horizon, geometry-aware procedural memory from a video demonstration. The robot observes a demonstration in which a stick attached to the end-effector navigates through a field of fixed vertical obstacles while visiting a sequence of targets. Successful completion requires circling each obstacle in the demonstrated direction (clockwise or counterclockwise) and following a single continuous trajectory.

**Language Goal.** *"Watch the video carefully, then use the stick attached to the robot to navigate around the obstacles on the table, following the same path shown in the video."*

**Task Characteristics.**

Each episode consists of a **video phase** and an **execution phase**.

- **Video:** The robot navigates with a stick around fixed obstacles and visits targets in sequence, implicitly specifying clockwise or counterclockwise circling directions.

- **Execution:** After reset, the robot must visit the same targets in the same order and match the demonstrated circling directions continually. The environment provides real-time visual feedback during execution:
  - **Target highlighting feedback:** A target turns red when visited by the stick, indicating successful contact.
  - **Trajectory trace feedback:** A line continuously renders the stick-tip trajectory in real time.

**Success Criteria.**

- **Correct completion:** The robot visits all targets in the demonstrated order, matches each obstacle-circling direction (clockwise/counterclockwise), and maintains a continuous trajectory.

- **Immediate failure:** the episode terminates early if the robot deviates from the demonstrated order.

