# OpenReview forum: "RoboMME: Benchmarking and Understanding Memory for Robotic Generalist Policies"
_ICML.cc/2026/Conference — ICML 2026 spotlight_

### Official Review · Reviewer_GUzZ · 2026-03-10

**Soundness:** 3
**Presentation:** 2
**Significance:** 4
**Originality:** 3
**Overall Recommendation:** 5
**Confidence:** 4

**Summary:**

The paper proposed a new benchmark to evaluate VLA models in long-horizon tasks, and a systematic analysis on how to equip existing VLA models with memorization ability. The proposed benchmark is categorized into four different dimensions and overall 16 different tasks. In the statistical analysis, the paper examined three kinds of different memory representations and three interation mechanisms.

**Compliance With Llm Reviewing Policy:**

Affirmed.

**Final Justification:**

I raised my recommendation as the author fully addressed my concerns in the rebuttal.

**Key Questions For Authors:**

1: Can the current popular VLA models, such as $\pi_0.5$ [2] and OpenVLA [1], be tested on the proposed benchmark without modifying their network structures？ Most current VLA models follow these two structures.

2: Can a VLA model that is not designed with memory tokens be tested in the benchmark?

[1] Kim, Moo Jin, Chelsea Finn, and Percy Liang. "Fine-tuning vision-language-action models: Optimizing speed and success." [2] Intelligence, Physical, et al. ": a vision-language-action model with open-world generalization." 0.5$: a vision-language-action model with open-world generalization (2025).

**Limitations:**

Yes.

**Strengths And Weaknesses:**

### Strengths:
1. **Well motivated and significant.**  Memory is a crucial problem in robotic learning. Currently, the robotics learning field still lacks high-quality, widely accepted, and long-horizontal benchmarks for fair and reproducible evaluation. Providing this kind of benchmark will indeed benefit the robot society.

2. **Sound and inspiring.** The categorization of the proposed benchmark into four dimensions is insightful, comprehensive, and looks plausible.

3. **Good visualization.**  The visualized images are quite supportive and easy to follow.

### Major Weakness:
1. **Compatbility.** In the dimension of Spatial Memory and Procedural Memory, the robot system is required to watch a video first. This design is incompatible with most current VLA models and might require extra modules beyond their original structure.  This may be a bottleneck to the benchmark's widespread adoption. A good benchmark design should free authors from the complexities of adaptation, enabling them to concentrate fully on methodology innovations instead of benchmark interfaces. It might be better to make the benchmark compatible with most VLA structures, even if the network is not specifically designed for memory tasks or without memory tokens.


### Minor Weakness:
1.  **Presentation.** I'd highly suggest not using the Question-Answering style in the Analysis section, which is hard to follow.

---

> ### Author Rebuttal · Authors · 2026-03-30
>
> We sincerely thank the reviewer for their thoughtful and constructive feedback, and for recognizing the strengths of our work. Below we address each concern.
>
> **W1: Compatibility. (Video-conditioned tasks require observing a video, which may be incompatible with many existing VLA models and hinder adoption.)**
>
> Thank you for raising this point. The video-conditioned setup is a deliberate design choice in RoboMME to better reflect realistic scenarios, where robots must observe (e.g., human demonstrations or other agents) before acting. This naturally introduces explicit memory requirements that are often overlooked in existing VLA models and benchmarks.
>
> However, RoboMME remains compatible with existing VLA models in three ways:
> 1. *Symbolic memory*: Models can leverage language subgoals by appending them to the task input without modifying the VLA backbone. The VLM processes the video and generates subgoals, while the VLA predicts actions. All our symbolic-memory MME-VLA variants follow this design.
> 2. *Memory-as-context integration*: Models can append sampled video frames to the input, requiring only modest changes to the input sequence (i.e., additional image tokens). Our `FrameSamp/TokenDrop+Context` variants follow this setup.
> 3. *No history*: Models can be trained using only current observations, without any memory adaptation. We include both a $\pi_{0.5}$ baseline and a Diffusion Policy baseline under this setting.
>
> Overall, RoboMME is *backward-compatible* with existing VLA designs, while encouraging future models to explicitly reason over both passive and active history.
>
>
> **W2: Presentation. (Q&A style in the Analysis section is hard to follow.)**
>
> Thank you for the helpful suggestion. We will reorganize the experiment section by first clearly stating the research questions, followed by a more natural and coherent analysis of the results.
>
> **Q1: Can popular VLA models (e.g., $\pi_{0.5}$, OpenVLA) be evaluated without modifying their architectures?**
>
> Yes. RoboMME is already built on the $\pi_{0.5}$ backbone, and both symbolic-memory MME-VLA variants and the no-memory baseline require *no architectural changes*. More generally, history can be incorporated purely through inputs (e.g., subgoals or additional frames), without any new module design. We are currently extending experiments to OpenVLA and will include results in the revision.
>
> **Q2: Can existing VLA models without memory tokens be tested?**
>
> Yes. As shown in Table 3, the $\pi_{0.5}$ baseline, without memory tokens, achieves an 18% success rate on RoboMME. Our symbolic-memory MME-VLA variants also avoid using neural memory tokens and achieve up to a 33% success rate.
>
> However, with memory-as-modulator integration, perceptual-memory variants achieve up to a 44.5% success rate. These results highlight a key limitation: *most existing VLA architectures rely solely on current observations without explicit memory*. RoboMME exposes this gap and motivates future VLA models to handle long-horizon, history-dependent tasks with improved memory design.

---

> > ### Author Rebuttal · Reviewer_GUzZ · 2026-04-03
> >
> > Thank you for the authors’ response. The authors have addressed my previous concerns, and I maintain a positive recommendation for the paper.
> >
> > In addition, I found the question raised by Reviewer rkon to be insightful and helpful. I look forward to the authors’ further discussion. From my side, I do not have additional concerns.

---

### Official Review · Reviewer_rkon · 2026-03-12

**Soundness:** 3
**Presentation:** 4
**Significance:** 3
**Originality:** 3
**Overall Recommendation:** 4
**Confidence:** 5

**Summary:**

This paper presents the RoboMME benchmark for studying memory capabilities in tabletop manipulation tasks using VLA policies. The benchmark includes 16 tasks of varying complexity, divided into four categories defined by the authors: temporal memory, spatial memory, object memory, and procedural memory. These tasks were used to test the baseline pi0.5 memoryless model, the SAM2Act+ and MemER memory models, and 14 memory-augmented VLA variants (the MME-VLA suite) were proposed. The proposed memory-augmented VLAs cover three methods of forming memory representations: symbolic memory, perceptual memory, and recurrent memory, as well as three methods of working with these memory representations: memory-as-context, memory-as-modulator, and memory-as-expert. Experiments showed that recurrent baselines perform the worst, perceptual+modulator memory performs the best, and no single memory representation consistently dominates across all tasks.

**Compliance With Llm Reviewing Policy:**

Affirmed.

**Final Justification:**

I still have concerns about the RMT and TTT baselines, so I'm leaving my score unchanged, as it's already positive.

**Key Questions For Authors:**

Overall, I really enjoyed the paper, as it addresses an important and relevant problems in memory research in the VLA, which I believe is currently one of the central challenges. However, I have a number of questions about the data, training and validation, and ablations. I am open to further discussion and am willing to revise my assessment if the authors provide satisfactory clarifications and adequately respond to the above comments and questions.

**Limitations:**

Yes

**Strengths And Weaknesses:**

**Strengths:**

1. Interesting new video-conditioned tasks

2. A large number of tasks of varying complexity with good documentation and demonstrations, a taxonomy of memory problems inspired by neuroscience, and an extensive study of baselines from the MME-VLA suite

3. A subset of procedural memory tasks is considered, which distinguishes RoboMME from other benchmarks, which typically focus on declarative memory


**Weaknesses and Questions:**

1. The data collection process isn't fully explained: how exactly were keyframe waypoints selected? How was the data collected, and how noisy is it? Is this data similar to data obtained through teleoperation, or it is more synthetic? How exactly were perturbations introduced to train models to recover normal trajectories?

2. How exactly did you train your data? In a multi-task mode on a dataset containing trajectories from all 16 tasks, followed by evaluation for each task separately?

3. According to the text, the recurrent baselines models did not exploit the temporal structure of the trajectories. Instead, RMT and TTT were used to process a sequence of visual tokens embeddings for observations from the current timestep $o_t$, which may explain why they performed the worst among MME-VLA suite

4. How exactly did training across dataset trajectories performed? In standard VLA training, we select a batch of random timesteps and train to predict a chunk of future actions of size $H$ for each timestep $t$. During evaluation, we perform forward once each chunk size $H$ timesteps. In the case of MME-VLA, how do training and validation performed? As far as I understand, training should be performed on an episodic dataset containing time-ordered trajectories in a batch, but this would require processing up to 1000 timesteps sequentially for fairly highly correlated data, which could lead to spurious correlations.

5. How do you use action chunking during evaluation? What happens if the start and end of an event occur within the same chunk?

6. The paper is seriously lacking ablations for MME-VLA. For example, how much does performance depend on the strength of the underlying model (what happens if you replace pi0.5 with OpenVLA, for example)? If I understand correctly, the memory size $M=512$ was chosen because the VLA processes two images with 256 tokens each. How will the results change if you decrease or increase the $M$ value? In the case of gating, did you look at what percentage of the data is used from memory and what percentage is based on current observations?

7. What specific proprioceptive state is being used? I didn't fully understand its dimensions and components from the text.

---

> ### Author Rebuttal · Authors · 2026-03-30
>
> We sincerely thank the reviewer for the thoughtful feedback and insightful questions. Below we address each concern:
>
> **Q1: Details on data collection.**
>
> Our pipeline is fully automated:
> - *Keyframe waypoints*: We define a sequence of waypoint poses (e.g., pre-grasp, grasp, pre-place, place) using privileged object states for each task (similar to RLBench).
> - *Trajectory generation*: An RRT-based planner executes waypoints to generate synthetic trajectories. Failed runs are filtered. No human teleoperation is needed.
> - *Failure-recovery augmentation*: 5% of episodes include perturbed *grasp* poses (e.g., variations in grasp height or gripper orientation), followed by recovery using the original pre-grasp and grasp waypoints.
>
> **Q2: Is it multi-task learning?**
>
> Yes. A single model is jointly trained and evaluated across all 16 tasks.
>
> **Q3: Does recurrent memory exploit temporal structure?**
>
> Yes. We use Multi-modal Rotary Position Embedding (M-RoPE) to encode the position of each visual token. These positional embeddings are fused via an MLP to form *position-aware visual features*, which are then fed into the recurrent modules. We will clarify this in the final version.
>
> **Q4: How is training performed?**
>
> We train on ⟨$O_{1:t-1}, O_t, G, A_{t:t+C}$⟩, where $O_t$ denotes the current observation, $O_{1:t-1}$ denotes the history observations. $G$ is the task goal and $C$ is the action chunk size.
> - For *Symbolic memory*: VLM predicts the current subgoal from ⟨`previous_subgoal`, $O_t, G$⟩ (optionally with video). The VLA then conditions on ⟨`current_subgoal`, $O_t, G$⟩ to predict $A_{t:t+C}$.
> - For *Perceptual memory*: We apply frame sampling or token dropping to extract visual tokens from history, denoted as $M_t$. The VLA conditions on ⟨$M_t, O_t, G$⟩ to predict $A_{t:t+C}$.
> - For *Recurrent memory*: We sample *up to 64 frames* from history and process them recurrently to update the hidden states, producing memory tokens. The VLA prediction follows the same formulation as perceptual memory.
>
> We do not process the full history, which may help mitigate spurious correlation.
>
> **Q5: Details on action chunking and subgoal transition handling.**
>
> Thank you for the question. We train with a 20-step horizon and execute the first 16 steps during evaluation. A subgoal may complete within a chunk, so later actions could correspond to the next subgoal.
>
> We follow $\pi_{0.5}$, using the model to predict the full chunk conditioned on the current subgoal. In practice, this works well as long as subgoals are updated promptly, and small mismatches do not cause severe issues.
>
> **Q6(a): Performance with increasing memory**
>
> We use `FrameSamp/TokenDrop+Modul` as case study and find that performance improves with increasing budget.
>
> | Memory Tokens | FrameSamp+Modul | TokenDrop+Modul |
> |--|--|--|
> | 64 | 30.42  | 18.11    |
> | 128 | 36.54  | 32.76    |
> | 256 | 42.90| 36.36  |
> | 512 | 44.51  | 38.02   |
> | 1024 | 45.87 | 40.17   |
>
> **Q6(b): Percentage of memory vs. observation.**
>
> Thanks for raising this point. We conduct a small-scale evaluation (10 episodes per task, 160 total) using perceptual memory as a case study.
>
> For memory-as-context and memory-as-expert, we quantify attention allocation by measuring the average probability mass assigned to memory, observation, and language tokens across all action tokens, reporting both `mean / max` values (\%) aggregated across all layers.
>
> | Method | Memory | Observation | Language |  Success |
> |--|:--:|:--:|:--:|:--:|
> | `FrameSamp+Context` | 1.5 / 8.9   | **60.2** / **84.9**  | 38.3 / 63.9 |  30.68 |
> | `TokenDrop+Context`   | 4.8 / 17.0  | 45.4 / 72.9  | 49.2 / 94.8 |  34.50 |
> | `TokenDrop+Expert`     | 0.7 / 12.7  |  47.5 / 71.9  |  51.8 / **97.8**  | 34.86 |
> | `FrameSamp+Expert`   | 1.3 / **21.8** | 46.0 / 68.2  |  **52.7** / 96.4 |  36.25 |
>
> For memory-as-modulator, we analyze the distribution of predicted scale and bias values over all feature dimensions, and report the `mean`$\pm$ `std`.
>
> | Method | Scale | Bias | Success |
> |--|:--:|:--:|:--:|
> | `TokenDrop+Modul` | 5.99 $\pm$ 38.67  | 0.024  $\pm$  14.19  | 38.04 |
> | `FrameSamp+Modul` | 3.74 $\pm$ 22.55  | 0.012  $\pm$  9.45 | 44.51  |
>
> We find that:
> 1. Memory receives less attention, as it is mainly needed at key decision points, while most actions rely on current observations.
> 2. Higher attention to memory and language correlates with better performance.
> 3. Modulation directly affects action features, but excessive variance may hurt performance.
>
> **Q6(c): Other VLA backbones.**
>
> Thank you for the suggestion. We are training OpenVLA and will include results in the final version. We will also release the benchmark and invite the community to contribute additional backbones.
>
> **Q7: What proprioceptive state is used?**
>
> Thank you for the thoughtful question. We do not use proprioceptive inputs, as they did not yield significant gains. This is also aligned with the $\pi_{0.5}$ config on LIBERO in their official codebase.

---

> > ### Author Rebuttal · Reviewer_rkon · 2026-04-02
> >
> > I thank the authors for their detailed response.
> >
> > The answers to all questions except Q3, Q4, and Q5 are satisfactory.
> >
> > Regarding these three questions, it's still unclear to me how recurrent training is performed and how inference is implemented. For example, what is the TBPTT length? What happens if an inference event (for example, a cube appearing and disappearing) occurs and completes within an action chunk when the model receives no new information from the environment? How exactly is VLA trained (which is initially designed to keep only the current $o_t$ in context) based on, say, 64 previous observations (how do they all fit into the context)? And so on. I expect a more detailed answer from the authors regarding recurrent models (RMT and TTT), since this information is not clearly described in the text, and I have concerns about the correctness of the training and evaluation of these baselines.
> >
> > Once again, I thank the authors for their response and look forward to further discussion.

---

> > > ### Author Response · Authors · 2026-04-03
> > >
> > > Thank you for the follow-up questions. We provide detailed clarifications about the recurrent memory design below.
> > >
> > > **Q3(a) How we sample history frames for recurrent memory.**
> > >
> > > Only front-view images are used for memory. Given the current timestep $t$ and execution start index $t_{\text{exec}}$ (i.e., when robot execution begins after the video):
> > >
> > > - *Non-video-conditioned tasks* ($t_{\text{exec}} = 0$): We sample one frame every $K=16$ steps from the episode start to $t$, using offset $t \bmod K$. For example, at $t=200$, indices are $\{8, 24, 40, \dots, 200\}$. Different offsets serve as data augmentation.
> > >
> > > - *Video-conditioned tasks* ($t_{\text{exec}} > 0$): We first sample the video portion $[0, t_{\text{exec}})$ using stride $K$ (or uniformly up to 40 frames if the video is too long). For the execution portion $[t_{\text{exec}}, t]$, we use the same stride-based rule as above, then concatenate both index lists.
> > >
> > > We keep the most recent **64** indices to represent the history, where the last index is always the current timestep $t$. This is enough to cover most episodes in our training data.
> > >
> > > If total indices are fewer than 64, we left-pad with zeros and apply a mask so that padded positions *do not affect* the recurrent hidden state.
> > >
> > > **Q3(b) How these 64 frames fit into the context.**
> > >
> > > The 64 history frames do *not* enter the VLA's language-vision context directly. Instead, they are first compressed by RMT/TTT into 512 memory tokens, which are then integrated into the VLA backbone as described below.
> > >
> > > - *Feature Extraction*: We extract pooled SigLIP visual features (8$\times$8 patches per frame), concatenate them with M-RoPE positional embeddings (globally unique in full history), and project to hidden size $H=512$, yielding $F \in \mathbb{R}^{T \times S \times H}$ with $T=64$ timesteps and $S=64$ patches. The recurrent module processes $F$ sequentially along the temporal axis.
> > >
> > > - *RMT*: The hidden state is a set of 512 learnable queries $M \in \mathbb{R}^{N \times H}$ ($N=512$). At each step, $M$ is updated via grouped-query cross-attention over $F_i \in \mathbb{R}^{S \times H}$. After $T$ steps, the final $M$ serves as the memory tokens.
> > >
> > > - *TTT*: The hidden state consists of fast weights $(W, b)$ of a linear MLP, initialized from learned parameters. We flatten $F$ into $F' \in \mathbb{R}^{TS \times H}$. At each step, TTT processes a chunk of $S$ tokens from $F'$, updates the fast weights, and outputs $S$ embeddings of dimension $H$. After $T$ steps, TTT produces an output sequence $O \in \mathbb{R}^{TS \times H}$, from which we take the last 8 steps (512 tokens) as the memory tokens.
> > >
> > > - *Integration*: Both methods produce 512 memory tokens, along with a validity mask, which are then projected to suitable dimensions for integration into $\pi_{0.5}$ via memory-as-context, memory-as-modulator or memory-as-expert mechanism.  The $\pi_{0.5}$ backbone still processes the current observation (512 image tokens) and language tokens as in the original model, but memory tokens will be incorporated. More details are provided in Appendix A.3.
> > >
> > > **Q4. How is training and inference performed?**
> > >
> > > During training, each sample is a *randomly* selected timestep from a *randomly* selected episode, ensuring independent samples with uncorrelated histories (i.e., we do not reuse one episode to construct temporally dependent samples). The history is encoded via recurrent computation into 512 memory tokens. The TBPTT length is *64*. Gradients flow through all 64 steps, but supervision comes from last steps (1 for RMT or 8 for TTT). We *do not* backpropagate through full trajectories (e.g., 1000 steps).
> > >
> > > During evaluation, we maintain a cumulative history buffer. For video-conditioned tasks, video frames are pre-sampled with the same rule as in training. During execution, we add the initial frame, then append one frame every $K=16$ steps (last frame of each chunk), keeping the most recent 64 frames and processing them consistently with training.
> > >
> > > **Q5: Events within an action chunk.**
> > >
> > > Our execution action chunk size is 16 steps. In RoboMME, all salient events (e.g., block highlighting, object masking) persist for at least two chunks (32 steps), so no event starts and completes within a single chunk.
> > >
> > > We acknowledge that limited reactivity within an action chunk is an inherent drawback of chunk-based VLA models such as $\pi_{0.5}$, which can reduce responsiveness in highly dynamic scenarios. Recent work such as Real-Time Chunking [1] and Bidirectional Decoding [2] have begun to address this limitation. We will further discuss this limitation in the revision and explore improved solutions in future work.
> > >
> > > [1] Black, Kevin, et al. "Real-time execution of action chunking flow policies." arXiv:2506.07339 (2025).
> > > [2] Liu, Yuejiang, et al. "Bidirectional decoding: Improving action chunking via guided test-time sampling."  ICLR 2025.
> > >
> > > ---
> > > We will include these details in the revision and are happy to clarify further if needed.

---

### Official Review · Reviewer_sC6R · 2026-03-13

**Soundness:** 3
**Presentation:** 4
**Significance:** 4
**Originality:** 3
**Overall Recommendation:** 5
**Confidence:** 5

**Summary:**

The paper introduces RoboMME, a comprehensive benchmark designed to evaluate memory-augmented Vision-Language-Action (VLA) policies. The benchmark categorizes memory into four dimensions: temporal, spatial, object, and procedural, comprising 16 long-horizon tabletop tasks. Alongside the benchmark, the authors systematically study 14 VLA variants built upon the  $\pi_{0.5}$ backbone, exploring three memory representations (symbolic, perceptual, recurrent) and three integration mechanisms (context, modulator, expert). The empirical results demonstrate that memory effectiveness is highly task-dependent, with perceptual memory (specifically FrameSamp + Modulator) offering the best balance of performance and computational efficiency, while symbolic memory excels in explicit counting tasks.

**Compliance With Llm Reviewing Policy:**

Affirmed.

**Final Justification:**

This work is valuable and well-written. I keep my original score and recommend it to be accepted.

**Key Questions For Authors:**

i) According to the experimental setup, the memory budget is capped at 512 tokens. Given that the average episode length for historical frames or reference videos in RoboMME spans several hundred steps, I am concerned that 512 tokens are insufficient to retain adequate historical information, particularly when utilizing perceptual memory. Would scaling to a longer memory context yield better performance?

ii) As shown in Table 3, within the Symbolic Memory setting, substituting the oracle textual memory with Gemini or QwenVL results in a significant drop in performance. This is somewhat perplexing. Considering that Gemini is a highly capable multimodal foundation model, is this severe degradation primarily due to the VLM's inability to generate stable and accurate textual subgoals, or is it because the underlying VLA lacks the robustness to understand and generalize from these generated subgoals?

**Limitations:**

yes

**Strengths And Weaknesses:**

Strengths:

i) This paper introduces a dedicated benchmark to evaluate the memory capabilities of Vision-Language-Action (VLA) models, effectively bridging a long-standing gap in the field. Compared to existing benchmarks, it features a more comprehensive design and offers greater practical usability.

ii) The authors conduct a comprehensive exploration of various memory integration strategies, using rigorous experiments to validate the impact of different memory paradigms across diverse task types. This analysis compellingly highlights the current deficiencies in the design and exploration of memory mechanisms within existing VLA architectures.

iii) Including human oracle performance and state-of-the-art baselines like MemER and SAM2Act+ effectively contextualizes the difficulty of the benchmark.

Weaknesses:
Please refer to the Section of "Key Questions For Authors".

---

> ### Author Rebuttal · Authors · 2026-03-30
>
> We sincerely thank the reviewer for their thoughtful and constructive feedback, and appreciate the reviewer's acknowledgment of the strengths of our paper. We provide detailed responses to each of their questions as follows.
>
> **Q1: Would scaling to a longer memory context improve performance?**
>
> Yes. Our results show that increasing memory context (e.g., from 64 to 1024 tokens) consistently improves performance across methods. In our experiments, `FrameSamp+Modul` uses 16 pooled tokens per image to retain more frames while improving training efficiency, whereas `TokenDrop+Modul` uses 64 pooled tokens per image because token dropping requires higher spatial resolution to operate effectively. In principle, using the full 256 tokens per image and scaling to longer memory contexts could yield additional gains, which we leave for future investigation.
>
> | #Memory Token | FrameSamp+Modul | TokenDrop+Modul |
> |--|--|--|
> | 64 | 30.42  | 18.11    |
> | 128 | 36.54  | 32.76    |
> | 256 | 42.90| 36.36  |
> | 512 | 44.51  | 38.02   |
> | 1024 | 45.87 | 40.17   |
>
> In the paper, we fix the memory budget to 512 tokens in order to match the token count of current observations (2x16x16), ensuring a controlled comparison across methods.  Our goal is not to maximize absolute performance, but to provide a standardized testbed for researchers to study memory mechanisms. We will release all code to facilitate further exploration of larger memory settings.
>
> **Q2: Is the performance degradation mainly due to unstable subgoal generation (VLM) or the VLA’s inability to utilize them?**
>
> Thank you for the insightful question. Our analysis indicates that the degradation primarily arises from the VLM (subgoal generation) rather than the VLA robustness.
>
> Although Gemini is a strong general-purpose multimodal model, we observe that it struggles to produce consistent and accurately grounded subgoals in the RoboMME setting, likely due to domain shift (e.g., simulator visuals and task structure). This is consistent with prior work’s findings (e.g., MemER), where a domain-adapted fine-tuned Qwen2.5-VL significantly outperforms Gemini-ER.
> Additional simulation rollout failures of `GroundSG+Gemini` are available on our anonymous website https://anonymtest1.github.io.
>
> Empirically, Table 3 shows that oracle subgoals achieve around 84% success, substantially higher than all symbolic-memory MME-VLA variants (≤33%). This large gap (~50%) indicates that most of the performance degradation stems from inaccurate subgoal generation, rather than limitations of the VLA in executing them.
>
> Table 8 provides further evidence. In that experiment, Gemini is prompted to select subgoals and an oracle planner is used for execution, isolating high-level task planning with low-level control. Under this setting, humans achieve about 90% success, whereas Gemini reaches only 48%. This further suggests that subgoal generation is the core challenge for VLMs.

---

> > ### Author Rebuttal · Reviewer_sC6R · 2026-04-03
> >
> > Thank you for the detailed responses.
> > My concerns about the effect of memery length and reason of performance degrad induced by VLM-based symbolic memory are well addressed.
> > I'll keep my score.

---

### Official Review · Reviewer_DqpL · 2026-03-13

**Soundness:** 4
**Presentation:** 4
**Significance:** 4
**Originality:** 3
**Overall Recommendation:** 5
**Confidence:** 4

**Summary:**

This paper introduces RoboMME, a large-scale, standardized benchmark specifically designed to evaluate and advance the capabilities of memory-augmented Vision-Language-Action (VLA) models in long-horizon, history-dependent robotic manipulation tasks. The authors address a critical gap in current robotic learning research, where most evaluations are confined to short-horizon, Markovian settings. Drawing inspiration from cognitive science, the benchmark categorizes memory into four distinct types: temporal, spatial, object, and procedural, encompassing 16 diverse manipulation tasks, 1,600 demonstrations, and over 773k high-quality timesteps.

Furthermore, the authors systematically implement and evaluate 14 VLA variants with memory augmentation built upon a unified $\pi_{0.5}$ backbone. They comprehensively explore various memory representations (symbolic, perceptual, recurrent) and integration strategies (e.g., memory-as-input, memory-as-modulator). The empirical results reveal that the efficacy of different memory designs is highly task-dependent, with no single approach dominating across all scenarios. Ultimately, the study concludes that combining perceptual memory with a "memory-as-modulator" integration strategy achieves the most favorable balance between task performance and computational efficiency.

**Compliance With Llm Reviewing Policy:**

Affirmed.

**Key Questions For Authors:**

1. **Regarding Simulator Selection and Sim-to-Real Gap:** Given that the current benchmark relies heavily on the modified ManiSkill environment, how do the authors assess the potential sim-to-real gap when tasks involve higher-fidelity physical interactions? Are there plans to incorporate simulators such as RoboTwin in the future to support the validation of bimanual long-horizon manipulation tasks?

2. **Regarding Error Accumulation in Real-World Environments:** In practical deployment, long-horizon tasks inevitably encounter accumulated perceptual drift and control errors. From the real-world experiments, how many of the primary failure cases were attributed to these physical errors "corrupting" the memory representations? How should anti-noise or error-correction mechanisms be designed within memory modules in the future?

3. **Regarding Benchmark Task Design Logic and Scalability:** Beyond the classification based on the four cognitive science memory types, what is the core design logic behind selecting these specific 16 manipulation tasks? Do the authors consider the current task set to be comprehensive enough? Is there room for further expansion or improvement in task coverage (e.g., considering long-horizon tasks with more dynamic distractors, complex multi-object deformations, or multi-agent interactions)?

4. **Discussion on the Optimal Memory Mechanism:** The paper explicitly states: "No single memory representation consistently dominates across all tasks. Instead, their strengths are highly task-dependent and complementary." Based on this significant finding, what do the authors envision as a truly "optimal" or "ideal" memory mechanism? It is strongly recommended that the authors add a corresponding section in the Discussion to explore how to construct an ideal memory architecture capable of dynamic scheduling and unified integration of these complementary advantages.

**Limitations:**

Yes. The authors appropriately discuss the current limitations in the conclusion, noting the focus on tabletop manipulation and the use of a single pre-trained backbone. While the aforementioned issues regarding dual-arm expansion and robustness against real-world cascading errors are objective limitations of the current scope, they do not diminish the core value of this work as a critical benchmark for the field.

**Strengths And Weaknesses:**

**Strengths**:

- **Significance & Originality**: This work holds exceptional forward-looking and academic value. Long-horizon tasks are currently the most critical bottleneck for embodied AI and robotic manipulation to achieve general-purpose utility, and memory mechanisms are the foundational capability required to solve these history-dependent tasks. This paper excellently constructs the first standardized benchmark specifically for long-horizon tasks. Furthermore, it provides a highly systematic survey and analysis of existing memory-based VLA approaches. This not only clarifies the current technological landscape but also offers crucial insights for future memory-based VLA architecture design, making it a foundational piece of literature for the community.
- **Soundness**: The experimental design is exceptionally rigorous and extensive. By unifying 14 different memory-augmented variants under the same $\pi_{0.5}$ backbone, the authors effectively eliminate confounding variables that often plague benchmark comparisons (such as differences in pre-training data or visual encoders). The empirical claims are strongly supported by a massive dataset (773k timesteps).
- **Presentation**: The paper is exceedingly well-written and logically structured. The complex architectural differences between various memory representations and integration mechanisms are made highly accessible through intuitive diagrams and clear mathematical formulations. The explicit categorization of memory types greatly aids the reader in understanding the specific challenges posed by each task suite.

**Weaknesses**:

- **Simulator Fidelity & Sim-to-Real Gap:** The benchmark currently relies heavily on modifications to the ManiSkill simulator. However, there remains uncertainty about whether this simulator can guarantee a smooth, high-fidelity sim-to-real transfer when handling highly complex physical contacts and friction. To further establish the benchmark's authority in complex physical interactions, it is highly recommended that future iterations incorporate simulators with higher-fidelity physical rendering, such as RoboTwin.

- **Scope of the Benchmark:** Currently, RoboMME is heavily focused on single-arm, tabletop manipulation tasks with a relatively fixed set of assets. While this is a reasonable and necessary starting point for standardizing memory evaluation, it limits the immediate applicability of the findings. The benchmark would be significantly strengthened and diversified by explicitly including bimanual (dual-arm) manipulation settings. Furthermore, expanding the scope to include mobile manipulation or open-world navigation—where spatial memory operates on a substantially larger scale in dynamic, unstructured environments—would greatly enhance its long-term impact.

- **Analysis of Recurrent Memory Failures:** The experimental results indicate that recurrent memory approaches yielded the poorest performance, which the authors attribute to training instability caused by fine-tuning shallow recurrent layers on the $\pi_{0.5}$ backbone. While objectively reporting negative results is commendable, the paper would benefit from a deeper theoretical analysis of *why* this instability occurs. Providing preliminary ablations (e.g., testing deeper recurrent layers, alternative learning rate schedules, or different initialization strategies) would help clarify whether this is an inherent architectural limitation or a surmountable training issue.

---

> ### Author Rebuttal · Authors · 2026-03-30
>
> We sincerely thank the reviewer for recognizing the novelty and significance of our work. We appreciate all the thoughtful feedback and suggestions.
>
> **W1&Q1: ManiSkill may increase the sim-to-real gap.**
> Thank you for raising this important point. Our current focus is not on sim-to-real transfer, but on building a *unified benchmark to systematically study memory* in robotic manipulation. Therefore, we choose ManiSkill for its ease of modification and deployment, and its sufficient support for our designed memory-intensive tasks.
>
> We agree that higher-fidelity simulators would further strengthen the benchmark, especially for modeling complex physical interactions such as contact and friction. We are actively exploring alternative platforms (e.g., RoboTwin) and plan to extend or migrate RoboMME to such environments in future versions.
>
> **W2: Scope of the benchmark is limited.**
> Thank you for this question. RoboMME is designed to systematically study memory in robotic manipulation under a *controlled and unified setting*. To enable fair comparison and isolate memory-related challenges, we adopt a standardized single-arm tabletop setup, prioritizing controllability over environmental scope at this stage.
>
> We agree that more diverse environments are important. As future work, we plan to extend to richer scenes (e.g., indoor and outdoor) and more complex embodiments (e.g., bimanual and mobile manipulation).
>
> **W3: More analysis of recurrent memory failures.**
> Thank you for the suggestion. We intentionally constrain the capacity of recurrent variants (e.g., number of layers) in order to match the computational budget of other memory representations. We also follow initialization strategies from prior work (e.g., LaCT [1]) and additionally experiment with different optimizers (e.g., Muon [2]) and learning rates for the recurrent modules; but  these did not materially improve performance. Therefore, we only report basic results.
>
> We observe that shallow integration of recurrent modules into modern transformer-based architectures is inherently unstable, especially under long-context settings. This suggests that recurrent memory may require deeper architectural integration or even new VLA designs.
>
> Overall, we view recurrent memory as a promising direction, but unlocking its full potential is beyond the scope of this work. We will include more details of our preliminary findings in the revision.
>
> [1] Zhang, Tianyuan, et al. "Test-time training done right." ICLR 2026.
> [2] Liu, Jingyuan, et al. "Muon is scalable for llm training." arXiv:2502.16982.
>
> **Q2: Error accumulation in real-world experiments.**
>
> Our real-world data collection explicitly includes *failure-recovery trajectories*, enabling the policy to learn self-correction and retry behaviors to mitigate error accumulation. We also collect data under diverse lighting conditions and object configurations to improve robustness to perceptual drift.
>
> In addition, we perform standard calibration (e.g., gravity compensation) and controller sanity checks before starting evaluation. While error accumulation cannot be fully eliminated in robots, these design choices improve the policy’s resilience in real-world settings.
>
> **Q3: What is the task design logic behind the benchmark?**
>
> Thank you for the thoughtful question. Beyond the four memory types, our task design follows two principles: (1) incorporating dynamic tasks, and (2) introducing ambiguity, where identical observations may require different actions depending on history. The 16 tasks are designed to instantiate these properties in a controlled and comparable manner.
>
> We do not claim that the current task set is exhaustive. RoboMME is intended as a first step toward standardizing memory evaluation, rather than covering all real-world scenarios. The task design is naturally extensible, and future versions will incorporate more challenging settings (e.g., dynamic physics, deformable objects, and diverse embodiments).
>
> **Q4: More discussion on the Optimal Memory Mechanism.**
>
> Thank you for the insightful suggestion. Our results show that no single memory representation is universally optimal; instead, different memory types exhibit complementary strengths across tasks.
>
> This suggests that an *"optimal"* memory mechanism should be *hybrid*, dynamically combining multiple forms of memory rather than relying on a single representation. For example, symbolic memory supports high-level reasoning, perceptual memory provides global visual context, and recurrent memory captures local temporal dependencies. One possible direction is a *Mixture-of-Memories* design, analogous to Mixture-of-Experts, where the VLA adaptively selects or integrates different memory types through gating.
>
> We believe RoboMME provides a useful foundation for studying such designs, and we will expand the discussion section in the paper to better articulate this direction toward adaptive memory architectures for robotic policies.

---

### Official Review · Reviewer_jTCo · 2026-03-13

**Soundness:** 3
**Presentation:** 4
**Significance:** 3
**Originality:** 3
**Overall Recommendation:** 5
**Confidence:** 3

**Summary:**

This paper introduces RoboMME, a unified benchmark for history-dependent robotic manipulation. The authors organize memory requirements into four task suites and compare 14 memory-augmented VLA variants, all built on the same $\pi_{0.5}$ backbone, within the benchmark.

**Compliance With Llm Reviewing Policy:**

Affirmed.

**Final Justification:**

I'll keep my score.

**Key Questions For Authors:**

1. Do the trends shown in the experiment part hold true across different backbones? In particular, is the conclusion that "FRAMESAMP+Modul is optimal" specific to the $\pi_{0.5}$ architecture and pre-training method?
2. Is the poor performance of the recurrent memory because recurrent representations are unsuitable for this type of task, or because this paper uses a plug-and-play shallow ensemble approach and does not include specialized pre-training for the recurrent model?

**Limitations:**

Yes

**Strengths And Weaknesses:**

#### **Strengths:**
1. The problem is clearly defined, and the research focus is well-defined. The paper structures memory requirements into four cognitive dimensions, which makes the benchmark’s objectives more specific than those of many general long-horizon benchmarks.
2. The benchmark design offers comprehensive coverage. It addresses four distinct categories of needs, giving it a clear advantage over several existing benchmarks.
3. The key findings of this study are of research value, demonstrating that memory effectiveness is clearly task-dependent.

#### **Weaknesses:**
1. The extension scope of the benchmark experiments is limited. The entire model family is built on a single $\pi_{0.5}$ backbone, and the benchmark is confined to the tabletop ManiSkill environment.
2. Compared to the simulation benchmark, the real-world evaluation is relatively weak. Conducting more real-world experiments will help demonstrate the broad robustness of the conclusions in practical robotics scenarios.

---

> ### Author Rebuttal · Authors · 2026-03-30
>
> We sincerely thank the reviewer for recognizing the contributions of this work and for their thoughtful and constructive feedback. We provide detailed responses to address each of their questions below.
>
> **W1: The benchmark is confined to a tabletop ManiSkill setup.**
>
> Thank you for raising this important point. The primary goal of RoboMME is to systematically study memory in robotic manipulation under a *controlled and unified setting*. To enable fair comparison and isolate memory-related challenges, we adopt a standardized tabletop setup in ManiSkill due to its ease of modification, rather than focusing on environmental diversity at this stage.
>
> We agree that broader and higher-fidelity environments are important. We plan to extend RoboMME to a more comprehensive version (e.g., RoboMME 2.0) with richer scenes, more realistic tasks, and more complex embodiments (e.g., bimanual and mobile manipulation), using higher-fidelity simulators (e.g., RoboTwin or IsaacSim). Our long-term goal is to build comprehensive memory-oriented tasks (including tabletop, indoor, and outdoor scenarios) while preserving the same level of rigor and comparability.
>
> **W1 & Q1: MME-VLA is built only on the $\pi_{0.5}$ backbone. Do the findings generalize to other backbones?**
>
> Thank you for the insightful question. We adopt $\pi_{0.5}$ as the backbone due to its strong performance and GPU efficiency, which enables large-scale controlled ablations. Using a fixed backbone allows us to isolate the effects of different memory representations and integration strategies, leading to clearer conclusions.
>
> We agree that cross-backbone generalization is important. Our design, particularly the memory representations and integration mechanisms, is modular and can be readily applied to other VLA architectures. We therefore expect the key trends to generalize. We are currently extending our experiments to additional backbones (e.g., OpenVLA). Due to the limited rebuttal timeline, we may not be able to include new results, but we will make our best efforts to provide further evidence in the final version.
>
> **W2: Real-world evaluation is relatively weak compared to simulation.**
>
> Our primary goal is to introduce a benchmark for systematically studying memory in robotic manipulation, rather than conducting large-scale real-world evaluation. Accordingly, our real-world experiments serve as targeted validation of whether key findings transfer qualitatively to physical settings.
>
> We replicate four representative tasks on a real robot and observe consistent trends with simulation, which supports our core claim: *insights derived from RoboMME are indicative of real-world behavior*. Nevertheless, we agree that more extensive real-world evaluation would further strengthen this claim.
>
> **Q2: Does the poor performance of recurrent memory reflect its unsuitability for these tasks, or the lack of deep integration and specialized pretraining?**
>
> Our results do not suggest that recurrent memory is inherently unsuitable for RoboMME. Instead, the observed underperformance primarily arises from current integration and training challenges.
>
> In our setup, recurrent modules (e.g., TTT, RMT) are incorporated in a plug-and-play, shallow manner without dedicated pretraining, which makes optimization difficult. In particular, we observe significant training instability under long contexts (e.g., thousands of tokens), such as gradient explosion. While stabilization techniques (e.g., pre-/post-norm) help, they also reduce model expressiveness. Similar issues arise in our pilot experiments with other recurrent architectures (e.g., LSTM, Mamba2), indicating that *shallow integration into pretrained VLA backbones is insufficient*.
>
> We therefore conjecture that effective recurrent memory for VLA models likely requires deeper architectural integration and large-scale pretraining, as demonstrated in recent works (e.g., LaCT [1], tttLRM [2]). Alternatively, more stable designs, such as short-horizon or chunk-level recurrent encoding, may offer a practical compromise. We hope RoboMME can serve as a valuable testbed for studying recurrent memory in VLA systems.
>
> [1] Zhang, Tianyuan, et al. "Test-time training done right." ICLR 2026.
> [2] Wang, Chen, et al. "tttLRM: Test-Time Training for Long Context and Autoregressive 3D Reconstruction." arXiv:2602.20160 (2026).

---

> > ### Author Rebuttal · Reviewer_jTCo · 2026-04-02
> >
> > Thank you for the rebuttal. I have no further questions and I'll keep my score.

---

### Decision · Program_Chairs · 2026-04-30

**Decision:**

Accept (spotlight)

**Comment:**

This paper considers an important topic in robot manipulation---memory.  In this paper, a comprehensive benchmark is proposed and many possible variants are implemented. This could possible give the community more information about how to design memory for VLAs. Thus, it is recommended to accept this paper. The authors are suggested to include the additional results during rebuttal in the final version.